# UrbanFusion: Stochastic Multimodal Fusion for Contrastive Learning of Robust Spatial Representations

## Abstract

Forecasting urban phenomena such as housing prices and public health indicators requires the effective integration of various geospatial data. Current methods primarily utilize task-specific models, while recent generic models for spatial representations often support only limited modalities and lack multimodal fusion capabilities. To overcome these challenges, we present UrbanFusion, a spatial representation model that features Stochastic Multimodal Fusion (SMF). The framework employs modality-specific encoders to process different types of inputs, including street view imagery, remote sensing data, cartographic maps, and points of interest (POIs) data. These multimodal inputs are integrated via a Transformer-based fusion module that learns unified representations. An extensive evaluation across 41 tasks in 56 cities worldwide demonstrates UrbanFusion's strong generalization and predictive performance compared to state-of-the-art GeoAI models. Specifically, it 1) outperforms prior models on location-encoding, 2) allows multimodal input during inference, and 3) generalizes well to regions unseen during training. UrbanFusion can flexibly utilize any subset of available modalities for a given location during both pretraining and inference, enabling broad applicability across diverse data availability scenarios.

## 1 Introduction

Urban areas currently accommodate the majority of the world's population and are expected to absorb billions more in the upcoming decades (United Nations, 2019). This rapid growth has increased the need for accurate forecasting tools to support urban planning and inform sustainable policy decisions through urban analytics (United Nations, 2024; Daniel & Pettit, 2025). Efficient urban operations increasingly depend on precise, location-specific predictions, such as housing price estimation (Yao et al., 2018), mobility prediction (Wiedemann, 2025), and land use classification (Che et al., 2025). These challenges have traditionally been tackled using task-specific models designed for a single domain and geographic region. A widely adopted approach to improve predictive performance is to augment coordinates with geospatial context data (Hong et al., 2023) such as census statistics, business directories, street view imagery, or remote sensing inputs (Mühlematter et al., 2024b; Wang et al., 2023).

However, these models often face limitations due to the availability and quality of context data, which can vary widely across regions (Klemmer et al., 2025). This variability restricts their scalability and applicability, while also making their development costly and dependent on domain expertise (Koldasbayeva et al., 2024). Recently, the rise of foundation models in language (Brown et al., 2020), vision (Assran et al., 2023), and multimodal domains (Radford et al., 2021) has inspired efforts to build general-purpose geospatial models, commonly referred to as *Geo-Foundation Models* (*GeoFMs*) (Mai et al., 2025; Jakubik et al., 2023).

Inspired by the success of language-image models like CLIP (Radford et al., 2021), recent research often uses self-supervised learning to align spatial coordinates (longitude and latitude) with other data types, such as satellite or street view imagery (Vivanco et al., 2023; Klemmer et al., 2025; Liu et al., 2025). These works offer general, task-agnostic representations of geographic locations that were shown to improve performance across a wide range of downstream tasks and domains. Nev-

Figure 1: *UrbanFusion.* Pretrained modality-specific encoders extract features projected into tokens. After random token masking, a Transformer fuses the tokens. The output feeds into two heads: one for **Contrastive Location Alignment (CL)**, the other for **Latent Modality Reconstruction (Rec.)**. For downstream tasks, coordinates or available modalities are input into the frozen encoder (green arrows) to obtain feature vectors for training downstream models.

ertheless, they are limited by the modalities they can support. The requirement for *paired samples* of all modalities at each location, coupled with the use of pairwise contrastive loss between modalities, makes the inclusion of additional modalities challenging. Moreover, each modality is treated independently, missing the opportunity to learn richer representations by modeling their interactions (Dufumier et al., 2025). Many geospatial tasks, however, rely on the combined signals of diverse spatial data. For instance, housing prices may depend not only on visual features from imagery but also on nearby infrastructure and services, such as road networks or points of interest (POIs) (Yao et al., 2018). To effectively support such tasks, models must go beyond isolated modality processing and instead learn representations that reflect the complex, layered nature of urban environments (Mai et al., 2025).

To bridge these gaps, we present *UrbanFusion*, a novel location embedding tailored for urban environments. *UrbanFusion* integrates multiple spatial modalities across various scales, including remote sensing imagery, street view images, cartographic basemaps, and POIs, into compact, multi-scale, task-agnostic embeddings. To better capture the multimodal context, we propose a training framework called *Stochastic Multimodal Fusion* (*SMF*), which combines contrastive learning with self-supervised reconstruction. At each training step, the model samples two distinct subsets of available modalities, aligns their embeddings, and reconstructs the modalities.

UrbanFusion not only outperforms state-of-the-art methods across diverse tasks, but also offers significant advantages: (1) it efficiently supports multiple modalities with minimal training overhead through modality masking, (2) enables joint training and inference on heterogeneous datasets containing arbitrary subsets of modalities, (3) adeptly learns to represent modality-specific features while integrating shared and synergistic information via *SMF*, and (4) generalizes well to regions unseen during training.

In summary, our contributions are as follows:

1. We propose *Stochastic Multimodal Fusion* (*SMF*), a model-agnostic contrastive learning framework that jointly captures modality-specific, shared, and synergistic information. While developed for spatial data, *SMF* is broadly applicable to general multimodal learning.

2. We introduce *UrbanFusion*, the first location embedding model to flexibly integrate street view imagery, remote sensing data, cartographic basemaps, and POIs into unified spatial representations.

3. We conduct a large-scale evaluation of *UrbanFusion* on 41 downstream tasks, including housing price prediction, healthcare, environmental variable estimation, land use and land cover classification, demographic inference, urban perception prediction, and energy con-

sumption forecasting. Our results show that *UrbanFusion* consistently outperforms state-of-the-art models across these domains.

## 2 RELATED WORK

| Method | Street view | Satellite | Maps | POIs | Tags | Others | Training Data | Loss | Inter. (PID) | Code | Weights |
|---|---|---|---|---|---|---|---|---|---|---|---|
| *Location embedding models* | | | | | | | | | | | |
| **UrbanFusion (ours)** | ✓ | ✓ | ✓ | ✓ | ✗ | ✗ | PP2-M (73k) | CL + rec. | R+U+S | ✓ | ✓ |
| **GeoCLIP (Vivanco et al., 2023)** | ✓ | ✗ | ✗ | ✗ | ✗ | ✗ | MP-16 (4.7m) | CL | R | ✓ | ✓ |
| **SatCLIP (Klemmer et al., 2025)** | ✗ | ✓ | ✗ | ✗ | ✗ | ✗ | S2-100K (100k) | CL | R | ✓ | ✓ |
| **GAIR (Liu et al., 2025)** | ✓ | ✓ | ✗ | ✗ | ✗ | ✗ | Streetscapes1M (1m) | CL | R | ✗ | ✗ |
| **CSP (Mai et al., 2023)** | ✓ | ✗ | ✗ | ✗ | ✗ | ✗ | iNat2018 (438k) | CL | R | ✓ | ✓ |
| | ✗ | ✓ | ✗ | ✗ | ✗ | ✗ | fMoW (364k) | CL | R | ✓ | ✓ |
| *Local Models for Location Encoding* | | | | | | | | | | | |
| **GPS2Vec (Yin et al., 2019)** | ✗ | ✗ | ✗ | ✗ | ✓ | ✗ | Flickr tags (1m) | Rec. | R | ✗ | ✓ |
| **GPS2Vec+ (Yin et al., 2021)** | ✓ | ✗ | ✗ | ✗ | ✓ | ✗ | YLI-GEO (6m) | Rec. | R | ✗ | ✓ |
| **PDFM (Agarwal et al., 2024)** | ✗ | ✗ | ✗ | ✗ | ✗ | ✓ | Google (35k) | Rec. | R+U+S | ✗ | ✗ |

Table 1: **Summary of existing and proposed methods for spatial representation learning.** Modalities and open access (✓ = available, ✗ = unavailable) are shown as binary indicators. Pre-training sources are listed with the number of training locations in parentheses. Training objectives include contrastive loss (CL) and reconstruction loss (Rec.). Interaction modeling (Inter.) is analyzed via partial information decomposition (PID), which quantifies modality interactions in terms of *redundancy* (R), *uniqueness* (U), and *synergy* (S).

**Spatial Representation Learning.** A core challenge in geospatial machine learning is learning transferable representations of locations from spatially distributed, heterogeneous data. A substantial body of work has focused on *local* spatial representation learning, where models are trained for specific cities or regions, often using self-supervised objectives (Jenkins et al., 2019; Wang et al., 2020; 2025). For instance, *GPS2Vec* created local encoders for 120 UTM zones based on Flickr image tags (Yin et al., 2019). Subsequent studies expanded this approach by incorporating visual features into the representations (Yin et al., 2021). The *Population Dynamics Foundation Model* (*PDFM*) used graph neural networks to learn multimodal embeddings for county and postal code regions in the US, integrating signals from Internet search trends, Google Maps data, activity levels, and environmental factors to model population dynamics (Agarwal et al., 2024). However, this research did not address geographic generalization beyond those areas represented in the training data.

This *local* trend evolved into *global* coordinate encoders inspired by CLIP-style contrastive training (Radford et al., 2021). For example, the *CSP* model aligned encoded coordinates with images from the iNaturalist and FMoW datasets (Mai et al., 2023). Meanwhile, *GeoCLIP* employed a similar approach, using *Random Fourier Features* to encode coordinates and aligning them with street view imagery (Vivanco et al., 2023; Tancik et al., 2020). To tackle the issue of unevenly distributed image data in previous methods, *SatCLIP* introduced a global model that utilized *Spherical Harmonics* and SIREN networks for coordinate encoding, learning representations through contrastive alignment with globally distributed Sentinel-2 satellite imagery (Klemmer et al., 2025; Rußwurm et al., 2024). Most recently, *GAIR* pioneered a multimodal setting by training encoders for coordinates, street view imagery, and satellite image modalities, employing a pairwise contrastive loss framework (Liu et al., 2025). Beyond coordinate–image contrastive encoders, recent research has explored richer multimodal and geometry-aware spatial representations. UrbanCLIP (Yan et al., 2024) and ReFound (Xiao et al., 2024) leverage web-scale text–image alignment to enhance urban region profiling, GeoLink (Bai et al., 2025) fuses satellite imagery with OSM-derived structural vectors, highlighting the complementary nature of cartographic and visual signals.

A detailed overview of all baseline models is provided in Table 1 and Appendix F.3. Our work builds upon these approaches by integrating new modalities essential for urban prediction tasks and proposing a cohesive, unified multimodal encoder trained using a novel technique that combines contrastive loss with masking.

**On the Limitations of Multimodal Contrastive Alignment.** While contrastive learning-based models demonstrated strong performance by aligning paired samples across different modalities, the information preserved in their embeddings is inherently limited. Contrastive losses, such as InfoNCE (van den Oord et al., 2018), primarily capture *redundant* information between modali-

ties while neglecting *unique* modality-specific content and failing to capture *synergistic* interactions between them (Dufumier et al., 2025). This decomposition of information is formalized by the partial information decomposition (PID) framework (Williams & Beer, 2010; Dufumier et al., 2025), which provides a principled way to disentangle the different types of information shared between input modalities and a target variable. Prior work has explored intra-modal alignment through data augmentations, aiming to capture the *uniqueness* of individual modalities (Liang et al., 2023; Yuan et al., 2021) or even their *synergistic* relationships (Dufumier et al., 2025). However, such augmentations are often handcrafted and not well defined across all modalities. Another approach involves retrieval-augmented generation (RAG) (Lewis et al., 2020), in which the model's representations can be used to query a database. This strategy has also been explored in the context of location representation learning (Dhakal et al., 2025). See Appendix D.1 for additional discussion. We propose a novel integration of reconstruction loss and random modality masking to mitigate the shortcomings of using contrastive loss alone.

## 3 METHODS

### 3.1 ARCHITECTURE AND TRAINING WITH STOCHASTIC MULTIMODAL FUSION (SMF)

Our overall model architecture is illustrated in Figure 1. Let $\mathcal{A}_i = \{m_1, m_2, \ldots, m_K\}$ be a set of available input modalities at a location $i$. Each modality $m \in \mathcal{A}_i$ is processed via a fixed, pretrained encoder $f_m$ to extract a latent feature vector $\mathbf{h}_m$, which is then projected into a token $\mathbf{t}_m \in \mathbb{R}^d$. The tokens are fused by a Transformer encoder $\mathcal{T}_\theta$, parameterized by $\theta$, followed by average pooling to obtain the final representation $\mathbf{z}_i = \mathcal{T}_\theta(\{\mathbf{t}_m : m \in \mathcal{A}_i\}) \in \mathbb{R}^d$ used for downstream tasks.

During training, we randomly mask a subset of modalities $\mathcal{M}_i \subset \mathcal{A}$, and denote the inverse non-masked complement subset as $\overline{\mathcal{M}}_i = \mathcal{A} \setminus \mathcal{M}_i$. Both $\mathcal{M}_i$ and $\overline{\mathcal{M}}_i$ are passed through the modality-specific encoders $f_m$ and the multimodal Transformer encoder $\mathcal{T}_\theta$, yielding embeddings $\mathbf{z}_i^{\mathcal{M}}$ and $\mathbf{z}_i^{\overline{\mathcal{M}}}$, respectively. Finally, two decoder heads are applied on top of the fused representation, each implemented as a lightweight projection network. The first head performs **Contrastive Location Alignment**: the fused embeddings $\mathbf{z}_i$ and $\mathbf{z}_i^{\overline{\mathcal{M}}}$, derived from the masked subset $\mathcal{M}_i$ and its complement $\overline{\mathcal{M}}_i$, respectively, are passed through a shared decoder $\mathcal{E}$ and aligned via a symmetric InfoNCE objective (van den Oord et al., 2018), formalized as:

$$\mathcal{L}_{\text{contr}} = -\frac{1}{N} \sum_{i=1}^{N} \left[ \log \frac{f(\mathbf{z}_i^{\mathcal{M}}, \mathbf{z}_i^{\overline{\mathcal{M}}})}{\sum_{j=1}^{N} f(\mathbf{z}_i^{\mathcal{M}}, \mathbf{z}_j^{\overline{\mathcal{M}}})} + \log \frac{f(\mathbf{z}_i^{\overline{\mathcal{M}}}, \mathbf{z}_i^{\mathcal{M}})}{\sum_{j=1}^{N} f(\mathbf{z}_i^{\overline{\mathcal{M}}}, \mathbf{z}_j^{\mathcal{M}})} \right],$$

$$f(\mathbf{a}, \mathbf{b}) \coloneqq \exp\left( \frac{\text{sim}(\mathcal{E}(\mathbf{a}), \mathcal{E}(\mathbf{b}))}{\tau} \right),$$

where $N$ denotes the number of training examples in the mini-batch. Here, $\text{sim}(u, v) = \frac{u^\top v}{\|u\|\|v\|}$ denotes cosine similarity and $\tau$ is a learnable temperature. Views derived from the same geographic location form *positive pairs*, while views from different locations act as *negatives*. It is important to note that at both pretraining and inference time, the model can flexibly operate on an arbitrary subset of modalities, depending on data availability and task-specific requirements. For example, consider the case where six modalities exist in total, but at location $i$ data is available for only three: $a$, $b$, and $c$. In this case, the training loss can be computed using $\mathcal{M}_i = a$ and $\overline{\mathcal{M}}_i = b, c$. This flexibility enables the combination of large-scale datasets with differing modality compositions for pretraining.

The second head performs **Latent Modality Reconstruction**: a modality-specific projection network $g_m$ is trained to reconstruct the latent vector $\mathbf{h}_m$ for *all* modalities $m \in \mathcal{A}$, based on the fused representation $\mathbf{z}_i$. The loss is computed as the average mean squared error over all modalities:

$$\mathcal{L}_{\text{recon}} = \frac{1}{2|\mathcal{A}|} \sum_{m \in \mathcal{A}} \left( \|g_m(\mathbf{z}_i^{\mathcal{M}}) - \mathbf{h}_m\|_2^2 + \|g_m(\mathbf{z}_i^{\overline{\mathcal{M}}}) - \mathbf{h}_m\|_2^2 \right),$$

where all latent features are z-score normalized (per dimension, mean = 0, variance = 1) to ensure equal contribution across modalities. By operating in the latent space, the reconstruction reduces the computational cost compared to full input reconstruction and encourages the model to focus on abstract representations rather than reconstructing input noise (Assran et al., 2023). The total training loss is a weighted sum of the two objectives: $\mathcal{L}_{\text{total}} = (1 - \lambda) \cdot \mathcal{L}_{\text{contr}} + \lambda \cdot \mathcal{L}_{\text{recon}}$, where $\lambda$ controls the balance between alignment and reconstruction.

This training method, termed *Stochastic Multimodal Fusion*, enables the model to capture modality-interactions beyond redundant information, alleviating the shortcomings of contrastive loss outlined above. Intuitively, the reconstruction loss together with random masking of modalities encourages the model to retain unique information from individual modalities as well as synergistic information from sets of modalities that could help to reconstruct another modality, retaining similar information as required to solving downstream tasks. This is formalized as follows:

**Lemma 1:** Assume that for each downstream task $Y$, there exists at least a proxy modality or subset $S_Y \subseteq \mathcal{A}$ such that predicting $S_Y$ is at least as demanding as predicting $Y$: $I(\mathcal{A} \setminus S_Y; Y) \leq I(\mathcal{A} \setminus S_Y; S_Y)$. Under this assumption, the *SMF* loss ($\mathcal{L}_{\text{total}}$) encourages $\mathcal{T}_\theta$ to retain *redundant*, *synergistic*, and *unique* information, thereby maximizes a lower bound on $I(m_1, \ldots, m_K; Y) = R + S + \sum_i^K U_i$.

**Proof:** For two random (masked/complement) views of modalities from the same location, $\mathcal{L}_{contr}$ maximizes a variational lower bound on the mutual information between the corresponding representations:
$$I(\mathbf{z}^{\mathcal{M}}; \mathbf{z}^{\overline{\mathcal{M}}}) \geq \log N - \mathcal{L}_{\text{contr}},$$
where $N$ is the batch size. Thus $\mathcal{L}_{contr}$ increases a computable lower bound on cross-modal shared information, i.e., *redundant information $R$* as shown by Oord et al. (2018); Dufumier et al. (2025). (For clarity we omit expectation notation; all bounds are understood in expectation over the data, the masking distribution, and negative sampling.)

The reconstruction head reconstructs the latent $\mathbf{h}_m$ from the fused representation $\mathbf{z}$ with $\mathcal{L}_{recon}$. Minimizing this *MSE* loss is equivalent to maximizing the average log-likelihood under a Gaussian variational decoder $q_\phi^{(m)}$ with fixed covariance Bishop & Nasrabadi (2006):

$$q_\phi^{(m)}(\mathbf{h}_m \mid \mathbf{z}) = \mathcal{N}\big(\mathbf{h}_m; g_\phi^{(m)}(\mathbf{z}), \sigma_m^2 I\big), \tag{1}$$

$$\log q_\phi^{(m)}(\mathbf{h}_m \mid \mathbf{z}) = -\tfrac{1}{2\sigma^2} \| g_\phi^{(m)}(\mathbf{z}) - \mathbf{h}_m \|^2 - \tfrac{d_m}{2} \log\big(2\pi\sigma_m^2\big) \tag{2}$$

where $d_m$ is the dimensionality of $\mathbf{h}_m$. Using

$$I(\mathbf{z}; \mathbf{h}_m) = H(\mathbf{h}_m) - H(\mathbf{h}_m \mid \mathbf{z}), \ H(\mathbf{h}_m \mid \mathbf{z}) = -\log p(\mathbf{h}_m \mid \mathbf{z}), \tag{3}$$

$$I(\mathbf{z}; \mathbf{h}_m) = H(\mathbf{h}_m) + \log p(\mathbf{h}_m \mid \mathbf{z}) \tag{4}$$

and replacing the intractable true conditional $p(\mathbf{h}_m \mid \mathbf{z})$ with the $q_\phi^{(m)}$, we obtain the Barber–Agakov lower bound Barber & Agakov (2004)

$$I(\mathbf{z}; \mathbf{h}_m) \geq H(\mathbf{h}_m) + \log q_\phi^{(m)}(\mathbf{h}_m \mid \mathbf{z}). \tag{5}$$

Since $H(\mathbf{h}_m)$ is constant, minimizing $\mathcal{L}_{recon}$ directly maximizes a computable lower bound on $I(\mathbf{z}; \mathbf{h}_m)$. Since random subsets of modalities $M$ are masked during *SMF* training (see Section 3.1, any set of modalities will be used at some point to create the embedding $\mathbf{z}$. When $|M| = 1$, $M$ contains a single input modality, the objective preserves that modality's *unique* information $U$; when $|M| \geq 2$, $M$ contains multiple modalities, simultaneously reconstructing all $\{\mathbf{h}_m\}$ forces $\mathbf{z}$ to integrate complementary cues, thereby promoting *synergistic $S$* information.

Finally, under Assumption 1, for each task $Y$ there exists at least one proxy subset $S_Y$ such that predicting $S_Y$ is at least as demanding as predicting $Y$. Hence minimizing the latent reconstruction loss maximizes Barber–Agakov–style computable lower bounds $\{I(\mathbf{z}; \mathbf{h}_m)\}_{m \in S_Y}$, prevents $z$ from discarding information required for $Y$. Together with the symmetric InfoNCE loss which increases the bound on $I(\mathbf{z}^{\mathcal{M}}; \mathbf{z}^{\overline{\mathcal{M}}})$, the total loss $\mathcal{L}_{\text{total}}$ maximizes two tractable mutual information surrogates and encourages retention of the PID components (*redundant*, *unique*, and *synergistic*) that matter for downstream tasks.

## 3.2 Encoders and Data

To ensure fair comparisons and consistency with prior work, we adopt modalities and encoders closely aligned with those used in previous studies. Wherever applicable, we use the same architectures and freeze the encoder weights during training, following standard practice in *GeoCLIP*, and *SatCLIP*. Although end-to-end fine-tuning may yield additional performance gains, particularly when combined with parameter-efficient strategies (Hu et al., 2022; Mühlematter et al., 2024), we leave this direction for future work.

For pretraining, we build upon the Place Pulse 2.0 (PP 2.0) dataset (Dubey et al., 2016), which contains 110'988 locations with corresponding geographic coordinates and street-view images across 56 cities spanning all continents except Antarctica. From the PP 2.0 dataset, we hold out six cities for testing *Cross-Regional Generalization* on unseen regions. The remaining data is split into 80 percent for training and 20 percent for validation. While PP2.0 provides core geographic signals in the form of coordinates and street-view imagery, we further enrich each location with additional geospatial modalities to provide a more comprehensive representation of place. We refer to the resulting enriched multimodal dataset as **PP2-M**, which we release publicly for reproducibility and future research. Below, we describe each modality and the corresponding encoder used in our model.

**Coordinates.** We represent geographic coordinates (longitude and latitude) using the Equal Earth projection (Šavrič et al., 2019), followed by *Random Fourier Features* computed at multiple spatial scales (Tancik et al., 2020). These features are passed through a multi-layer perceptron (MLP). This approach, also adopted in *GeoCLIP* and *GAIR*, helps the model capture location information across resolutions (Vivanco et al., 2023; Liu et al., 2025).

**Street-view imagery.** We encode the PP2.0 images using the CLIP ViT-L/14 model, pretrained on large-scale image-text datasets (Radford et al., 2021). Thanks to its strong generalization capabilities and prior success in *GeoCLIP* (Vivanco et al., 2023), this model provides a reliable and consistent basis for evaluation.

**Remote sensing imagery.** We enrich the dataset with 12-channel multispectral Sentinel-2 imagery for each location (Drusch et al., 2012), following prior work that incorporates large-scale contextual signals (Klemmer et al., 2025). For encoding, we use the ViT-S/16 model pretrained on Sentinel-2 data (Wang et al., 2022), adopting the same configuration as in *SatCLIP* (Klemmer et al., 2025).

**Cartographic basemaps.** We extract cartographic basemaps from OpenStreetMap (OpenStreetMap contributors, 2017), generating patches at 300 m, 600 m, and 1200 m resolutions. These maps provide a multi-scale, human-interpretable source of geospatial information, including buildings, land cover, and transportation infrastructure (Mühlematter et al., 2024a). To encode the maps, we pretrain a ViT-B/16 backbone using the Masked Autoencoder (MAE) framework (He et al., 2022). The model is initialized with ImageNet-pretrained weights.

**POIs.** We augment each location with POI data from OpenStreetMap (OpenStreetMap contributors, 2017), collected within a 200 m radius. For each location, we extract the 15 nearest POIs and generate textual prompts describing their names, categories, and distances. These prompts are encoded using the BAAI/bge-small-en-v1.5 language model (Xiao et al., 2023), similar to prior work (Wang et al., 2025).

**Multimodal Fusion Encoder.** We use a single-block transformer with an embedding dimension of 768, eight attention heads, and learned positional encodings (Vaswani et al., 2017).

Further information about the dataset and encoders can be found in Appendix E.1 and F.1.

## 4 Experiments

### 4.1 Experimental Setting

We report results on *downstream prediction tasks* in urban environments. A small downstream model (linear regression or small MLP) is trained to predict the labels based on the location embedding generated with the pretrained and frozen *UrbanFusion* model (see Figure 1). We first test *Coordinate-Only Spatial Encoding*, where the pretrained base models receives only geographical coordinates as input, and secondly investigate *Multimodal Spatial Encoding*, which uses additional modalities also at inference time. Both approaches are evaluated on out-of-sample locations within the same geographic region as the training data, representing an interpolation setting. To assess extrapolation, we test *Cross-Regional Generalization* by applying models to cities entirely outside

the spatial extent of the training data. *UrbanFusion* and other methods trained on the PP2-M dataset are pretrained for 400 epochs. Pretraining details, training accuracy, and loss curves, are provided in Appendix F.2.

We use a large suite of urban prediction datasets. Our primary source is the PP2-M dataset (Dubey et al., 2016), from which we select out-of-sample locations and assign target variables for multi-modal tasks. Alternatively, for some *Coordinate-Only* downstream tasks, we use the locations provided directly by the corresponding task-specific datasets. We evaluate regression tasks involving **Housing Prices**, **Energy Consumption**, **Urban Perception**, **Crime Incidence**, and postal code-level **Health**, **Socioeconomic**, and **Environmental Indicators**. Classification tasks include **Land Cover Prediction** and **Coarse-to-Fine Land Use Classification** in Europe. Ridge regression and small MLPs are used for regression, while logistic regression and small MLPs are used for classification.

We compare *UrbanFusion* against the most relevant existing methods: *SatCLIP* (Klemmer et al., 2025), *GeoCLIP* (Vivanco et al., 2023), and *GAIR* (Liu et al., 2025). To ensure a fair evaluation, we include both the original versions of these models as well as variants trained on the PP2-M dataset for 400 epochs each. Additionally, we evaluate against other location representation approaches, including *CSP* (Mai et al., 2023), as well as local models such as both versions of *GPS2Vec* (Yin et al., 2019; 2021) and *PDFM* (Agarwal et al., 2024). As a simple baseline, we also include an *Identity* model that uses raw geographical coordinates directly as input, without any transformation. In the following result tables, we indicate below each method the dataset on which the model was trained.

Following prior work (Klemmer et al., 2025), raw geographical coordinates are concatenated to the model embeddings for evaluation. For each task, the dataset is split into 60% for training, 20% for validation and hyperparameter tuning, and 20% for testing. We report linear probing performance on the held-out test set, with hyperparameters optimized using the Optuna framework (Akiba et al., 2019). Complete results, including MLP performance and additional baselines, are provided in Appendix B, with further details on the datasets, baselines, evaluation protocols available in Appendix E.2, F.3, and G, respectively.

### 4.2 COORDINATE-ONLY SPATIAL ENCODING

Table 2 presents linear probing results for a widely studied use case of location representations: generating embeddings from raw geographical coordinates. *UrbanFusion* outperforms all other methods on 5 out of 8 datasets. Notably, *UrbanFusion* consistently outperforms the most closely related methods when all models are trained on the PP2-M dataset. The local model *GPS2Vec* achieves superior performance on the Housing Prices and Energy Consumption prediction tasks compared to *UrbanFusion*. This is at least partially due to the epistemic uncertainty in *UrbanFusion*, stemming from the limited training set of only 2'146 locations in the covered regions, whereas *GPS2Vec* may benefit from a denser coverage, as suggested by the significantly larger training dataset (see Table 1). Notably, *UrbanFusion* surpasses even Google's *PDFM*, a domain-specific model specifically designed for ZIP code prediction, on this task, which we find particularly surprising. *PDFM* performs best on ZIP Code-level health indicator tasks, potentially due to the inclusion of internet search trends. Notably, it is the only model in this comparison where evaluated locations are not out-of-sample, as the published representations correspond to in-sample training locations.

In contrast to prior work by Klemmer et al. (2025), but consistent with findings from Agarwal et al. (2024), *GeoCLIP* consistently outperforms *SatCLIP*, even when evaluated against the proposed fine-grained model, *L40*. This performance gap can be attributed to two key factors: (1) satellite imagery captures broader spatial context compared to street-view images, providing less fine-grained information; and (2) *Random Fourier Features* offer a more effective representation of high-frequency spatial variations than *Spherical Harmonics* (see Tancik et al. (2020) and Ji et al. (2024)). A visual analysis (Appendix Figure 4) further supports this finding, showing the embeddings reduced to 3D by PCA and mapped to RGB color codes. While *UrbanFusion* produces smooth and fine-grained representations, *GeoCLIP* exhibits less spatial granularity, likely due to the absence of explicitly multimodal and multiscale inputs. In contrast, *SatCLIP*'s location encoder fails to adequately model high-frequency intra-city variation, limiting its performance on urban prediction tasks.

| | UrbanFusion | GAIR | GeoCLIP | SatCLIP$_{L40}$ | GeoCLIP | SatCLIP$_{L40}$ | GPS2Vec | | PDFM | Identity |
|---|---|---|---|---|---|---|---|---|---|---|
| | | | PP2-M | | MP-16 | S2-100K | tag | visual | Google | $y \sim g(c)$ |
| *Regression* (%R$^2$ ↑) | | | | | | | | | | |
| Housing Prices | [78.7] | 78.5 | 78.4 | 72.7 | 78.6 | 73.1 | **79.2** | 79.0 | - | 66.6 |
| Energy Consumption | [20.1] | 18.4 | 18.5 | 2.6 | 18.7 | 3.3 | **22.3** | 20.0 | - | 1.5 |
| Crime Incidence | [**87.4**] | 85.4 | 84.0 | 65.9 | 86.3 | 61.5 | 84.4 | 76.5 | 74.5 | 22.1 |
| Urban Perception (avg. 6 tasks*) | [**9.5**] | 7.8 | 8.0 | 6.1 | 8.1 | 5.7 | - | - | - | 1.3 |
| ZIP Code (weighted avg. 29 tasks*) | [**74.3**] | 64.6 | 67.1 | 54.4 | 66.9 | 52.3 | 55.6 | 48.6 | 74.0 | 3.0 |
| *Classification* (%F1↑) | | | | | | | | | | |
| Land Cover | [**56.9**] | 53.3 | 53.2 | 46.4 | 54.6 | 45.9 | 53.3 | 54.4 | 51.3 | 34.4 |
| Land Use – Coarse | [**58.9**] | 57.3 | 57.5 | 57.4 | 57.0 | 53.2 | 58.6 | 54.2 | - | 48.2 |
| Land Use – Fine | 47.7 | [**49.9**] | 48.3 | 48.0 | 48.6 | 45.7 | 48.3 | 46.6 | - | 42.7 |

* Detailed results in Tables 14 and 15.

Table 2: **Evaluation of coordinate-only spatial encoding.** Best results are shown in **bold**, second-best are underlined, and top scores across all models trained on PP2-M are indicated in [brackets].

## 4.3 MULTIMODAL SPATIAL ENCODING

Table 3 presents results using linear probing for an additional use case of *GeoFMs*: incorporating not only coordinates but also auxiliary location information such as satellite or street view imagery. Since the benefit of a specific modality often depends on the downstream task, we select the subset of modalities for each model based on validation performance. While *UrbanFusion* is the only model that natively supports multimodal inputs by fusing them into a single embedding vector, we compare to the baseline models by concatenating the representations from the individual modality encoders. For example, for *GAIR*, we concatenate representations derived separately from coordinates, satellite, and street-view inputs.

*UrbanFusion* outperforms all other methods in 4 out of 6 downstream tasks when they are trained on the PP2-M dataset, underperforming only in land cover and coarse land use classification tasks. Further investigation (see Appendix C.7) revealed that simply concatenating the output of encoding models results in better performance for UrbanFusion in these particular tasks. This suggests that such tasks may not necessitate a fused representation. The closest competitor overall is *GeoCLIP*, trained on the MP-16 dataset, which achieves the best performance on 3 out of 6 datasets. It is worth emphasizing that *GeoCLIP* is trained on approximately 65 times more locations than *UrbanFusion* (see Table 1).

| | UrbanFusion | GAIR | GeoCLIP | SatCLIP$_{L40}$ | GeoCLIP | SatCLIP$_{L40}$ | GPS2Vec | | PDFM | Identity |
|---|---|---|---|---|---|---|---|---|---|---|
| | | | PP2-M | | MP-16 | S2-100K | tag | visual | Google | $y \sim g(c)$ |
| *Regression* (%R$^2$ ↑) | | | | | | | | | | |
| Crime Incidence | [**88.5**] | 85.4 | 84.0 | 69.1 | 86.3 | 66.9 | 84.4 | 76.5 | 74.5 | 22.1 |
| Urban Perception (avg. 6 tasks*) | [**18.8**] | 17.4 | 15.5 | 9.5 | **19.2** | 9.5 | - | - | - | 1.3 |
| ZIP Code (weighted avg. 29 tasks*) | [**75.1**] | 70.5 | 70.0 | 69.7 | 69.2 | 68.8 | 55.6 | 48.6 | 74.0 | 3.0 |
| *Classification* (%F1↑) | | | | | | | | | | |
| Land Cover | 65.6 | 65.4 | [67.1] | 56.1 | **69.1** | 56.3 | 53.3 | 54.4 | 51.3 | 34.4 |
| Land Use – Coarse | 59.3 | [61.7] | 61.6 | 57.2 | **62.2** | 57.3 | 58.6 | 54.2 | - | 48.2 |
| Land Use – Fine | [**55.2**] | 54.7 | 54.2 | 49.2 | 55.1 | 47.3 | 48.3 | 46.6 | - | 42.7 |

*Detailed results in Tables 18 and 19.

Table 3: **Evaluation of multi-modal spatial encoding.** Best results are shown in **bold**, second-best are underlined, and top scores across all models trained on PP2-M are indicated in [brackets].

## 4.4 CROSS-REGIONAL GENERALIZATION

As training a global, fine-grained *GeoFM* for location representations is often infeasible due to compute and data limitations, a third use case of such models is zero-shot generalization to unseen regions. Since coordinate encodings do not generalize well to unseen regions, the model receives only non-coordinate modalities at inference time, using the same multimodal evaluation setup as in the previous section. Table 4 presents results using linear models on cities that were held out during training. To ensure that no model has seen samples from the unseen regions during training, we evaluate only models trained on the PP2-M dataset, excluding the selected cities during training.

*UrbanFusion* outperforms other methods, ranking first on 5 out of 6 tasks. The closest competitor is *GAIR*, which performs best on one tasks. Both models clearly outperform baselines with only a

single modality aside from coordinates, such as *SatCLIP* and *GeoCLIP*, highlighting the benefits of multimodal representation learning for geographic generalization.

| | UrbanFusion | GAIR | GeoCLIP PP2-M | SatCLIP$_{L10}$ | SatCLIP$_{L40}$ | Identity $y \sim g(c)$ |
|---|---|---|---|---|---|---|
| *Regression* (%R$^2$ ↑) | | | | | | |
| Crime Incidence | **76.7** | 68.0 | 44.3 | 63.4 | 63.6 | 10.4 |
| Urban Perception (avg. 6 tasks*) | **21.2** | 20.4 | 20.0 | 12.9 | 13.2 | 6.6 |
| ZIP Code (weighted avg. 29 tasks*) | 56.7 | **62.5** | 42.1 | 60.8 | 59.8 | 17.7 |
| | | | | | | |
| *Classification* (%F1↑) | | | | | | |
| Land Cover | **70.9** | 69.9 | 68.6 | 61.3 | 61.1 | 53.9 |
| Land Use – Coarse | **66.7** | 65.9 | 60.6 | 60.6 | 59.4 | 55.1 |
| Land Use – Fine | **61.0** | 60.4 | 55.3 | 53.5 | 53.7 | 49.5 |

*Detailed results in Tables 23 and 24.

Table 4: **Evaluation of cross-region spatial encoding.** Best results are shown in **bold**, second-best are underlined.

Additional insights into the superior performance of multimodal models over single-modal models on various tasks are provided by the k-means clusters shown in Figure 2. While *UrbanFusion* produces spatially smooth clusters that still preserve high-frequency variations, the street-view representations of *GeoCLIP* lack spatial smoothness despite the obvious conceptual similarities of nearby city districts. In contrast, the satellite-view representations of *SatCLIP* fail to capture high-frequency changes.

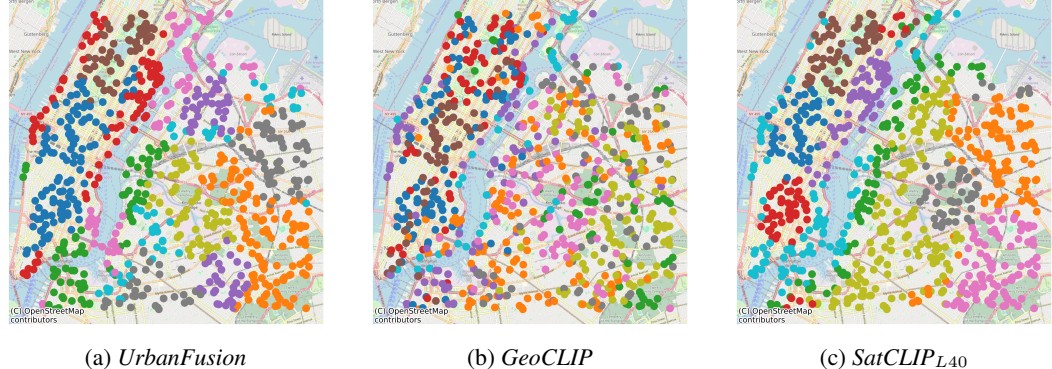

(a) *UrbanFusion*      (b) *GeoCLIP*      (c) *SatCLIP$_{L40}$*

Figure 2: Visual comparison of *Multimodal* location embeddings, all trained on the PP2-M dataset. Embeddings are grouped into 10 clusters using $k$-means.

## 4.5 EMPIRICAL ANALYSIS OF INFORMATION PRESERVATION IN GEOFM MODELS FOR LOCATION REPRESENTATION

As discussed in Section 2 and 3.1, *UrbanFusion* captures *synergistic* and *unique* information, unlike other methods that rely on contrastive losses. To validate this property empirically, we construct synthetic data with random coordinates and two synthetic modalities. We assign to each modality two feature dimensions within the range $[0, 1]$, which uniquely identify the location. These dimensions constitute the *redundant* information shared across both modalities and the coordinate representation, as illustrated in Figure 6 in Appendix C.2. Designing feature dimensions that capture solely *unique* (modality-specific) information is more challenging. Even randomly sampled noise per location can unintentionally assist geolocalization. To address this, we introduce a third feature dimension to each modality. During training, this dimension is batch-augmented: a single random value is sampled from the range $[0, 1]$ and assigned to all locations within the batch. During inference, this dimension contains independently sampled values per location. This strategy ensures that the third feature dimension has zero mutual information with the geolocation task, thereby functioning as a truly *unique* signal. We then evaluate three downstream tasks on the synthetic data:

Geolocalizaition (requires redundant information), predicting the unique features (requires retaining unique information), and predicting the sum of unique features (requires synergistic information). with *unique* random values per modality, location-specific values as *redundant* information, and predicting the sum of the unique values as *synergistic* information (see Appendix C.2 for details). As shown in Figure 3a, contrastive-only methods fail to capture the full information spectrum. *Geo-CLIP* cannot recover *synergistic* content due to its contrastive-learning design. *GAIR* shows some *unique* feature recovery, likely due to convergence artifacts rather than objective design. This is supported by average *unique* signal contributions in the first encoder layer (Figure 3b). *UrbanFusion* reliably captures all mutual information components. Finally, our results are in line with real-world findings in Dhakal et al. (2025), that contrastive methods often neglect task-relevant unique signals.

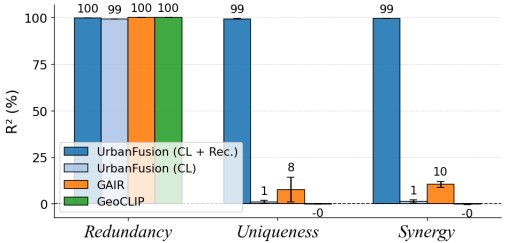
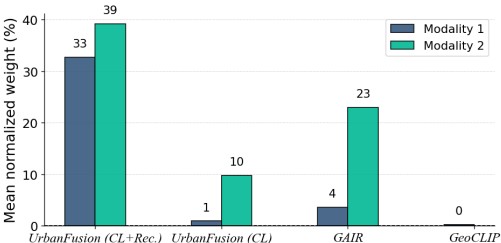

(a) Predictive performance across different information types in *Multi-Modal Spatial Encoding*.

(b) Average normalized first-layer weights for modality-specific (*unique*) feature dimensions.

Figure 3: Empirical results on synthetic data analyzing the preservation of *redundant*, *unique*, and *synergistic* information using the Partial Information Decomposition (PID) framework.

### 4.6 TRAINING WITH INCOMPLETE MULTIMODAL DATA

All modalities used for training *UrbanFusion* are open-source and globally available, except street-view imagery, which limits performance in regions without coverage (Klemmer et al., 2025). However, existing multimodal methods typically require that each location has all modalities present. Ideally, a *GeoFM* shall reuse a collection of existing datasets for pretraining, even if not geographically aligned. We therefore conduct an ablation study where each location has only coordinates and one modality (*Bimodal*). Despite using only ∼25% of the data per modality, this reduced-modality setting retains 99.35% of the performance of the model trained with complete modality pairs, matching or outperforming it in 40% of the evaluated domains. These results show that *UrbanFusion* is both flexible and data-efficient, as even incomplete modality pairs can be used for effective pretraining. Aligning a single modality with coordinates already yields strong performance. This result is consistent with Girdhar et al. (2023) and addresses a key limitation of prior work, which required fully paired modalities during training. More details are in Appendix C.1.

## 5 DISCUSSION AND CONCLUSION

We introduced *UrbanFusion*, a spatial embedding model that learns fused, multimodal representations of urban locations via *Stochastic Multimodal Fusion* (*SMF*). The model flexibly supports varying modality combinations, generalizes across diverse urban settings, enables scaling with large, heterogeneous global datasets. UrbanFusion achieves the state-of-the-art performance on a majority of tasks, with its few underperformances primarily due to the smaller dataset size for specific modalities (e.g. trained on 64x fewer SV images than GeoCLIP), while its strength lies in tasks requiring fusion to retain unique and synergistic information. Limitations include imperfect temporal alignment across modalities and our focus on point- or postal code-level data in urban environments, which may limit applicability to rural areas or data on other scales. Future directions include incorporating temporal data (e.g., satellite image sequences) for dynamic tasks such as land-use change detection or urban growth forecasting, as well as expanding to new modalities like mobility traces, social media, or location descriptions. Incorporating better encoders such as recent work on vectorized OSM embeddings Bai et al. (2025) could improve the performance. Beyond geospatial data, *SMF* offers a general-purpose framework for multimodal learning, capturing cross-modal interactions without handcrafted augmentations. We hope this work advances scalable, transferable GeoAI representations and inspires broader innovation in multimodal learning.

## REPRODUCIBILITY STATEMENT

Our results are fully reproducible with the code available at `https://anonymous.4open.science/r/SpatialFoundationModel-9551/`. We have included scripts for preprocessing, training, and evaluation to facilitate accurate reproduction of our experiments. Additionally, to enhance accessibility for a broader audience, we provide tutorial notebooks that guide users step-by-step through the process. We have also published our modified version of the Place Pulse 2.0 dataset on Hugging Face (link will be included upon publication for anonymity). This dataset includes original SVI, extracted POIs, OSM BaseMaps, and Remote Sensing data.

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

# APPENDIX

## CONTENTS

## A    ABBREVIATIONS

| | |
|---|---|
| **Coords** | Encoded coordinates |
| **SV** | Street view image |
| **RS** | Remote sensing imagery |
| **OSM** | Cartographic basemap from OpenStreetMap |
| **POI** | Point of interest |
| **MLP** | Multilayer Perceptron |
| **ViT** | Vision Transformer |
| **GPU** | Graphics Processing Unit |
| **GeoFM** | Geo-Foundation Model |

## B    DETAILED RESULTS OVERVIEW

In this section, we provide and discuss further results, including additional baselines, downstream learners, and ablations.

### B.1    COORDINATE-ONLY SPATIAL ENCODING

Table 13 presents linear probing and MLP results for a widely studied use case of location representations: generating embeddings from raw geographical coordinates. *UrbanFusion* outperforms all other methods on 5 out of 8 datasets for linear probing and consistently outperforms the most closely related baselines when all models are trained on the PP2-M dataset, as extensively discussed in Section 4.2. These findings are generally consistent with the MLP-based results, where *UrbanFusion* achieves superior performance on the majority of tasks. Notably, we observe that MLP results tend to be less stable across the board, often performing worse than simple linear probes for certain tasks and methods, especially on smaller datasets. This instability suggests that the added model capacity of MLPs can lead to overfitting when training data is limited or noisy, despite extensive hyperparameter tuning. An exception is the *Identity* baseline, which directly uses raw geographical coordinates as input. This method benefits significantly from the added capacity of MLPs, which aligns with expectations: since raw coordinates are not embedded in a higher-dimensional space, MLPs are better suited to model the nonlinear decision boundaries required to extract meaningful patterns directly from the coordinate space.

Table 14 provides detailed results for the Place Pulse 2.0 Urban Perception task, where all methods exhibit low $R^2$ values. This can be attributed to the inherent noise and bias in the dataset, which is based on human perceptions of street view imagery. The subjective nature of the annotations, influenced by factors such as weather conditions or traffic, likely reduces the spatial correlation of the target labels. Even more flexible models such as MLPs tend to perform poorly on this task, potentially due to overfitting to these noisy and weakly spatially structured labels.

Table 15 and Table 16 present detailed results for ZIP Code-level prediction tasks using linear probing and MLPs, respectively. On average across all categories, *UrbanFusion* achieves the highest performance among all evaluated methods, even outperforming Google's *PDFM* model, which was explicitly designed for this task. While *PDFM* performs strongly on health-related tasks, likely due to its use of web search trend data, it underperforms on environmental tasks, possibly due to the absence of visual inputs such as street view images, satellite imagery, or cartographic basemaps. In general, we find that MLPs perform worse than linear models on ZIP Code-level tasks, often resulting in catastrophic overfitting for some baselines and different random seeds, despite extensive hyperparameter tuning. This can be explained by the relatively small number of ZIP codes within the evaluated urban areas, which limits the amount of training data and increases the risk of overfitting when using higher-capacity models.

Figure 4 presents representations produced by *UrbanFusion* and several baseline models, reduced to three dimensions via principal component analysis (PCA) and mapped to RGB color space. A visual inspection reveals that *UrbanFusion* produces the most detailed and spatially coherent representations, likely due to its multimodal fusion strategy. *GAIR* and *GeoCLIP* follow closely, although with

slightly less granularity. In contrast, *SatCLIP* yields coarser and less structured spatial patterns. This aligns with its lower quantitative performance and can be largely attributed to its use of *Spherical Harmonics* for coordinate encoding (for a more detailed discussion, see Section C.4).

*GPS2Vec (tag)* yields smooth and detailed representations, whereas *GPS2Vec (visual)* exhibits more high-frequency variation, resulting in lower spatial smoothness. *CSP* displays limited spatial variation, suggesting lower sensitivity to local features. Finally, *PDFM* exhibits limited spatial smoothness—an expected outcome, given that its design limits spatial resolution to the ZIP code level, assigning identical representations to all locations within the same ZIP code during inference.

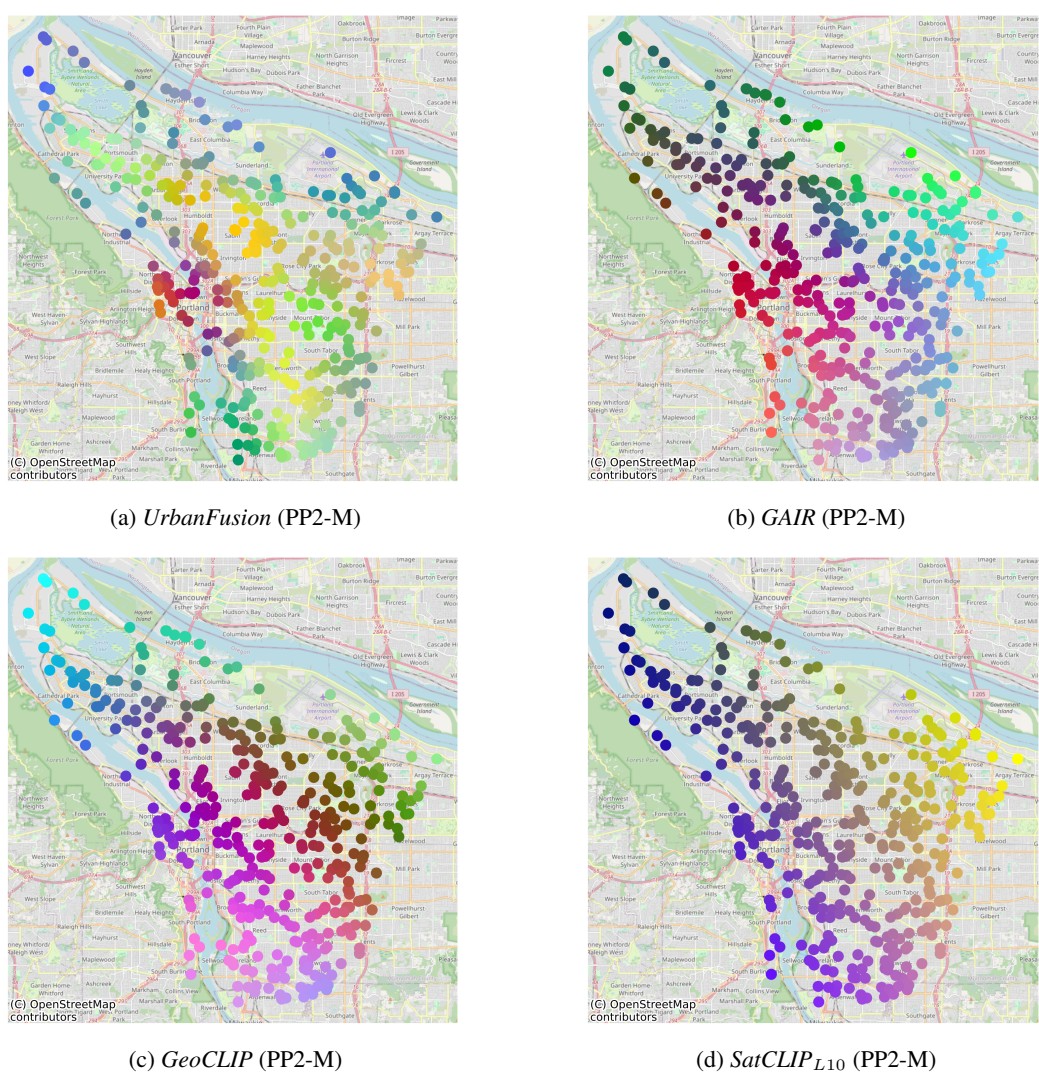

(a) *UrbanFusion* (PP2-M)

(b) *GAIR* (PP2-M)

(c) *GeoCLIP* (PP2-M)

(d) *SatCLIP$_{L10}$* (PP2-M)

Figure 4: **RGB composite image of the top three principal components of location representations computed globally for the Portland area using only coordinates. (part 1 of 2)**

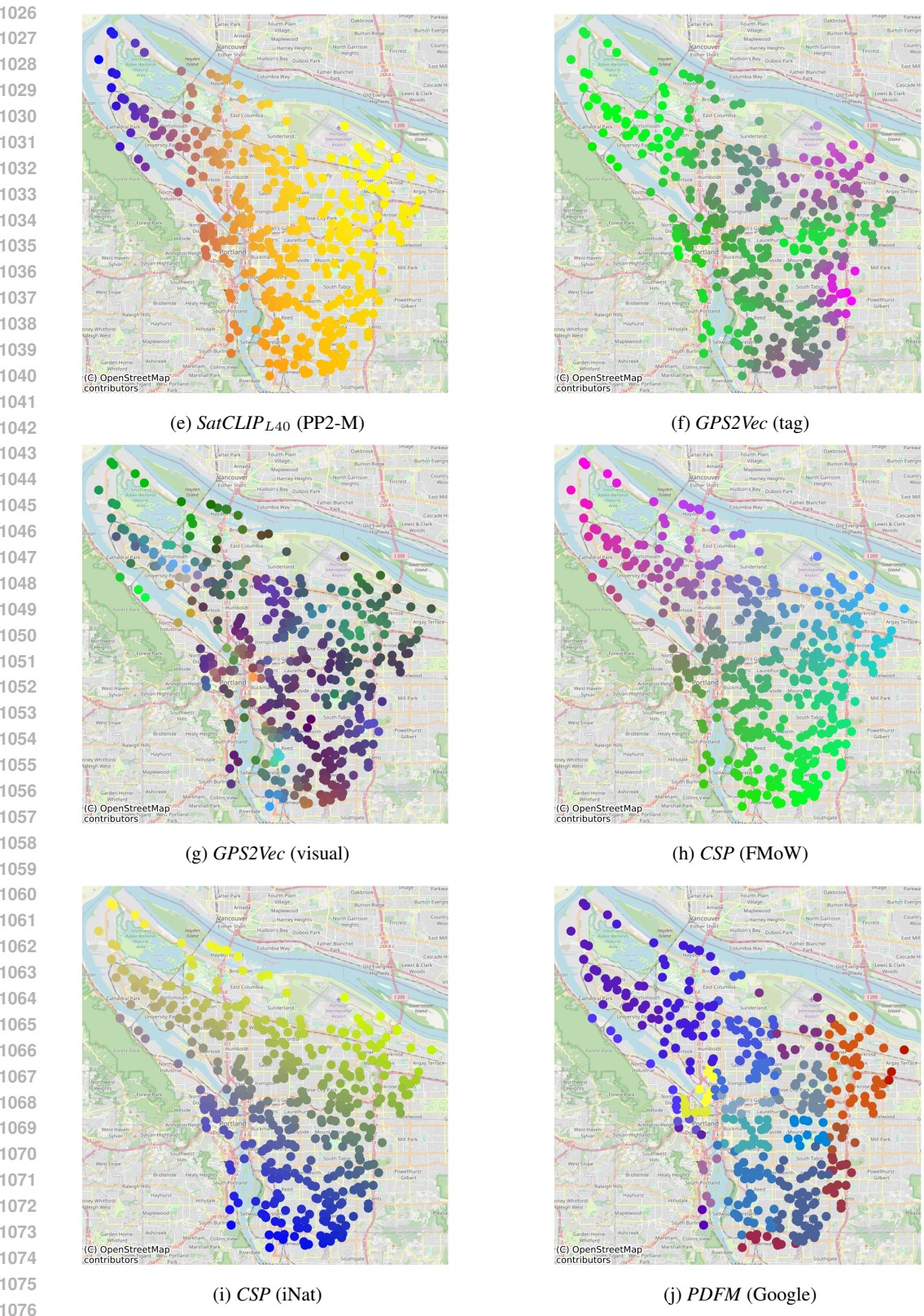

(e) $SatCLIP_{L40}$ (PP2-M)

(f) $GPS2Vec$ (tag)

(g) $GPS2Vec$ (visual)

(h) $CSP$ (FMoW)

(i) $CSP$ (iNat)

(j) $PDFM$ (Google)

Figure 4: **RGB composite image of the top three principal components of location representations computed globally for the Portland area using only coordinates. (part 2 of 2)**

## B.2 MULTIMODAL SPATIAL ENCODING

Table 17 presents results using linear probing and MLP for an additional use case of *GeoFMs*: incorporating not only coordinates but also auxiliary location information such as satellite imagery or street view imagery as input to foundation models. Since the benefit of a specific modality often depends on the downstream task, we select the subset of modalities for each model based on validation performance. While *UrbanFusion* is the only model that natively supports multimodal inputs, we also evaluate baseline models by concatenating representations obtained from individual modality encoders. For example, for *GAIR*, we concatenate representations derived separately from coordinates, satellite, and street view inputs. Results for linear probing are extensively discussed in Section 4.3 of the main paper. For MLPs, *UrbanFusion* outperforms all other methods on 3 out of 6 datasets, and on 5 out of 6 datasets when comparing only to baselines trained on the PP2-M dataset. In general, performance improves compared to *Coordinate-Only* encoding for most methods that support additional modalities, highlighting the limitations of purely coordinate-based encoders. Moreover, when analyzing the selected modalities for each downstream task, it becomes evident that modality selection is highly task-dependent. Notably, previously underexplored modalities such as cartographic basemaps and point-of-interest data also serve as valuable predictors.

Table 18 presents results for the Urban Perception task. All baselines that support encoding street view imagery perform significantly better than those relying solely on coordinate inputs. This highlights the potentially low spatial autocorrelation and biases introduced during the data collection process, as discussed in Section B.1.

Tables 19 and 20 provide detailed results for ZIP Code-level tasks using linear probing and MLPs, respectively, while Table 21 lists the selected modalities for each task. Again, it is evident that models capable of encoding additional modalities beyond coordinates consistently achieve higher performance.

## B.3 CROSS-REGIONAL GENERALIZATION

Training a global, fine-grained *GeoFM* for location representations is often infeasible due to computational and data limitations. A third important use case for such models is zero-shot generalization to entirely unseen regions. Since coordinate encodings do not generalize well to locations outside the training distribution, the model is provided only with non-coordinate modalities at inference time, using the same multimodal evaluation setup as described in Section 4.3. Table 22 presents results for linear probing on cities that were completely held out during training. To ensure that no model has been exposed to these regions, we evaluate only models trained on the PP2-M dataset while explicitly excluding the selected cities from the training set.

*UrbanFusion* outperforms all other methods, ranking first on 5 out of 6 tasks for linear probing. The closest competitor is *GAIR*, which achieves the best performance on one task. Both models clearly outperform baselines with fewer modalities such as *SatCLIP* and *GeoCLIP*, demonstrating the advantages of multimodal representation learning for geographic generalization. A similar pattern is observed for MLP results, where *UrbanFusion* achieves the highest performance on most tasks.

Table 23 provides detailed results for the Place Pulse 2.0 Urban Perception task, where *UrbanFusion* achieves the strongest performance. Tables 24, 25, and 26 present detailed results for ZIP Code-level tasks, in which *GAIR* outperforms all other methods for linear probing. Interestingly, for MLP models, raw coordinates yield the best results. This may be explained by the small number of input dimensions, which can reduce the risk of overfitting in downstream learners, despite extensive hyperparameter tuning. In contrast, some higher-capacity models exhibit extremely poor fits for specific tasks and random seeds, as shown in Table 25.

Additional insights into the superior performance of multimodal models compared to single-modal models are provided by the k-means clustering visualizations shown in Figure 5. *UrbanFusion* produces spatially smooth clusters that still preserve high-frequency variations, indicating an effective balance between global coherence and local detail. In contrast, the street view representations produced by *GeoCLIP* lack spatial smoothness, while the satellite-view representations of *SatCLIP* fail to capture high-frequency spatial changes. The multimodal *GAIR* model produces clusters that are smoother than those of *GeoCLIP*, but less smooth than those of *UrbanFusion* and *SatCLIP*.

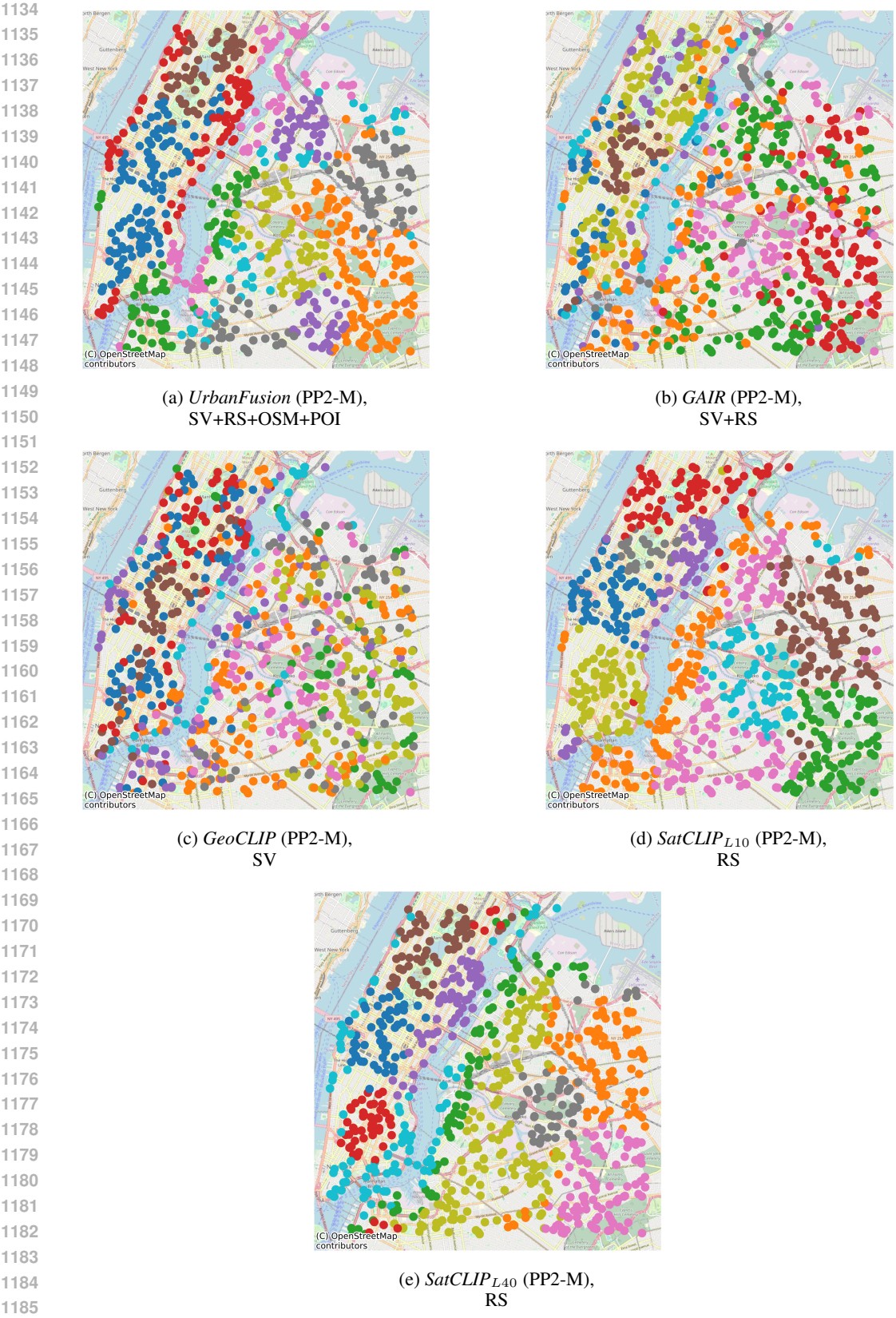

Figure 5: **KMeans clustering (k=10) results for New York City, based on *Cross-Regional Generalization***. The full names of all modality abbreviations are provided in the Appendix A.

# C    ABLATION STUDIES

This chapter presents additional ablation studies aimed at gaining deeper insights into both *Urban-Fusion* and relevant baselines. Specifically, we investigate whether *UrbanFusion* can be trained on incomplete modality sets; analyze how different methods capture modality interactions through the lens of the partial information decomposition framework; evaluate the effect of various loss function choices for *UrbanFusion*; and examine the limitations of the *Spherical Harmonics*-based location encoding used in the state-of-the-art model *SatCLIP*.

## C.1    TRAINING WITH INCOMPLETE MULTIMODAL DATA

*UrbanFusion* utilizes open-source and globally accessible data sources, including Sentinel-2 imagery and OpenStreetMap (OSM) data, both of which are freely available worldwide. In contrast, street view imagery presents a significant limitation due to its restricted spatial availability. This constraint has previously been shown to degrade the performance of *GeoFM's* in regions where such imagery is absent (Klemmer et al., 2025).

Although it is technically feasible to collect data on a global scale, training a *GeoFM* model typically requires that all modalities be available for each location. To improve data efficiency and enable better reuse of existing datasets, which may be incomplete or not geographically aligned, we explore training strategies that tolerate partially missing modalities.

To this end, we conduct an ablation study using incomplete multimodal datasets, which we refer as *Partial*. Specifically, we randomly drop modalities across the training data as follows:

- 25% of the locations contain coordinates along with all four modalities
- 25% contain coordinates and three modalities
- 25% contain coordinates and two modalities
- 25% contain coordinates and only one modality

We also examine a *Bimodal* training setup in which each location includes only coordinates and a single modality. In this case, each of the four possible coordinate-modality combinations constitutes 25% of the training set. To accelerate training, we sample batches such that all examples within a batch share the same set of available modalities.

As shown in Table 4.6, the model maintains competitive performance even under these constrained conditions. Remarkably, the *Bimodal* setup leverages only approximately 25% of the data per modality, yet still achieves strong results. This finding indicates that aligning a single modality with coordinates can be effective for pretraining, consistent with observations from (Girdhar et al., 2023). The approach broadens the scope of pretraining by enabling the use of arbitrary, modality-specific datasets, without requiring full geospatial alignment across all modalities. These results highlight the flexibility and data efficiency of our method, making it well-suited for training in regions where comprehensive modality coverage is lacking.

|  | *UrbanFusion (All)* | *UrbanFusion (Partial)* | *UrbanFusion (Bimodal)* |
|---|---|---|---|
|  |  | PP2-M |  |
| *Regression* (%$R^2$ ↑) |  |  |  |
| Housing Prices | 78.7 | **78.9** | **78.9** |
| Energy Consumption | 20.1 | **20.2** | 19.9 |
| Crime Incidence | 87.4 / **88.5** / 76.7 | 86.5 / 87.9 / 75.3 | **88.1** / 88.1 / **79.2** |
| Urban Perception (avg. 6 tasks) | 9.5 / **18.8** / **21.2** | **9.7** / 17.6 / 20.6 | 9.6 / 17.8 / 20.6 |
| ZIP Code (weighted avg. 29 tasks) | **74.3** / **75.1** / **56.7** | 71.8 / 72.3 / 55.1 | 69.4 / 72.3 / 56.4 |
| *Classification* (%F1↑) |  |  |  |
| Land Cover | 56.9 / 65.6 / **70.9** | 56.0 / **68.1** / 69.2 | 57.8 / 66.3 / 69.1 |
| Land Use – Coarse | **58.9** / 59.3 / 66.7 | 56.7 / **61.4** / **66.9** | 56.5 / 60.3 / 66.1 |
| Land Use – Fine | 47.7 / 55.2 / 61.0 | 49.7 / **55.3** / **61.9** | **51.6** / 53.7 / 60.9 |

Table 5: ***UrbanFusion* performance with varying modality input.** *All:* full modalities; *Partial:* coords + 2–5 modalities; *Bimodal:* coords + 1 modality. Results are *Coordinates-Only / Multimodal / Cross-Regional*. Best scores are in **bold**, the second-best underlined.

## C.2 EXPERIMENTS ON SYNTHETIC DATA DEMONSTRATING EMPIRICAL INFORMATION DECOMPOSITION

This section presents a complementary ablation study aimed at deepening our understanding of how multimodal contrastive alignment and *Stochastic Multimodal Fusion* (*SMF*) preserve or discard different types of information. As outlined in Chapter 2, contrastive alignment of multiple modalities using loss functions such as InfoNCE is theoretically prone to preserving *redundant* information, while *unique* (modality-specific) and *synergistic* information tends to be neglected (Dufumier et al., 2025; Dhakal et al., 2025). In *UrbanFusion*, we address this limitation by augmenting Contrastive Location Alignment with Latent Modality Reconstruction.

In real-world scenarios, we often deal with high-dimensional input data such as street view imagery or multispectral remote sensing data. In these cases, unambiguously assigning feature dimensions to either *redundant* or *unique* modality-specific information is inherently challenging. To enable systematic analysis of the preserved information, we design carefully controlled low-dimensional feature representations and auxiliary tasks, an approach inspired by similar analyses conducted in non-spatial multimodal models (see (Dufumier et al., 2025)).

### C.2.1 SYNTHETIC DATA GENERATION

Contrastive spatial representation learning models differ fundamentally from other multimodal models such as CLIP (Radford et al., 2021) or CoMM (Dufumier et al., 2025), primarily due to the use of high-capacity location encoders that directly process geographical coordinates as one of the modalities. Consequently, the notion of *redundant* information in these models includes any signal that is useful for geolocalization.

In our experiments, we focus on aligning geographical coordinates with two synthetic modalities. To simulate geolocatable signals, we assign to each modality and each location two feature dimensions within the range $[0, 1]$, which uniquely identify the location. These dimensions constitute the *redundant* information shared across both modalities and the coordinate representation, as illustrated in Figure 6. Designing feature dimensions that capture solely *unique* (modality-specific) information is more challenging. Even randomly sampled noise per location can unintentionally assist geolocalization, similar to how postal codes can implicitly encode spatial structure. To address this, we introduce a third feature dimension to each modality. During training, this dimension is batch-augmented: a single random value is sampled from the range $[0, 1]$ and assigned to all locations within the batch. During inference, however, this dimension contains independently sampled values per location. This strategy ensures that the third feature dimension has zero mutual information with the geolocation task, thereby functioning as a truly *unique*, non-redundant modality-specific signal.

We sample 40'000 equally spaced locations covering a rectangular area defined by latitudes 36.9 and 37.1, and longitudes -122.1 and -121.9. From this dataset, we use all locations in the third quad-

rant (36.9 - 37.0 latitude and from -122.1 to -122.0 longitude) for *Cross-Regional Generalization* experiments. This helps us evaluate the extrapolation capabilities of our models to unseen regions. The remaining locations are randomly sampled, with 80% used for pretraining the spatial models and 20% reserved for validation, as shown in Figure 6.

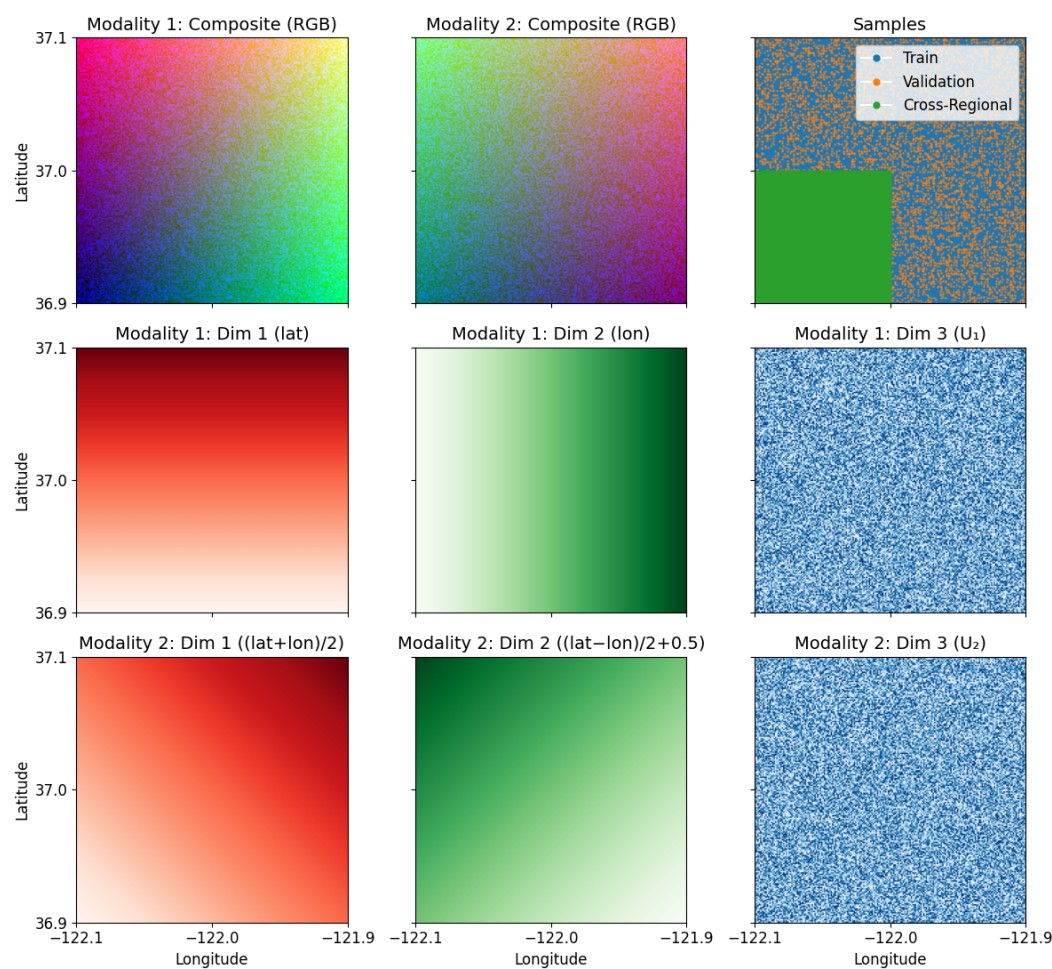

Figure 6: **Synthetic data generation.** Each location includes two modalities, each represented by a three-dimensional vector. The first two dimensions contain localization-relevant information. The third dimension consists of random values during inference, and during training, is batch-augmented to remain constant across samples. This enforces zero mutual information with location, effectively decoupling *unique* information from *redundant* information.

### C.2.2 ARCHITECTURE AND PRETRAINING

For all compared methods, we use the same location encoder: *Random Fourier Features* (Tancik et al., 2020) configured identically to those used in our other experiments (Vivanco et al., 2023), followed by a multilayer perceptron (MLP) with two hidden layers of 128 dimensions each. The modality encoders take the three-dimensional input vectors and process them through a single hidden layer with a dimensionality of four. The resulting final representation has a dimensionality of 9, corresponding to three dimensions per modality. This setup is designed to strike a balance. On one hand, it avoids the need for aggressive compression, since tasks can be solved by simply copying the input. On the other hand, it prevents excessive overparameterization, which would make the experiments less representative. In practice, input modalities often have much higher dimensionality than the learned spatial representations.

We train all models using stochastic gradient descent (SGD) with a learning rate of 0.0003, momentum, and a cosine decay schedule (Robbins & Monro, 1951). This choice is motivated by the well-understood convergence behavior of this optimizer. We apply no weight decay in order to isolate the optimization dynamics induced purely by the loss functions. All models are trained for 250 epochs, ensuring approximate convergence.

We compare *UrbanFusion* with and without Latent Modality Reconstruction against *GAIR*, as well as against *GeoCLIP*, which uses only modality 1 as input.

### C.2.3 TASKS

To evaluate the types of information captured by the learned location representations, we formulate a set of targeted tasks designed to probe for *redundant*, *unique*, and *synergistic* information. The experimental setup is as follows:

**Redundant** information in our setup corresponds to geolocation (i.e., the shared signal between the modalities and the coordinates). To quantify this, we regress the learned location embeddings onto the original geographical coordinates.

To measure **unique** (modality-specific) information, we focus on the third feature dimension of each modality, which contains random values that are uninformative for geolocation. The task is to reconstruct this feature dimension from the shared location representation. Since this dimension is intentionally made independent of the geolocation, successful reconstruction implies preservation of modality-unique information.

**Synergistic** information refers to signals that emerge only when multiple modalities are combined. To assess this, we define a task in which the target is the sum of the *unique* feature dimensions (i.e., the third dimension) from both modalities. A regression model is trained to predict this sum from the location embeddings. This task cannot be solved using information from a single modality alone, requiring cross-modal *synergy* in the learned representation.

### C.2.4 EVALUATION PROTOCOL

We evaluate all tasks on out-of-sample locations. For the *Multimodal Spatial Encoding*, we compute a fused *UrbanFusion* representation; for comparison methods, we concatenate the outputs of the individual modality-specific encoders. For each task, we train a ridge regression model using five-fold cross-validation. The regularization parameter $\alpha$ is selected from 100 logarithmically spaced values in the range $[10^{-4}, 10^4]$. Parameter tuning is performed using closed-form leave-one-out cross-validation on the training folds. For the *uniqueness* task, we report the average $R^2$ score across both modalities' reconstruction regressors.

## C.3 Effectiveness of Combining Contrastive Learning with Reconstruction Loss

As discussed in Section 3.1, there is theoretical motivation for combining contrastive learning with a reconstruction loss. While contrastive learning has proven highly effective for spatial representation learning, it primarily emphasizes features relevant for geolocalization, often overlooking modality-specific information that, although less critical for localization, may be valuable for downstream tasks.

Incorporating a reconstruction loss addresses this limitation by encouraging the model to capture richer, modality-specific details in addition to localization cues. To evaluate this empirically, we trained models using contrastive loss only *(CL)*, reconstruction loss only *(Rec.)*, and a combination of both (CL+Rec.). The results, presented in Table 6, show that assigning a higher weight to the reconstruction objective improves performance in *Multimodal* and *Cross-Regional* Generalization settings, whereas contrastive learning alone is more effective for *Coordinate-Only* encoding. Although certain tasks benefit more from one objective than the other, the combined approach consistently produces the strongest location encoder overall.

| | UrbanFusion (CL+rec.) | UrbanFusion (CL) | UrbanFusion (Rec.) |
| --- | --- | --- | --- |
| | | PP2-M | |
| *Regression* (%$R^2$ ↑) | | | |
| Housing Prices | **78.7** | 78.6 | 78.5 |
| Energy Consumption | **20.1** | 20.0 | 19.7 |
| Crime Incidence | **87.4** / 88.5 / 76.7 | 87.3 / **89.6** / **77.7** | 87.0 / 87.0 / 77.4 |
| Urban Perception (avg. 6 tasks) | 9.5 / 18.8 / 21.2 | 9.1 / 18.7 / 21.3 | **9.5** / **19.0** / **21.8** |
| ZIP Code (weighted avg. 29 tasks) | **74.3** / 75.1 / 56.7 | 71.4 / **75.5** / 57.2 | 73.0 / 72.6 / **59.8** |
| | | | |
| *Classification* (%F1↑) | | | |
| Land Cover | 56.9 / 65.6 / 70.9 | **57.4** / 66.6 / 70.4 | 56.3 / **67.7** / **71.6** |
| Land Use – Coarse | **58.9** / 59.3 / **66.7** | 58.4 / 61.8 / 65.3 | 58.0 / **61.9** /64.3 |
| Land Use – Fine | 47.7 / 55.2 / 61.0 | **50.5** / 55.0 / **61.3** | 49.0/ **56.0** / 60.4 |

Table 6: **UrbanFusion performance with varying loss functions.** Results are *Coordinates-Only* / *Multimodal* / *Cross-Regional*. Best scores are in **bold**, the second best are underlined.

## C.4 LIMITATIONS OF SPHERICAL HARMONICS FOR MODELING URBAN AREAS WITH HIGH-FREQUENCY VARIATIONS

*Spherical Harmonics* are a widely used approach for global location representation learning due to their ability to model arbitrary functions on the sphere without introducing artifacts (Rußwurm et al., 2024). This method has demonstrated high effectiveness for spatial location encoding (Klemmer et al., 2025), particularly also in ecological applications (Dollinger et al., 2025). However, in our evaluation, consistent with the results in Agarwal et al. (2024), the performance of *Spherical Harmonics* is limited in urban tasks. This is because they model smooth functions on the sphere and therefore struggle to capture high-frequency variations (Tancik et al., 2020; Ji et al., 2024), which are essential for distinguishing fine-grained spatial patterns in urban environments, as shown in Figure 4.

The expressive power of *Spherical Harmonics* can be increased by raising the order of the Legendre polynomials, allowing for the modeling of higher-frequency variations on the Earth's surface. However, the number of basis functions grows on the order of $\mathcal{O}(n^2)$ with the polynomial order, resulting in very high-dimensional feature vectors. For example, the authors of *SatCLIP* recommend setting the $L$ hyperparameter to 40 for local tasks such as housing price prediction. In their implementation, this corresponds to using Legendre polynomials up to degree 39 ($l = 0 \ldots 39$), as degrees are included up to $L - 1$.

Given the limited performance observed, we conducted an ablation study in which we increased the $L$ hyperparameter to 100 for even more local expressiveness, the highest value supported for analytical calculation in the codebase of Klemmer et al. (2025). The results of this study are shown in Table 7. Even with the increased order resulting in 10'000 basis functions, the capacity to capture high-frequency variations remains limited, while computing these functions incurs substantial additional costs and leads to unstable training.

| | $SatCLIP_{L10}$ | $SatCLIP_{L40}$ PP2-M | $SatCLIP_{L100}$ |
|---|---|---|---|
| *Regression* (%R$^2$ ↑) | | | |
| Housing Prices | 72.6 | **72.7** | 66.2 |
| Energy Consumption | 2.5 | **2.6** | 1.1 |
| Crime Incidence | 59.4 / 67.2 / 63.4 | **65.9** / **69.1** / 63.6 | 14.2 / 56.9 / **65.6** |
| Urban Perception (avg. 6 tasks*) | 5.3 / **9.5** / 12.9 | **6.1** / **9.5** / 13.2 | 0.4 / 8.8 / **13.4** |
| ZIP Code (weighted avg. 29 tasks*) | 49.2 / 69.2 / **60.8** | 54.4 / **69.7** / 59.8 | 2.3 / 65.0 / 59.3 |
| | | | |
| *Classification* (%F1↑) | | | |
| Land Cover | 45.6 / 55.9 / 61.3 | 46.4 / 56.1 / 61.1 | 31.0 / **56.6** / **63.7** |
| Land Use – Coarse | 54.3 / **57.5** / 66.7 | **57.4** / 57.2 / **66.9** | 54.0 / 57.4 / 66.1 |
| Land Use – Fine | 45.7 / 48.9 / 53.5 | **48.0** / **49.2** / 53.7 | 44.1 / 47.7 / **54.8** |
| # Basis functions | 100 | 1'600 | 10'000 |
| Time forward pass | $\sim 0.015$s | $\sim 0.530$s | $\sim 7.350$s |

Table 7: **SatCLIP performance with varying hyperparameter** $L$. Results are *Coordinates-Only / Multimodal / Cross-Regional*. Best scores are in **bold**, the second best are underlined. Time forward pass is measured as a single forward pass through the module using analytic spherical harmonic expressions, implemented as in Klemmer et al. (2025), with a batch size of 256 on an NVIDIA RTX 3090 GPU.

## C.5 Influence of Coordinates on Downstream Task Performance

For most evaluations in our work, we concatenated the geographical coordinates with the spatial representations, as this information is always available to a downstream learner and is consistent with prior research (Klemmer et al., 2025). To more thoroughly analyze the influence of raw geographical coordinates, we additionally evaluated *UrbanFusion* representations *without* concatenating coordinates.

Table 8 reports the performance of *UrbanFusion* with concatenated coordinates, without coordinates, and an *Identity* baseline that feeds only the raw coordinates to the downstream learner. As shown, concatenating raw coordinates does not yield a notable improvement on downstream tasks—and in some cases even leads to slightly lower performance—while relying solely on coordinates results in clearly inferior performance.

| | UrbanFusion (with coords) | UrbanFusion (no coords) | Identity (only coords) |
| --- | --- | --- | --- |
| | | PP2-M | |
| *Regression* (%$R^2$ ↑) | | | |
| Housing Prices | **78.7** | **78.7** | 66.2 |
| Energy Consumption | **20.1** | **20.1** | 1.5 |
| I Crime Incidence | **87.4 / 88.5 / 76.7** | 87.3 / 87.7 / 76.4 | 22.1 / 22.1 / 10.4 |
| Urban Perception (avg. 6 tasks*) | **9.5 / 18.8 / 21.2** | 9.4 / 18.6 / **21.2** | 1.3 / 1.3 / 6.6 |
| ZIP Code (weighted avg. 29 tasks*) | **74.3** / 75.1 / 56.7 | 74.0 / **75.2 / 59.3** | 3.0 / 3.0 / 17.7 |
| | | | |
| *Classification* (%F1↑) | | | |
| Land Cover | 56.9 / **65.6 / 70.9** | 57.1 / 64.4 / 70.6 | 34.4 / 34.4 / 53.9 |
| Land Use – Coarse | 58.9 / 59.3 / **66.7** | **59.2 / 59.5** / 66.4 | 48.2 / 48.2 / 55.1 |
| Land Use – Fine | 47.7 / **55.2** / 61.0 | **50.6** / 54.7 / **66.4** | 42.7 / 42.7 / 49.5 |

Table 8: **Impact of incorporating raw coordinates on downstream task performance.** Results are *Coordinates-Only / Multimodal / Cross-Regional*. Best scores are in **bold**, the second best are underlined.

## C.6 ARCHITECTURAL ABLATION STUDIES

We investigated the influence of different neural network design choices on downstream task performance. A crucial component of our framework is the *Multimodal Fusion Encoder*, which fuses the different input modalities into a unified multimodal representation. We evaluate multiple architectural variants of this encoder, specifically comparing bidirectional Transformer encoders (Devlin et al., 2019; Vaswani et al., 2017) and bidirectional Long Short-Term Memory networks (LSTMs) (Graves et al., 2005; Hochreiter & Schmidhuber, 1997) across varying encoder depths. Additionally, we examine different pooling mechanisms, contrasting the use of a dedicated [CLS] token for information aggregation (Devlin et al., 2019) with average pooling over token representations.

Overall, the results shown in Table 9 are consistent across architectural configurations, suggesting that the model's performance primarily stems from the learning framework that is robust to architectural variations. Transformers outperform LSTMs, with single-layer Transformer encoders achieving the strongest overall performance. Both pooling approaches perform well, though average pooling shows a slight advantage on most tasks.

| Fusion Encoder | Transformer | Transformer | Transformer | Transformer | LSTM | LSTM | LSTM |
|---|---|---|---|---|---|---|---|
| Pooling Mechanism | Average | Average | Average | CLS | End-state | End-state | End-state |
| Depth Fusion Encoder | 1 | 2 | 3 | 1 | 1 | 2 | 3 |
| *Regression* (%$R^2$ ↑) | | | | | | | |
| Housing Prices | 78.7 | 78.7 | **78.8** | **78.8** | 78.6 | 78.6 | 78.7 |
| Energy Consumption | 20.1 | **20.4** | 20.2 | 19.3 | 19.9 | 19.1 | 19.8 |
| Crime Incidence | 87.4 | 87.2 | **87.7** | **87.7** | 86.3 | 85.1 | 84.8 |
| Urban Perception (avg. 6 tasks*) | **9.5** | 9.3 | 8.9 | 9.3 | 9.0 | 9.1 | 9.2 |
| ZIP Code (weighted avg. 29 tasks*) | **74.3** | 71.1 | 69.7 | 69.6 | 70.8 | 69.4 | 71.0 |
| *Classification* (%F1↑) | | | | | | | |
| Land Cover | **56.9** | 55.9 | 54.4 | 56.0 | 55.3 | 55.6 | 56.8 |
| Land Use – Coarse | **58.9** | 56.7 | 58.3 | 58.5 | 57.4 | 56.8 | 57.6 |
| Land Use – Fine | 47.7 | 50.3 | 50.1 | **50.5** | 49.3 | 50.2 | 48.4 |

Table 9: **Ablation study of architectural design choices for *Coordinates-Only Encoding*.** Best scores are in **bold**, the second best are underlined.

## C.7 EVALUATING MODALITY-SPECIFIC FEATURE CONCATENATION

We compare *UrbanFusion* on the task of multimodal spatial encoding against linear regression models trained on concatenated features from the modality-specific encoders. Table 10 shows that *UrbanFusion* outperforms this baseline on most tasks, which can be attributed to its ability to model synergies between modalities through *Stochastic Multimodal Fusion*, while also reducing the dimensionality of the representations. The concatenated features perform well on the land-cover and land-use tasks, potentially because these tasks rely primarily on information unique to each modality.

| | *UrbanFusion* | *Encoded Modalities* PP2-M |
|---|---|---|
| *Regression* (%$R^2$ ↑) | | |
| Crime Incidence | **88.5** | 62.6 |
| Urban Perception (avg. 6 tasks*) | **18.8** | 17.1 |
| ZIP Code (weighted avg. 29 tasks*) | **75.1** | 70.8 |
| *Classification* (%F1↑) | | |
| Land Cover | 65.6 | **67.3** |
| Land Use – Coarse | 59.3 | **59.7** |
| Land Use – Fine | **55.2** | 53.1 |

Table 10: **Evaluation of multimodal spatial encoding.** Best results are shown in **bold**.

# D LEARNING REDUNDANT, UNIQUE, AND SYNERGISTIC INFORMATION

## D.1 ON THE LIMITATIONS OF MULTIMODAL CONTRASTIVE ALIGNMENT

While models based on contrastive learning demonstrated strong performance by aligning paired samples across different modalities, the information preserved in their embeddings is inherently limited. Contrastive losses, such as InfoNCE (van den Oord et al., 2018), primarily capture *redundant* information between modalities while neglecting *unique* modality-specific content and failing to capture *synergistic* interactions between them (Dufumier et al., 2025). This decomposition of information is formalized by the partial information decomposition (PID) framework (Williams & Beer, 2010; Dufumier et al., 2025), which provides a principled way to disentangle the different types of information shared between input modalities and a target variable.

More concretely, let $Y$ represent a downstream task variable (e.g., land use), and let $m_1$ and $m_2$ denote two input modalities (e.g., satellite and street view imagery). The mutual information $I$ between $m_1, m_2$, and $Y$ can be decomposed as:

$$I(m_1, m_2; Y) = R + U_{m_1} + U_{m_2} + S, \tag{6}$$

where:

- $R$ is the **redundant** information, present in both $m_1$ and $m_2$.
- $U_{m_1}$ and $U_{m_2}$ are the **unique**, modality-specific contributions from $m_1$ and $m_2$ respectively.
- $S$ is the **synergistic** information that is only accessible when combining $m_1$ and $m_2$ jointly.

This decomposition ensures:

$$I(m_1; Y) = R + U_{m_1} \tag{7}$$
$$I(m_2; Y) = R + U_{m_2} \tag{8}$$

Standard contrastive objectives such as InfoNCE typically use modality-specific encoders $f_m$, with representations $z_{m_1} = f_{m_1}(m_1)$ and $z_{m_2} = f_{m_2}(m_2)$. The loss is

$$\mathcal{L}_{\text{InfoNCE}} = -\log \frac{\exp\left(\text{sim}(z_{m_1}, z_{m_2})/\tau\right)}{\sum_j \exp\left(\text{sim}(z_{m_1}, z_{m_2}^{(j)})/\tau\right)}, \tag{9}$$

where $\text{sim}(\cdot, \cdot)$ is a similarity metric (e.g., cosine similarity), and $\tau$ is a temperature hyperparameter. This objective encourages $z_{m_1}$ and $z_{m_2}$ to become maximally similar for matching pairs and dissimilar otherwise. Importantly, van den Oord et al. (2018) proved that optimizing this objective maximizes a lower bound on the mutual information between the two representations:

$$I(z_{m_1}; z_{m_2}) \geq \log(N) - \mathcal{L}_{\text{InfoNCE}}, \tag{10}$$

where $N$ is the total number of samples in a mini-batch. Since $z_{m_1}$ and $z_{m_2}$ are encodings of $m_1$ and $m_2$, respectively, this bound effectively encourages preservation of the information shared between the two modalities, by treating $I(z_{m_1}; z_{m_2})$ as a proxy for $I(m_1; m_2)$.

This connection suggests that contrastive objectives implicitly assume that the mutual information between paired modalities, $I(m_1; m_2)$, serves as a good approximation for task-relevant information $I(m_1; Y)$. Under this assumption, InfoNCE maximizes the *redundant* information shared across modalities. However, as shown by Dufumier et al. (2025), this training objective is blind to information that is *unique* to a single modality ($U_{m_1}, U_{m_2}$) or *synergistic* ($S$), that is information that only emerges from combining modalities. Such content is not aligned and is therefore suppressed in the learned representations, limiting downstream performance when tasks depend on modality-specific or joint signals (see Dufumier et al. (2025) for a rigorous treatment).

To address these limitations, prior work has explored intra-modal alignment through data augmentations, aiming to capture the *uniqueness* of individual modalities (Liang et al., 2023; Yuan et al.,

2021) or even their *synergistic* relationships (Dufumier et al., 2025). However, such augmentations are often handcrafted and not well defined across all modalities. Another approach involves retrieval-augmented generation (RAG) (Lewis et al., 2020), in which the model's representations can be used to query a database. This strategy has also been explored in the context of location representation learning (Dhakal et al., 2025).

Optimally, an embedding model should be trained to retain all relevant information, including *redundant*, *unique*, and *synergistic* components, in its embeddings while enabling flexible inference with any available subset of input modalities. We address these goals with *UrbanFusion*, which leverages the underlying *Stochastic Multimodal Fusion* (SMF) framework to capture the full spectrum of modality interactions through a unified fusion strategy, without relying on handcrafted data augmentations.

### D.2 LEARNING MULTIMODAL INFORMATION WITH SMF

In the following, we present a proof that *SMF* retains all types of interactions. Proving that the embedding $z_i$ maximizes the retained mutual information is challenging as the downstream task $Y$ is unknown. Dufumier et al. (2025) circumvented this difficulty by posing a strong assumption, claiming that the mutual information between an input modality $m$ and its *augmentation* $m'$, $I(m; m')$, is the same as $I(m; Y)$. Here, we propose a significantly weaker assumption on $Y$: We assume that the mutual information between the inputs and $Y$ is lower equal than the information between the inputs and another modality or a set of modalities:

**Assumption 1:** For each downstream task $Y$, there exists at least a proxy modality or subset $S_Y \subseteq \mathcal{A}$ such that predicting $S_Y$ is at least as demanding as predicting $Y$:

$$I(\mathcal{A} \setminus S_Y; Y) \leq I(\mathcal{A} \setminus S_Y; S_Y). \tag{A}$$

**Lemma 1:** The *SMF* loss ($\mathcal{L}_{\text{total}}$) encourages $\mathcal{T}_\theta$ to retain redundant, synergistic, and unique information, thereby maximizes a lower bound on $I(m_1, \ldots, m_K; Y) = R + S + \sum_i^K U_i$.

**Proof:** For two random (masked/complement) views of modalities from the same location, $\mathcal{L}_{contr}$ maximizes a variational lower bound on the mutual information between the corresponding representations:

$$I(\mathbf{z}^{\mathcal{M}}; \mathbf{z}^{\overline{\mathcal{M}}}) \geq \log N - \mathcal{L}_{\text{contr}},$$

where $N$ is the batch size. Thus $\mathcal{L}_{contr}$ increases a computable lower bound on cross-modal shared information, i.e., *redundant information R* as shown by Oord et al. (2018); Dufumier et al. (2025). (For clarity we omit expectation notation; all bounds are understood in expectation over the data, the masking distribution, and negative sampling.)

The reconstruction head reconstructs the latent $\mathbf{h}_m$ from the fused representation $\mathbf{z}$ with $\mathcal{L}_{recon}$. Minimizing this *MSE* loss is equivalent to maximizing the average log-likelihood under a Gaussian variational decoder $q_\phi^{(m)}$ with fixed covariance Bishop & Nasrabadi (2006):

$$q_\phi^{(m)}(\mathbf{h}_m \mid \mathbf{z}) = \mathcal{N}\big(\mathbf{h}_m; g_\phi^{(m)}(\mathbf{z}), \sigma_m^2 I\big), \tag{11}$$

$$\log q_\phi^{(m)}(\mathbf{h}_m \mid \mathbf{z}) = -\tfrac{1}{2\sigma^2} \|g_\phi^{(m)}(\mathbf{z}) - \mathbf{h}_m\|^2 - \tfrac{d_m}{2} \log\big(2\pi\sigma_m^2\big) \tag{12}$$

where $d_m$ is the dimensionality of $\mathbf{h}_m$. Using

$$I(\mathbf{z}; \mathbf{h}_m) = H(\mathbf{h}_m) - H(\mathbf{h}_m \mid \mathbf{z}), \ H(\mathbf{h}_m \mid \mathbf{z}) = -\log p(\mathbf{h}_m \mid \mathbf{z}), \tag{13}$$

$$I(\mathbf{z}; \mathbf{h}_m) = H(\mathbf{h}_m) + \log p(\mathbf{h}_m \mid \mathbf{z}) \tag{14}$$

and replacing the intractable true conditional $p(\mathbf{h}_m \mid \mathbf{z})$ with the $q_\phi^{(m)}$ , we obtain the Barber–Agakov lower bound Barber & Agakov (2004)

$$I(\mathbf{z}; \mathbf{h}_m) \geq H(\mathbf{h}_m) + \log q_\phi^{(m)}(\mathbf{h}_m \mid \mathbf{z}). \tag{15}$$

Since $H(\mathbf{h}_m)$ is constant, minimizing $\mathcal{L}_{recon}$ directly maximizes a computable lower bound on $I(\mathbf{z}; \mathbf{h}_m)$. Since random subsets of modalities $M$ are masked during *SMF* training (see Section 3.1,

any set of modalities will be used at some point to create the embedding $\mathbf{z}$. When $|M| = 1$, $M$ contains a single input modality, the objective preserves that modality's *unique* information $U$; when $|M| \geq 2$, $M$ contains multiple modalities, simultaneously reconstructing all $\{\mathbf{h}_m\}$ forces $\mathbf{z}$ to integrate complementary cues, thereby promoting *synergistic $S$* information.

Finally, under Assumption 1, for each task $Y$ there exists at least one proxy subset $S_Y$ such that predicting $S_Y$ is at least as demanding as predicting $Y$. Hence minimizing the latent reconstruction loss maximizes Barber–Agakov–style computable lower bounds $\{I(\mathbf{z}; \mathbf{h}_m)\}_{m \in S_Y}$, prevents $z$ from discarding information required for $Y$. Together with the symmetric InfoNCE loss which increases the bound on $I(\mathbf{z}^{\mathcal{M}}; \mathbf{z}^{\overline{\mathcal{M}}})$, the total loss $\mathcal{L}_{\text{total}}$ maximizes two tractable mutual information surrogates and encourages retention of the PID components (*redundant*, *unique*, and *synergistic*) that matter for downstream tasks.

## E  DATA

### E.1  PRETRAINING

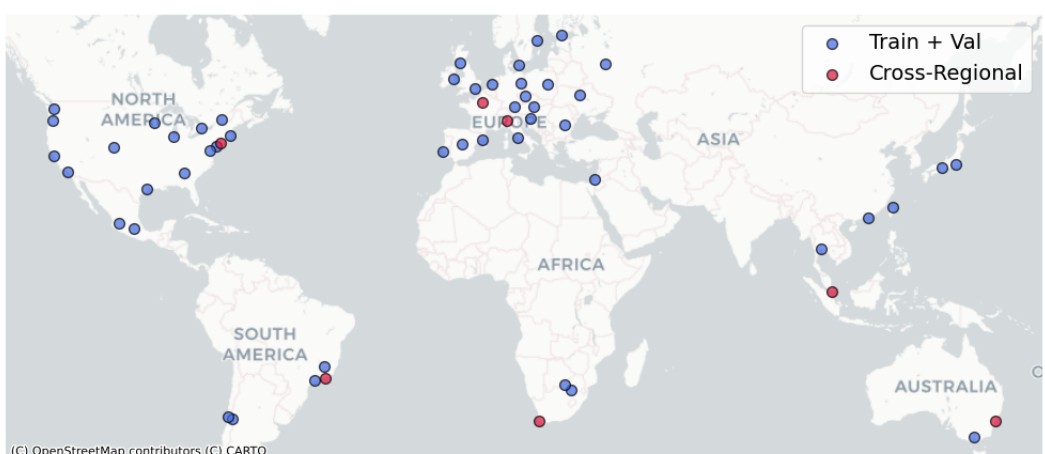

Figure 7: **Urban areas covered by the PP2-M dataset and the corresponding data splits.**

For pretraining, we leverage the Place Pulse 2.0 (PP 2.0) dataset (Dubey et al., 2016), which comprises 110'988 locations, each with associated geographic **coordinates** and **street view images**. We enrich this dataset with additional modalities, referring to the resulting extended version as PP2-M. The dataset spans 56 cities across 28 countries on all continents except Antarctica. We divide it into three parts. First, we select seven cities from six distinct continents to construct a *Cross-Regional Generalization* split, enabling evaluation of the model's ability to generalize beyond regions seen during training. From the remaining locations, we perform an 80/20 split into training and validation sets. The spatial distribution of the included urban regions is illustrated in Figure 7, while the number of samples per city is detailed in Table 12. The street view images were obtained from Google Street View (Google LLC, 2007) and have a resolution of $400 \times 300$ pixels. To ensure compatibility with the CLIP backbone and other popular models, we apply standard preprocessing following established practices from *GeoCLIP* (Vivanco et al., 2023): each image is resized to $256 \times 256$ pixels, center-cropped to $224 \times 224$, converted to a float tensor, and normalized using mean values (0.485, 0.456, 0.406) and standard deviation values (0.229, 0.224, 0.225).

We enrich the dataset with **remote sensing imagery** by downloading Sentinel-2 Level-2A images (Drusch et al., 2012) acquired between January 1, 2024 and December 31, 2024, selecting only scenes with minimal cloud coverage. Each image patch includes the spectral bands: B01, B02, B03, B04, B05, B06, B07, B08, B08A, B09, B11, and B12, and has a resolution of $256 \times 256$ pixels. Since the dataset contains 12 multispectral bands and our pretrained encoder accepts only 13 input channels (Wang et al., 2022), we append the B10 band as a zero-valued mask to match the expected input dimensionality. The raw reflectance values are normalized by dividing by 10'000, and the patches are center-cropped to $224 \times 224$ pixels to align with the input resolution of the ViT-S/16

model (Dosovitskiy et al., 2021). The entire preprocessing pipeline for remote sensing imagery follows the same procedure as used in *SatCLIP* (Klemmer et al., 2025).

**Cartographic basemaps** offer a globally available and human-interpretable source of geographic information, capturing features such as buildings, land cover, and transportation networks (Mühlematter et al., 2024a). Despite their richness, this modality has been largely overlooked in prior work. To address this, we further enrich the PP 2.0 dataset with basemaps from OpenStreetMap tile server at zoom levels 15, 16, and 17 (OpenStreetMap contributors, 2017), corresponding to spatial resolutions of 1200 m, 600 m, and 300 m, respectively. The tiles were downloaded in May 2025. Each map tile is rendered at a resolution of $256 \times 256$ pixels. For compatibility with our Masked Autoencoder (MAE) model for feature extraction from the basemaps (He et al., 2022), we resize each image to $224 \times 224$ pixels using bilinear interpolation and apply channel-wise normalization using mean values $(0.485, 0.456, 0.406)$ and standard deviation values $(0.229, 0.224, 0.225)$.

To further enrich the dataset with semantic information about the built environment, we extract **points of interest** (POIs) from OpenStreetMap for each location in the dataset (OpenStreetMap contributors, 2017). For every location, we identify the 15 nearest POIs within an adaptive radius of up to 200 meters. This adaptive search radius captures highly local context in dense urban areas while ensuring sufficient coverage in sparser regions, following principles commonly used in geostatistical analysis (Oshan et al., 2019). We retain POIs with tags under the following key-value pairs: `amenity`, `shop`, `leisure`, `tourism`, `healthcare`, `theatre`, `cinema`, `building=religious`, `building=transportation`, and `public_transport=station`. Entries tagged as `parking`, `parking_space`, `bench`, `bicycle_parking`, `motorcycle_parking`, `post_box`, and `toilets` are excluded. Each retained POI is assigned a single representative category, determined by prioritizing tags in the following order: `amenity`, `leisure`, `religion`, `public_transport`, `shop`, and `tourism`. If no relevant tag is present, the POI is labeled as `healthcare` if any tag contains the substring `healthcare`, or as `museum` if the name includes the word `museum`. Only POIs with both a defined type and a name are retained for further use. The final set of POIs for each location is used to construct a textual prompt that describes each POI's name, category, and distance. An example of such a prompt is shown in Figure 8.

---

**Example POIs Text Prompt**

Lila (type: clothes) with distance 11m,
Barber/Stylist (type: hairdresser) with distance 11m,
Martin Pulli (type: jewelry) with distance 14m,
Vape & Artisan Glass Gallery (type: tobacco) with distance 18m,
Bendi (type: jewelry) with distance 22m,
Chabaa (type: restaurant) with distance 26m,
Pizza Jawn (type: fast_food) with distance 26m,
Martelli's (type: hairdresser) with distance 28m,
Yanako (type: restaurant) with distance 31m,
JGlow Beauty (type: beauty) with distance 32m,
Dtxfy (type: beauty) with distance 33m,
Hero Complex (type: books) with distance 36m,
Jinxed (type: variety_store) with distance 36m,
Pitchers Pub (type: pub) with distance 39m,
Han Dynasty (type: restaurant) with distance 39m

---

Figure 8: **Example points of interest (POIs) text prompt provided as input to a language model for a single location.** The example corresponds to coordinates $40.025, -75.223$ in Philadelphia.

### E.2 DOWNSTREAM TASKS

This study addresses prediction tasks in urban environments. We evaluate methods for *Coordinate-Only Spatial Encoding*, where models take only raw geographic coordinates as input, and explore *Multimodal Spatial Encoding*, which enhances spatial representations with additional contextual information. Both approaches are assessed on out-of-sample locations within the same geographic

area as the training set, representing an interpolation scenario. To examine extrapolation, we evaluate *Cross-Regional Generalization*, where models are tested on cities entirely outside the spatial extent of the training data. Our experiments draw on a diverse collection of urban prediction datasets. The primary source is our PP2-M dataset, from which we select held-out locations and assign target variables for multimodal prediction tasks. Two additional datasets, covering the same region as PP2-M, are used to support large-scale experiments in the *Coordinate-Only* setting. The remainder of this section provides detailed descriptions of these datasets.

**Housing Prices (Wright, 2025).** This dataset includes valuable information on residential properties in London, including both historical and current market data. It includes property-specific attributes such as geographic coordinates, sale prices, and structural features like floor area. We restrict our analysis to properties located within the convex hull of the PP2-M training region in London, and further filter for transactions that occurred in the year 2023. The sale prices, which serve as the target variable for regression models, are log-transformed to reduce skewness in their distribution. After preprocessing, the dataset consists of 38'208 residential locations. In addition to using encoded geographic coordinates as inputs to downstream learners, we include several continuous features: `num_bathrooms`, `num_bedrooms`, `num_living_rooms`, and `log_floor_area_sqm`. We also incorporate categorical variables using one-hot encoding: `tenure_type`, `property_category`, and `energy_rating`. These features collectively support a more realistic evaluation in downstream modeling tasks.

**Energy Consumption (Department for Energy Security and Net Zero, 2024).** This dataset contains postcode-level electricity usage data for all domestic meters in the United Kingdom during the year 2023. It includes the number of electricity meters and the total energy consumption per postcode, measured in kilowatt-hours (kWh). We focus on the London region by filtering the convex hull of the training area defined in PP2-M for evaluation purposes. Next, we download the geographical centroids of all UK postcodes (Free Map Tools, 2024). For each centroid, we compute the mean energy consumption by aggregating the total energy consumption and the total number of meters, then dividing the total energy by the number of meters. This mean value serves as the target variable for regression models. The final dataset contains 60'326 distinct geographical locations.

**Crime Incidence (Ashby, 2017).** The Crime Open Database (CODE) provides detailed crime records for various United States cities for the year 2021. Each record includes the type of crime, the date of occurrence, and the geographical coordinates of the incident. We use the out-of-sample locations from the PP2-M dataset and construct a buffer of 500 meters around each location. We then count the number of crimes falling within each buffer. Due to the skewness of the resulting distribution, the counts are transformed using the natural logarithm of one plus the count, since there are also locations with zero crimes. This transformed value serves as the target variable for regression models. Additionally, we removed 17 locations to facilitate the evaluation of the *PDFM* model, which does not cover all locations. In total, there are *2'454* locations in the cities of Boston, Chicago, Houston, Los Angeles, Minneapolis, San Francisco, and Seattle for the *Coordinate-Only* and *Multimodal* settings, and *3'398* locations in New York City for *Cross-Regional Generalization*.

**Urban Perception (Dubey et al., 2016).** While the PP2-M dataset serves as the basis for pretraining our models, we use the out-of-sample locations and the six included downstream tasks to evaluate human perception of urban environments based on street view imagery across 56 cities. Specifically, 1'170'000 pairwise comparisons between street view images were collected from 81'630 online volunteers, who assessed six perceptual attributes: safe, lively, boring, wealthy, depressing, and beautiful. The Microsoft TrueSkill algorithm is applied to generate ranking scores for each image across all six attributes, allowing for quantitative comparison (Herbrich et al., 2006). These TrueSkill scores serve as the target variable for the regression models. We recognize the potential limitations of this method, as noted in previous studies (Dubey et al., 2016), including the reliance on virtual representations instead of in-person experiences. Nevertheless, we believe it is a valuable task for modeling human perception of urban environments. In total, we have 18'233 locations for *Coordinate-Only* and *Multimodal* settings, and 19'727 locations for *Cross-Regional Generalization*.

**ZIP Code Tasks (Agarwal et al., 2024).** We evaluate our method on postal code level prediction tasks in the United States, using datasets previously introduced and released by *PDFM* (Agarwal et al., 2024). Below, we describe the data acquisition process. The dataset includes the general geospatial benchmark introduced by Sun et al. (2024), accessed via Data Commons (2024) and the Earth Engine Catalog (Gorelick et al., 2017). All health-related tasks are based on the 2022 CDC

PLACES metrics, which are available at the postal code resolution through Data Commons. Socioeconomic and environmental variables are selected following Rolf et al. (2021), including income, home value, night lights, tree cover, and elevation, with all data retrieved from either Data Commons or the Earth Engine Catalog. Additionally, we include postal code level poverty data from 2022, accessed through Data Commons. We augment the PP2-M dataset by incorporating out-of-sample locations and attaching the corresponding target variables for regression tasks. In total, we construct 28 downstream tasks across major urban areas, including Philadelphia, Denver, Atlanta, Portland, Houston, Minneapolis, Chicago, Seattle, Washington, D.C., Boston, San Francisco, Los Angeles for *Coordinate-Only* and *Multimodal* settings, and New York for *Cross-Regional Generalization*.

**Land Cover (U.S. Geological Survey, Earth Resources Observation and Science Center, 2024).** The Annual National Land Cover Database provides land cover data across the continental United States at a spatial resolution of 30 meters, comprising 16 land cover classes. To support comprehensive evaluation scenarios, we augment the out-of-sample locations of the PP2-M dataset with corresponding land use labels. Since our focus is on urban areas, some classes are underrepresented. To address this, we merge all forest-related categories into a single *Forest* class and exclude the category *Emergent Herbaceous Wetlands*. Further, we removed 44 locations to evaluate the *PDFM model*, which does not cover all locations. For both *Coordinate-Only* and *Multimodal Encoding* tasks, we focus on five land cover classes: *Developed Open Space*, *Developed Low Intensity*, *Developed Medium Intensity*, *Developed High Intensity*, and *Forest*. The resulting dataset contains 4'826 labeled observations spanning the urban areas of Philadelphia, Denver, Atlanta, Portland, Houston, Minneapolis, Chicago, Seattle, Washington, D.C., Boston, San Francisco, and Los Angeles. For the *Cross-Regional Generalization* setting, which focuses exclusively on the city of New York, we exclude the *Open Water* and *Forest* classes due to insufficient sample sizes, leaving four classes: *Developed Open Space*, *Developed Low Intensity*, *Developed Medium Intensity*, and *Developed High Intensity*, with a total of 3'394 labeled locations.

**Coarse to Fine Land Use Classification (Copernicus Land Monitoring Service & European Environment Agency, 2021).** The *Urban Atlas Land Cover and Land Use 2018* dataset contains 27 categories at a spatial resolution of 10m, covering 785 Functional Urban Areas across Europe with populations exceeding 50'000. We augment the out-of-sample locations from PP2-M with corresponding land use labels. Due to a highly imbalanced distribution of classes, we evaluate this dataset through two distinct tasks. For all tasks, we exclude the category *Construction sites* due to their temporary nature and potential misalignment with modalities such as remote sensing or street view imagery, and retain only categories with more than 10 samples for each task. In the first task, we group the 27 categories (25 of which are present in the cities we analyze) into 7 supercategories, as illustrated in Table 11. This results in 6'094 labeled locations for the *Coordinate-Only* and *Multimodal* settings across 7 classes, and 4'178 locations across 6 classes in the *Cross-Regional Generalization* setup, which includes the cities of Milan and Paris. The second task focuses on fine-grained land use classification using the original Urban Atlas categories. Despite the strong class imbalance, this setup allows us to evaluate model performance in highly challenging scenarios, including few-shot cases. This yields 6'109 locations across 18 classes for the *Coordinate-Only* and *Multimodal* settings, and 4'178 locations across 13 classes for *Cross-Regional Generalization*.

| Urban Atlas Class (2018) | Supercategory |
|---|---|
| Continuous urban fabric (S.L. : ¿ 80%)
Discontinuous dense urban fabric (S.L. : 50% - 80%)
Discontinuous medium density urban fabric (S.L. : 30% - 50%)
Discontinuous low density urban fabric (S.L. : 10% - 30%)
Discontinuous very low density urban fabric (S.L. : ¡ 10%) | Urban fabric |
| Other roads and associated land
Fast transit roads and associated land
Railways and associated land
Port areas
Airports | Transportation |
| Industrial, commercial, public, military and private units
Isolated structures
Mineral extraction and dump sites | Industrial & built-up |
| Green urban areas
Sports and leisure facilities | Green & recreation |
| Arable land (annual crops)
Pastures
Permanent crops (vineyards, fruit trees, olive groves)
Complex and mixed cultivation patterns | Cropland & pasture |
| Forests
Herbaceous vegetation associations (natural grassland, moors...)
Wetlands | Natural vegetation |
| Water
Land without current use | Water & unused land |
| Construction sites | *Excluded from task* |

Table 11: **Mapping of Urban Atlas land use classes to supercategories used in coarse-grained classification.**

| City | Training | Validation | Cross-Regional Generalization | Total Samples |
|---|---|---|---|---|
| Atlanta | 3228 | 806 | - | 4034 |
| Berlin | 3188 | 796 | - | 3984 |
| Tokyo | 3029 | 757 | - | 3786 |
| Rio De Janeiro | - | - | 3659 | 3659 |
| Santiago | 2799 | 699 | - | 3498 |
| New York | - | - | 3398 | 3398 |
| Sydney | - | - | 3359 | 3359 |
| Toronto | 2630 | 657 | - | 3287 |
| Chicago | 2574 | 643 | - | 3217 |
| Houston | 2467 | 616 | - | 3083 |
| Warsaw | 2396 | 599 | - | 2995 |
| Sao Paulo | 2380 | 594 | - | 2974 |
| Moscow | 2304 | 576 | - | 2880 |
| Philadelphia | 2226 | 556 | - | 2782 |
| Melbourne | 2179 | 544 | - | 2723 |
| London | 2146 | 536 | - | 2682 |
| Montreal | 2096 | 524 | - | 2620 |
| Singapore | - | - | 2600 | 2600 |
| Cape Town | - | - | 2513 | 2513 |
| Paris | - | - | 2478 | 2478 |
| Denver | 1920 | 480 | - | 2400 |
| Munich | 1781 | 445 | - | 2226 |
| Rome | 1742 | 435 | - | 2177 |
| Bucharest | 1732 | 433 | - | 2165 |
| Madrid | 1725 | 431 | - | 2156 |
| Mexico City | 1677 | 419 | - | 2096 |
| Belo Horizonte | 1572 | 393 | - | 1965 |
| Portland | 1544 | 385 | - | 1929 |
| Lisbon | 1497 | 374 | - | 1871 |
| Johannesburg | 1494 | 373 | - | 1867 |
| Prague | 1384 | 346 | - | 1730 |
| Milan | - | - | 1720 | 1720 |
| Bangkok | 1272 | 318 | - | 1590 |
| Dublin | 1259 | 314 | - | 1573 |
| Guadalajara | 1239 | 309 | - | 1548 |
| Seattle | 1207 | 301 | - | 1508 |
| Barcelona | 1153 | 288 | - | 1441 |
| Taipei | 1109 | 277 | - | 1386 |
| Boston | 1062 | 265 | - | 1327 |
| Los Angeles | 1036 | 258 | - | 1294 |
| Stockholm | 940 | 234 | - | 1174 |
| Zagreb | 865 | 216 | - | 1081 |
| San Francisco | 815 | 203 | - | 1018 |
| Washington DC | 762 | 190 | - | 952 |
| Glasgow | 761 | 190 | - | 951 |
| Kiev | 711 | 177 | - | 888 |
| Minneapolis | 674 | 168 | - | 842 |
| Kyoto | 577 | 144 | - | 721 |
| Gaborone | 552 | 137 | - | 689 |
| Helsinki | 550 | 137 | - | 687 |
| Tel Aviv | 512 | 128 | - | 640 |
| Bratislava | 512 | 127 | - | 639 |
| Amsterdam | 510 | 127 | - | 637 |
| Hong Kong | 496 | 123 | - | 619 |
| Copenhagen | 401 | 100 | - | 501 |
| Valparaiso | 343 | 85 | - | 428 |
| Total | 73028 | 18233 | 19727 | 110988 |

Table 12: **List of urban areas in the PP2-M dataset with corresponding sample counts for training, validation, and cross-regional generalization splits.**

# F IMPLEMENTATION DETAILS

## F.1 DETAILED IMPLEMENTATION OF URBANFUSION'S ENCODERS AND FUSION MODULES

To ensure consistency with previous research and allow fair comparisons, we use modality-specific encoders that reflect those typically used in past studies. Whenever possible, we use the same pre-trained network architectures and freeze their weights during training, following established practices in studies like *GeoCLIP* (Vivanco et al., 2023) and *SatCLIP* (Klemmer et al., 2025). While full fine-tuning, particularly with parameter-efficient techniques like Low-Rank Adaptation (LoRA), has shown promise across various modalities (Hu et al., 2022; Mühlematter et al., 2024), we will leave the exploration of this avenue for future research.

Our approach supports flexible integration of arbitrary encoders and modalities, each producing a dense representation of variable size. To enable multimodal fusion, these representations are linearly projected to a shared dimensionality. This projection can map each modality to a single or multiple tokens; in practice, we find that a single token per modality is sufficient. Each linear layer is followed by a GELU activation (Hendrycks & Gimpel, 2016), which introduces non-linearity, before a learned positional embedding is added to each token. In the following, we describe the implementation of all modality-specific encoders, the multimodal fusion module, and the associated decoders, as also illustrated in Figure 1.

**Location Encoder.** To encode geographic coordinates (latitude and longitude), we first apply the Equal Earth projection (Šavrič et al., 2019), yielding a globally consistent 2D spatial representation. We then apply *Random Fourier Features* (*RFF*) with multi-scale encoding to capture spatial patterns at different resolutions (Tancik et al., 2020). Specifically, the projected coordinates are mapped into a 256-dimensional frequency space using sinusoidal functions (sine and cosine), resulting in a 512-dimensional *RFF* output. This process is performed independently for three spatial scales, using $\sigma$ values of $\sigma \in \{2^0, 2^4, 2^8\}$. Each scale-specific *RFF* embedding is then passed through a multi-layer perceptron (MLP) with three hidden layers of 1024 units each. The outputs of the MLPs across the three scales are subsequently summed to produce the final location representation. Our approach closely follows prior work such as *GeoCLIP* and *GAIR* (Vivanco et al., 2023; Liu et al., 2025), using the same *RFF* configuration and MLP hyperparameters to ensure consistency and comparability.

**Street View Imagery Encoder.** We adopt the CLIP ViT-L/14 architecture, pretrained on large-scale vision-language datasets (Radford et al., 2021; Dosovitskiy et al., 2021), to process street view images. Its demonstrated robustness across a wide range of tasks and prior use in *GeoCLIP* (Vivanco et al., 2023) supports consistent and fair evaluation.

**Remote Sensing Imagery Encoder.** To encode satellite imagery, we utilize the ViT-S/16 model trained on Sentinel-2 data using the momentum contrast (MoCo) approach (Dosovitskiy et al., 2021; Wang et al., 2022; He et al., 2020). This setup mirrors the configuration adopted in *SatCLIP* (Klemmer et al., 2025), ensuring methodological consistency.

**Cartographic Basemap Encoder.** Despite the rich semantic information contained in cartographic basemaps and the global availability of such data through sources like OpenStreetMap (OpenStreetMap contributors, 2017), their usage in visual representation learning remains limited. A key missing component is the lack of pretrained encoders tailored for feature extraction from this modality. To extract features, we fine-tune a Masked Autoencoder (MAE) (He et al., 2022) with a ViT-B/16 backbone (Dosovitskiy et al., 2021) that was initially pretrained on ImageNet (Deng et al., 2009). Our training setup employs a batch size of 256 over 100 epochs, with early stopping based on validation loss evaluated every 5 epochs. We use the AdamW optimizer with a base learning rate of $3 \times 10^{-4}$, weight decay of 0.05, and betas set to $(0.9, 0.95)$ (Loshchilov & Hutter, 2019). A cosine learning rate decay schedule is applied with 2140 warmup steps. We adopt a masking ratio of 0.6 during training. The training data consists of the training split of our PP2-M dataset. Qualitative results of basemap reconstruction are shown in Figure 9 for the validation set, and in Figure 10 for inputs from regions not covered in the training data. The model successfully reconstructs the semantic context of the images, including features such as land cover, buildings, and streets. The blurry patches in the reconstruction come from the fact that the decoder receives the non-masked patches as input and reconstructs them as well, but since the loss is only applied to masked tokens, fine details in the non-masked patches are not preserved. During application of the model, we only

use the encoder and conduct average pooling over all image tokens. We repeat this process for all three spatial scales and combine their embeddings with a learned projection.

**Points of Interest (POI) Encoder.** Each generated prompt is embedded using the BAAI/bge-small-en-v1.5 language model (Xiao et al., 2023), following prior work in multimodal location modeling (Wang et al., 2025).

**Multimodal Fusion Encoder.** We employ a single Transformer block to integrate information across all modalities (Vaswani et al., 2017). This design enables joint processing of heterogeneous input representations in an efficient and unified manner. The Transformer uses an embedding dimension of 768 and the GELU activation function (Hendrycks & Gimpel, 2016). [MASK] tokens are represented as zero vectors. Instead of relying on a [CLS] token, we apply average pooling over all output token embeddings to construct the final fused representation, which is then used for downstream tasks.

**Contrastive Learning Head.** Instead of directly using the downstream task representation for contrastive learning, as done in prior work (Vivanco et al., 2023; Klemmer et al., 2025; Liu et al., 2025; Mai et al., 2023), we observe that incorporating an additional decoding step improves performance, particularly when combined with Latent Modality Reconstruction, as noted in previous studies (e.g., Chen et al. (2020)). We introduce a lightweight decoder head composed of a LayerNorm layer (Ba et al., 2016), a GELU activation function (Hendrycks & Gimpel, 2016), and a linear projection to 512 dimensions, following established designs in prior work (Vivanco et al., 2023; Liu et al., 2025).

**Reconstruction Head.** For the Latent Modality Reconstruction objective, we employ a lightweight decoder head composed of a LayerNorm layer (Ba et al., 2016), a GELU activation function (Hendrycks & Gimpel, 2016), and a linear projection to 3842 dimensions. This output dimensionality corresponds to the concatenated length of all modality-specific latent representations produced by the pretrained encoders.

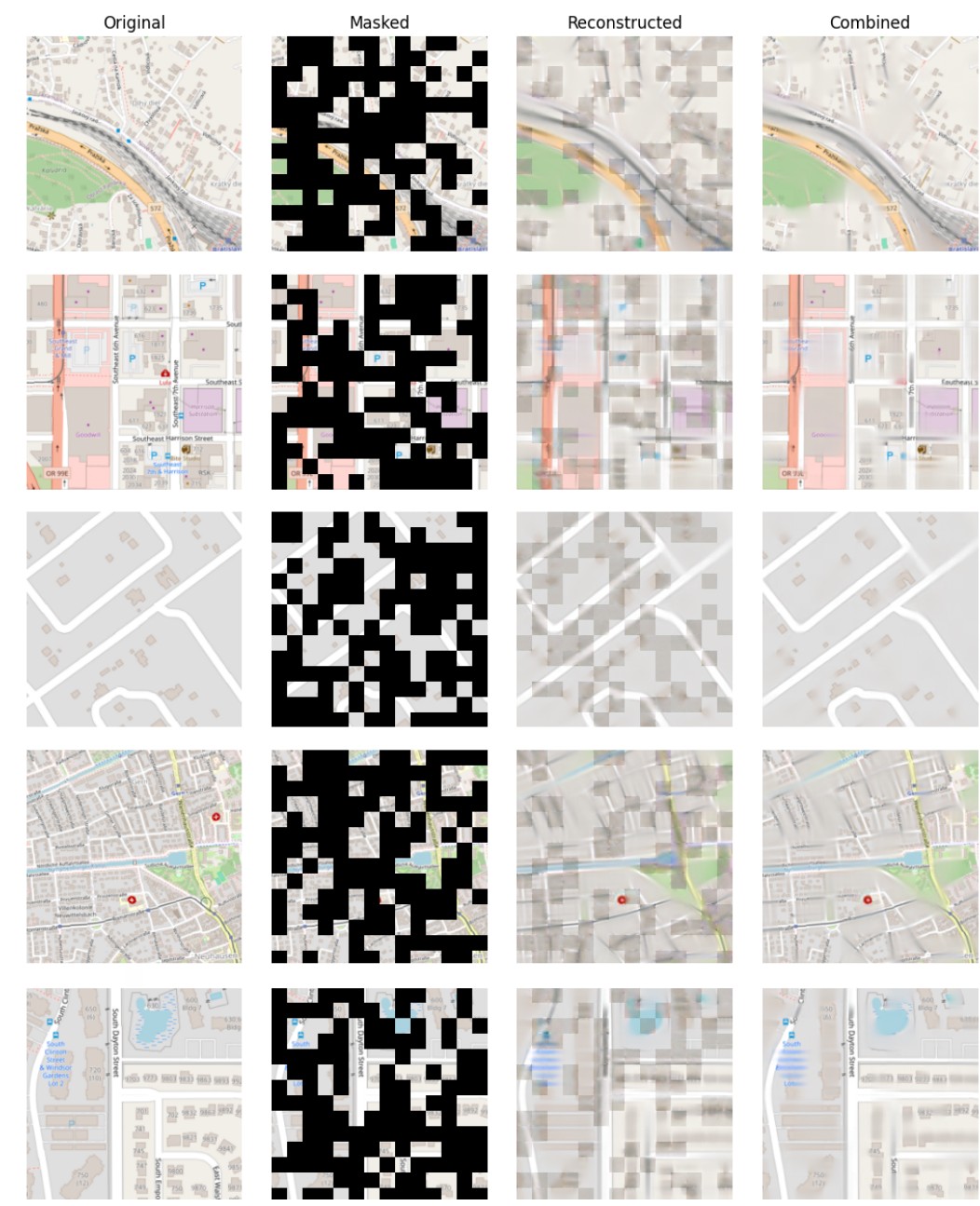

Figure 9: **Cartographic basemap reconstruction using MAE on the validation set.** The masked view is used as input to the encoder, while the reconstructed view is the output of the decoder. Since the decoder is trained to reconstruct only the masked tokens (not the visible ones), we additionally present a combined view that merges the input tokens with the reconstructed tokens for better visualization.

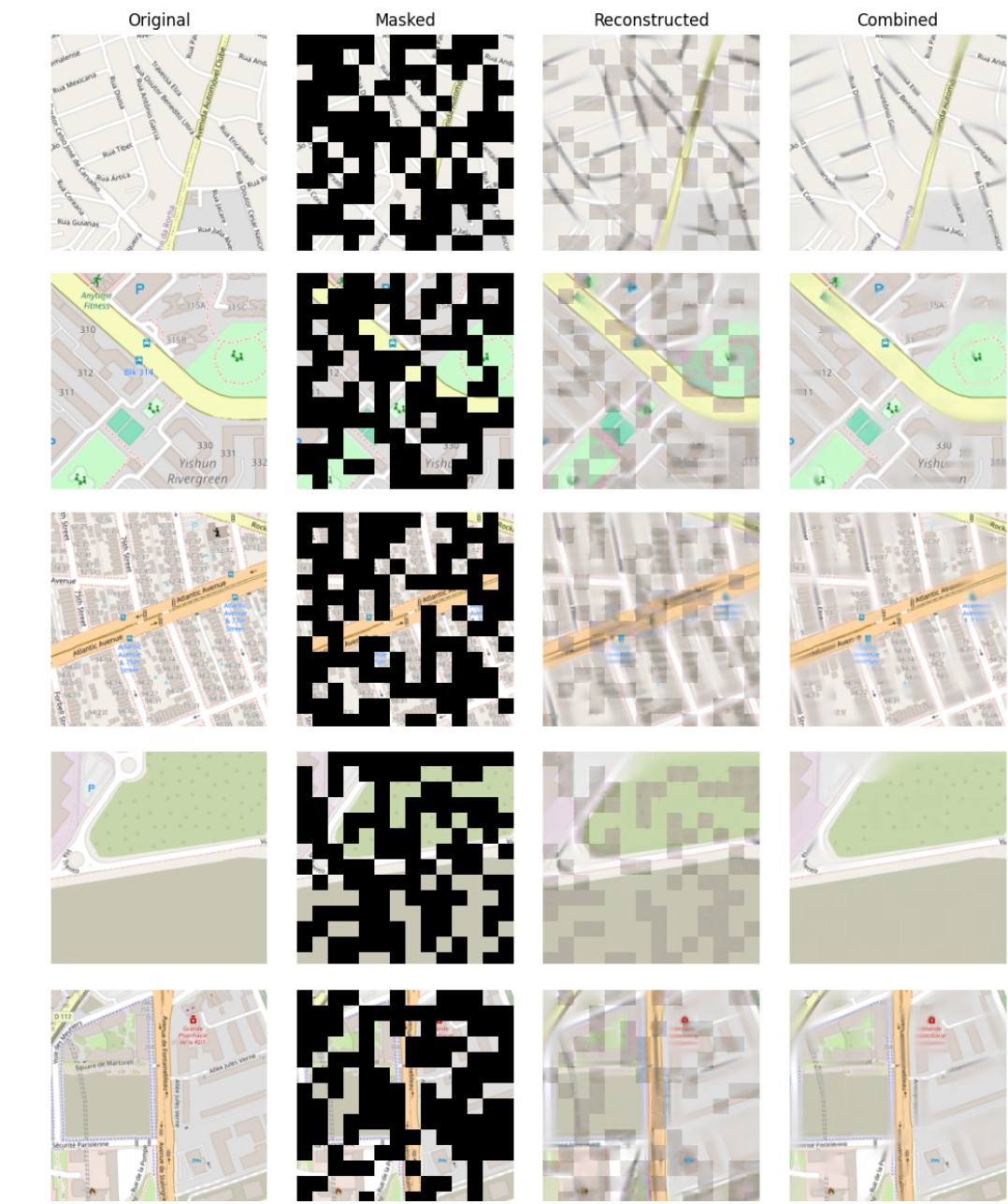

Figure 10: **Cartographic basemap reconstruction using MAE for generalization to regions unseen during training.** The masked view is input to the encoder, and the decoder outputs the reconstruction of the masked tokens. For improved visualization, we also show a combined view that merges the original input tokens with the reconstructed ones.

### F.2  TRAINING URBANFUSION

We train *UrbanFusion* for 400 epochs using the AdamW optimizer (Loshchilov & Hutter, 2019) with a base learning rate of $1 \times 10^{-4}$ and a weight decay of $1 \times 10^{-5}$. The learning rate follows a cosine decay schedule with linear warm-up during the first 5% of training steps. The batch size is set to 2'560, and early stopping is applied based on the validation loss. At each training step, random masking schemes are sampled uniformly. The validation loss is computed as the mean over all possible masking schemes and is evaluated every ten epochs. For the loss function, we assign a weight $\lambda = 0.0625$ to the Latent Modality Reconstruction term, chosen empirically to ensure that the magnitudes of both loss terms are similar during early training. The contrastive term uses a learned temperature parameter in the InfoNCE loss, initialized to 0.07. The implementation is based on `torchvision 0.21.0` (Paszke et al., 2019).

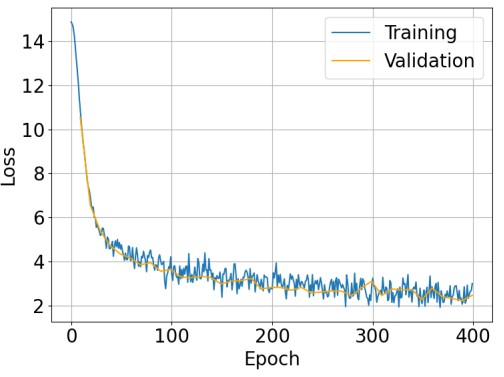
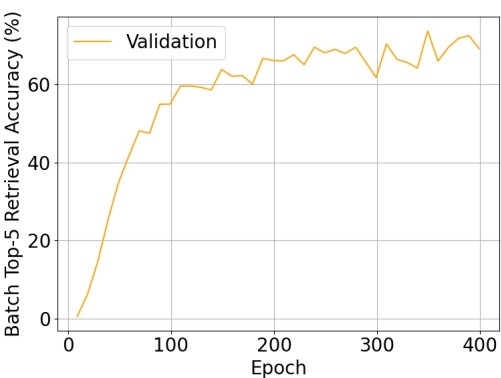

(a) Training and validation loss per epoch.          (b) Validation Top-5 accuracy per epoch.

Figure 11: **Training curves for *UrbanFusion***: (a) Training and validation loss, and (b) validation batch Top-5 retrieval accuracy. Validation Top-5 batch retrieval accuracy is computed on the similarity matrix between two masked views. Accuracy is averaged over both query→key and key→query retrieval directions within each batch, and then across all validation batches

To reduce memory usage and improve training speed, modality features are precomputed using the frozen modality-specific encoder networks. This reduces GPU memory consumption and speeds up training significantly. The final model is trained on a single NVIDIA RTX 3090 (24 GB) for approximately 8 hours. Figure 11a shows the training and validation loss curves. Training loss is averaged over an entire epoch with randomly sampled masks per batch, whereas validation loss is averaged over all possible masking combinations. The higher variability in the training loss compared to the validation loss is due to the random masking. Figure 11b reports the Top-5 retrieval accuracy within a batch, which converges to approximately 70% for a batch size of 2'560.

### F.3  BASELINES

To benchmark *UrbanFusion*, we focus on recent *Geo-Foundation Models* (*GeoFMs*) that satisfy two key criteria. First, they must support *coordinate encoding*, that is, the ability to generate representations directly from raw geographic coordinates, with optional integration of additional modalities. Second, the models should be applicable across multiple urban areas, rather than being tailored to a specific city.

Based on these criteria, we select *GeoCLIP* (Vivanco et al., 2023), *SatCLIP* (Klemmer et al., 2025), and the preprint version of *GAIR* (Liu et al., 2025) as primary baselines. These models represent recent state-of-the-art approaches and can be trained on the same dataset as *UrbanFusion*, allowing for a fair comparison of the underlying learning frameworks. Where available, we additionally evaluate the models using their original pretrained weights. Due to variations in dataset sizes and spatial coverage, direct comparisons are difficult. Nevertheless, we believe that such comparisons can provide insight into the effects of pretraining conditions and scaling.

We also include several additional models to ensure a comprehensive comparison. *CSP* (Mai et al., 2023) is the first CLIP-style framework developed specifically for location encoding. *PDFM* is

Google's Population Dynamics Foundation Model, designed for ZIP code-level tasks within the United States (Agarwal et al., 2024). Finally, we compare our approach against the set of local models from *GPS2Vec*, which serve as strong local baselines (Yin et al., 2019; 2021), as well as a simple *Identity* model that uses raw coordinates without any transformation.

Some of these models have been commonly used as baselines in prior work for location representation learning (Klemmer et al., 2025; Agarwal et al., 2024; Vivanco et al., 2023). However, to the best of our knowledge, this is the first systematic evaluation of *GeoFMs* on urban prediction tasks.

### F.3.1 *GeoCLIP*

*GeoCLIP* (Vivanco et al., 2023) proposes a CLIP-inspired framework for *image-to-GPS retrieval*, jointly embedding images and geographic coordinates into a shared latent space. The location encoder transforms GPS coordinates into high-dimensional representations using *Random Fourier Features* (*RFF*) (Tancik et al., 2020) combined with a hierarchical multi-resolution design, which mitigates spectral bias in MLPs and outperforms traditional discrete region-based classifiers (Vivanco et al., 2023). *GeoCLIP* achieves state-of-the-art performance on image geolocalization benchmarks, surpassing region-classification baselines across diverse thresholds and performs robustly even in low-data settings. Its GPS encoder also generalizes well to downstream tasks such as coordinate regression and classification (Vivanco et al., 2023; Klemmer et al., 2025; Agarwal et al., 2024).

The model is trained contrastively on 4,7 million geo-tagged images, with a frozen CLIP image encoder and a trainable two-layer projection head. The location encoder is optimized via contrastive loss, aided by a memory bank of sampled GPS embeddings (Vivanco et al., 2023).

We evaluate *UrbanFusion* against the released *GeoCLIP* weights and also retrain the model on our PP2-M dataset. Following original training settings from Vivanco et al. (2023), we use a batch size of 512, Adam optimizer (Kingma & Ba, 2015), a learning rate of 3e-5 with step decay, weight decay of 1e-6, and a memory bank with queue size 4096. Training proceeds for 400 epochs with early stopping based on validation loss.

### F.3.2 *Satellite Contrastive Location Image Pretraining (SatCLIP)*

Klemmer et al. (2025) introduces a dual-encoder, CLIP-style framework that embeds satellite imagery and GPS coordinates into a shared latent space via contrastive learning, following a similar paradigm to *GeoCLIP* Vivanco et al. (2023). The satellite image encoder (based on CNN or ViT architectures) processes multispectral Sentinel-2 tiles, while the coordinate encoder generates continuous GPS embeddings using *Spherical Harmonics* and SIREN-based MLPs (Rußwurm et al., 2024). In Klemmer et al. (2025), *SatCLIP* embeddings have been shown to outperform previous coordinate encoders such as *GeoCLIP* (Vivanco et al., 2023), *CSP* (Mai et al., 2023), and *GPS2Vec* (Yin et al., 2019; 2021) across nine geospatial downstream tasks, including temperature prediction, population density estimation, housing prices, median income, biome classification, and animal species recognition. They also demonstrate strong geographic generalization, particularly in underrepresented regions and continents (Klemmer et al., 2025).

The model is pretrained on the newly introduced S2-100K dataset, which contains 100'000 globally distributed Sentinel-2 image tiles with associated GPS coordinates. This uniform global sampling mitigates the geographic bias found in prior datasets and provides comprehensive spatial coverage Klemmer et al. (2025).

We compare *UrbanFusion* against both the original *SatCLIP* pretrained weights and versions pretrained on our PP2-M dataset. For training, we follow the original setup from Klemmer et al. (2025), using a batch size of 8192, learning rate of 1e-4, weight decay of 0.01, and the AdamW optimizer over 400 epochs (Loshchilov & Hutter, 2019). We use the backbone ViT-S/16 pretrained on Sentinel-2, which is also used in *UrbanFusion* (Dosovitskiy et al., 2021; Wang et al., 2022). We evaluate both published variants *L10* and *L40*.

### F.3.3 *GAIR*

*GAIR* (Liu et al., 2025) proposes the first multimodal CLIP-style location representation model, unifying remote sensing imagery, street view imagery, and GPS coordinates into a shared embedding

space. Although *GAIR* had not undergone peer review at the time of writing, it demonstrates state-of-the-art performance across multiple geospatial benchmarks.

Its architecture consists of three modality-specific encoders: one each for Sentinel-2 satellite tiles, street view images, and GPS coordinates. A key innovation is the Implicit Neural Representation (INR) module, which enables continuous spatial embeddings by interpolating within the satellite image representations at the precise location of the corresponding street view image. These modality-specific embeddings are then aligned through pairwise contrastive learning. *GAIR* is pretrained on a globally sampled dataset of 1 million paired Sentinel-2 and street view images (Liu et al., 2025).

The official code and pretrained weights were not publicly available at the time of writing. We reimplemented the method following the paper's description, with minor modifications. Specifically, we use the same encoders for street view and satellite imagery as in *UrbanFusion*, *GeoCLIP* (Vivanco et al., 2023), and *SatCLIP* (Klemmer et al., 2025) to facilitate consistent comparison. While this omits one of GAIR's core contribution, the INR module, we note that *UrbanFusion* could similarly support INR, making the comparison still informative.

We train the model on PP2-M using the hyperparameters reported in the original paper (Liu et al., 2025): a batch size of 256, a base learning rate of $1.5 \times 10^{-6}$ with a warm-up over the first 5% of training epochs. We use the AdamW optimizer with $\beta_1 = 0.9$, $\beta_2 = 0.999$, a weight decay of 0.01, and a memory bank with queue size 4096.

### F.3.4 *GPS2Vec*

*GPS2Vec* (Yin et al., 2019) introduces a two-level, grid-based approach for encoding global GPS coordinates. The Earth is first partitioned into UTM zones, with a lightweight neural network trained per zone. Each model learns to predict semantic tags directly from GPS coordinates using supervision from one million geo-tagged Flickr images, selecting the 2'000 most frequent tags as the target vocabulary. This method achieved state-of-the-art performance in geo-tagged image classification. The approach was later extended to *GPS2Vec+* (Yin et al., 2021), which incorporates additionally visual features extracted from RGB images. This enhanced version, trained on six million Flickr images with tags, further improved performance over the original *GPS2Vec*.

Although *GPS2Vec* is not a global *GeoFM*, but rather a collection of local models, we include it in our benchmark as a strong local baseline. Following previous convention in Klemmer et al. (2025), we refer to the original version as *GPS2Vec (tag)* and the multimodal version as *GPS2Vec (visual)*.

### F.3.5 *Contrastive Spatial Pre-Training (CSP)*

*CSP* (Mai et al., 2023) introduces the first global-scale CLIP-style location encoding framework that aligns raw GPS coordinates with ground-level (iNaturalist) (Van Horn et al., 2018) or satellite images (FMoW) (Christie et al., 2018) through contrastive learning. It pioneers the treatment of "location" as a distinct modality in a multimodal embedding space, learned by matching coordinate encodings via positional encodings and neural networks to corresponding visual features extracted by a CNN or ViT. *CSP* extends traditional CLIP objectives with spatially aware sampling strategies, such as random location negatives and SimCSE-inspired views (Gao et al., 2021). *CSP* achieves strong performance across tasks like image geolocation, geo-aware image classification, and spatial retrieval (Mai et al., 2023).

Due to the two versions of the model trained on iNaturalist and FMoW, we only include results from the coordinate encoder and do not utilize the image encoder. Previous research indicated that *CSP* performed worse than other baselines (Klemmer et al., 2025), and the modality encoders are not compatible with our set of geolocated modalities. In particular, street view imagery does not typically contain animal photographs as in iNaturalist, and the spectral channels in the FMoW dataset differ from those used by *UrbanFusion* and *SatCLIP*, making direct integration with our framework infeasible.

### F.3.6 *Population Dynamics Foundation Model (PDFM)*

*PDFM* (Agarwal et al., 2024) is a novel geospatial foundation model developed by Google Research that learns location embeddings by integrating diverse data modalities through a graph neural network (GNN). It constructs a geo-indexed dataset covering U.S. ZIP codes and counties, aggregating

information such as busyness, map features, search trends, weather, and air quality. A GNN captures spatial relationships across these modalities, yielding fixed-size embeddings for each location that can support a wide variety of downstream prediction tasks.

Its performance surpasses that of *GeoCLIP* and *SatCLIP*, achieving state-of-the-art results on ZIP code tasks. Additionally, when combined with time series forecasting models, *PDFM* enhances predictions for variables such as unemployment and poverty rates, outperforming fully supervised baselines (Agarwal et al., 2024).

Neither the code, training data, nor model weights were publicly available at the time of writing. However, we were able to obtain the model's predicted representations for ZIP codes within the United States. Because the model learns a lower-dimensional embedding for each county and ZIP code, these representations are inherently in-sample, in contrast to other baselines where evaluation locations are consistently out-of-sample.

### F.3.7  *Identity*

As a basic sanity check, instead of first constructing a spatial representation to be used as input for a downstream task, we directly use the raw geographical coordinates $\mathbf{c} = [\text{lat}, \text{lon}]$ as input to the predictive model $g$. The problem is thus formulated as $y \sim g(\mathbf{c})$, where $y$ is the target variable of interest, and $g$ denotes the predictive model (e.g., a linear model or a MLP).

## G  EVALUATION PROTOCOLS FOR DOWNSTREAM TASK PERFORMANCE

### G.1  EVALUATION METRICS

In this section, we outline the metrics used to evaluate model performance on downstream tasks. These metrics allow us to compare the effectiveness of different representations across both classification and regression settings.

#### G.1.1  $R^2$ SCORE (COEFFICIENT OF DETERMINATION)

The $R^2$ score, or coefficient of determination, is a commonly used metric to evaluate the performance of regression models. It measures the proportion of variance in the dependent variable that is predictable from the independent variables. The score is given by:

$$R^2 = 1 - \frac{\sum_{i=1}^{n}(y_i - \hat{y}_i)^2}{\sum_{i=1}^{n}(y_i - \bar{y})^2}, \tag{16}$$

where $y_i$ is the true value, $\hat{y}_i$ is the predicted value, $\bar{y}$ is the mean of the true values, and $n$ is the number of data points.

An $R^2$ score of 1 indicates perfect prediction, while a score of 0 implies that the model does no better than simply predicting the mean of the target values. Negative values indicate that the model performs worse than the mean predictor.

#### G.1.2  WEIGHTED F1-SCORE

The weighted F1-score is a metric used to evaluate classification performance by averaging the F1-scores of all classes, weighting each class by its support (the number of true instances for that class). This approach accounts for class imbalance by giving more influence to classes with more samples. It is defined as:

$$F1_{\text{weighted}} = \frac{\sum_{j=1}^{C} n_j \cdot F1_j}{\sum_{j=1}^{C} n_j}, \tag{17}$$

where $C$ is the total number of classes, $n_j$ is the number of true instances of class $j$, and $F1_j$ is the F1-score for class $j$, given by

$$F1_j = \frac{2 p_j r_j}{p_j + r_j}.$$

Here, $p_j$ and $r_j$ denote the Precision and Recall for class $j$, respectively:

$$r_j = \frac{TP_j}{TP_j + FN_j}, \quad p_j = \frac{TP_j}{TP_j + FP_j},$$

with $TP_j$, $FP_j$, and $FN_j$ representing the number of True Positives, False Positives, and False Negatives for class $j$.

### G.2  LOSS FUNCTIONS

We also describe the loss functions used during downstream model training. These losses guide the optimization of classifiers or regressors applied on top of the pretrained representations.

#### G.2.1  MEAN SQUARED ERROR (MSE)

The mean squared error (MSE) is a standard metric for training regression models. It calculates the average of the squared differences between predicted and true values, penalizing larger errors more heavily. MSE is defined as:

$$\text{MSE} = \frac{1}{n} \sum_{i=1}^{n} (y_i - \hat{y}_i)^2, \tag{18}$$

where $y_i$ is the true value, $\hat{y}_i$ is the predicted value, and $n$ is the number of data points. Lower values indicate better performance.

### G.2.2 CROSS-ENTROPY

Cross-entropy is a commonly used loss for classification tasks, measuring the dissimilarity between the predicted probability distribution and the true distribution. For a dataset of $n$ samples with one-hot encoded target vectors $y_i$ and predicted probabilities $p_i$, the mean cross-entropy is defined as:

$$\mathcal{L} = -\frac{1}{n} \sum_{i=1}^{n} \sum_{k=1}^{K} y_{i,k} \log(p_{i,k}), \tag{19}$$

where $K$ is the number of classes, $y_{i,k}$ is 1 if sample $i$ belongs to class $k$ and 0 otherwise, and $p_{i,k}$ is the predicted probability for class $k$ in sample $i$.

### G.3 DOWNSTREAM EVALUATION PROCEDURE

For evaluation on downstream tasks, we assess exclusively out-of-sample locations. Specifically, for training downstream models we use only locations and input modalities that were not used for pretraining the models, in order to analyze the generalization capability of the methods. The only exception is *PDFM*, since all published representations are compressed representations at the postal code level and correspond to in-sample data from the training phase.

For each downstream task, we split the available locations into training (60 percent), validation (20 percent), and test (20 percent) sets. For classification tasks, the splits are stratified to preserve the label distribution across sets. Hyperparameter tuning is performed using the Optuna framework (Akiba et al., 2019). For each method, we conduct 20 trials optimizing one of the following:

- The alpha regularization parameter of scikit-learn's ridge regression (Pedregosa et al., 2011), sampled logarithmically from the interval $[10^{-4}, 10^4]$.

- The C regularization parameter of scikit-learn's logistic regression (Pedregosa et al., 2011), also logarithmically within $[10^{-4}, 10^4]$.

- For MLPs implemented in PyTorch (Paszke et al., 2019), we tune the learning rate sampled logarithmically between $[10^{-5}, 10^{-1}]$ and weight decay similarly between $[10^{-6}, 10^{-1}]$.

The MLP architecture consists of two hidden layers with 512 and 256 units respectively. Models are trained for 40 epochs using the AdamW optimizer with a batch size of 64 (Loshchilov & Hutter, 2019). Early stopping is applied with a patience of 10 epochs, based on validation performance.

All representations are concatenated with raw coordinates, following common practice in prior work, and normalized to have zero mean and unit variance (Klemmer et al., 2025). Hyperparameter tuning, early stopping, and modality selection are all performed based on the validation score. We use mean squared error (Equation 18) for regression tasks and cross-entropy (Equation 19) for classification tasks. After selecting the best hyperparameters based on validation, we train five MLP models with different random seeds and report both the mean and standard deviation of their performance.

A specialized evaluation procedure is applied to ZIP code level tasks. The *PDFM* model is specifically designed for postal code scale, whereas other methods produce representations at varying or multi-scale resolutions. In the PP2-M dataset, multiple locations may exist within a single postal code. To prevent overrepresentation of specific ZIP codes and to ensure consistency, we use a grouped evaluation protocol: during evaluation, predictions for all locations within the same ZIP code are averaged before computing metrics.

## H    STATEMENT ON THE USE OF GENERATIVE AI AND DECLARATION OF ORIGINALITY

In the preparation of this paper, generative AI (ChatGPT version 4o) was utilized for language corrections, including grammar and style improvements. The use of AI was limited to improving readability; it was not used to generate original content, conduct research, or contribute to the intellectual development of the work.

## I    ADDITIONAL TABLES

This section presents additional results, including the baselines *CSP* and *SatCLIP$_{L10}$*, as well as detailed results for MLP models, and results on the Urban Perception and ZIP Code tasks.

| | UrbanFusion | GAIR | GeoCLIP PP2-M | SatCLIP$_{L10}$ | SatCLIP$_{L40}$ | GeoCLIP MP-16 | SatCLIP$_{L10}$ S2-100K | SatCLIP$_{L40}$ S2-100K | GPS2Vec tag | GPS2Vec visual | CSP FMoW | CSP iNat | PDFM Google | Identity $y \sim g(c)$ |
|---|---|---|---|---|---|---|---|---|---|---|---|---|---|---|
| **Linear** | | | | | | | | | | | | | | |
| *Regression (%R² ↑)* | | | | | | | | | | | | | | |
| Housing Prices | [78.7] | 78.5 | 78.4 | 72.6 | 72.7 | 78.6 | 72.7 | 73.1 | **79.2** | 79.0 | 70.9 | 70.6 | - | 66.6 |
| Energy Consumption | [20.1] | 18.4 | 18.5 | 2.5 | 2.6 | 18.7 | 2.5 | 3.3 | **22.3** | 20.0 | 1.6 | 1.7 | - | 1.5 |
| Crime Incidence | [87.4] | 85.4 | 84.0 | 59.4 | 65.9 | 86.3 | 58.9 | 61.5 | 84.4 | 76.5 | 50.8 | 52.5 | 74.5 | 22.1 |
| Urban Perception (6 tasks*) | [9.5] | 7.8 | 8.0 | 5.3 | 6.1 | 8.1 | 5.2 | 5.7 | - | - | 4.8 | 4.8 | - | 1.3 |
| ZIP Code (29 tasks*) | [74.3] | 64.6 | 67.1 | 49.2 | 54.4 | 66.9 | 44.0 | 52.3 | 55.6 | 48.6 | 41.5 | 42.2 | 74.0 | 3.0 |
| *Classification (%F1↑)* | | | | | | | | | | | | | | |
| Land Cover | [56.9] | 53.3 | 53.2 | 45.6 | 46.4 | 54.6 | 44.7 | 45.9 | 53.3 | 54.4 | 39.3 | 44.4 | 51.3 | 34.4 |
| Land Use – Coarse | [58.9] | 57.3 | 57.5 | 54.3 | 57.4 | 57.0 | 54.5 | 53.2 | 58.6 | 54.2 | 54.4 | 54.5 | - | 48.2 |
| Land Use – Fine | 47.7 | [49.9] | 48.3 | 45.7 | 48.0 | 48.6 | 45.7 | 45.7 | 48.3 | 46.6 | 45.7 | 45.7 | - | 42.7 |
| **MLP** | | | | | | | | | | | | | | |
| *Regression (%R² ↑)* | | | | | | | | | | | | | | |
| Housing Prices | 77.0 ± 2.9 | 75.4 ± 4.0 | 76.4 ± 4.4 | [77.2] ± 1.2 | 77.2 ± 2.0 | 74.6 ± 4.6 | 74.6 ± 1.2 | 77.0 ± 2.4 | 78.5 ± 4.2 | **79.6** ± 0.0 | 72.4 ± 3.0 | 75.3 ± 1.7 | - | 76.9 ± 2.8 |
| Energy Consumption | 19.9 ± 0.2 | −13.4 ± 64.0 | 19.1 ± 0.3 | 11.0 ± 0.2 | 11.2 ± 0.1 | 20.3 ± 0.1 | 10.8 ± 0.8 | 10.4 ± 3.6 | −145.1 ± 0.0 | **22.3** ± 0.1 | 11.4 ± 0.2 | 11.7 ± 0.6 | - | 15.5 ± 1.0 |
| Crime Incidence | 89.4 ± 0.1 | 87.5 ± 0.2 | 86.5 ± 0.1 | 28.6 ± 0.8 | 46.5 ± 3.1 | 89.2 ± 0.2 | 27.6 ± 0.5 | 46.1 ± 0.4 | 86.6 ± 0.9 | 73.3 ± 0.7 | 27.0 ± 0.3 | 27.1 ± 0.2 | 78.2 ± 1.3 | 28.1 ± 0.2 |
| Urban Perception (6 tasks*) | [5.5] | 4.7 | 4.6 | 3.7 | 4.7 | 6.2 | 3.8 | 4.2 | - | - | 4.2 | 3.8 | - | 3.9 |
| ZIP Code (29 tasks*) | [67.6] | 34.5 | 65.3 | 36.4 | −8.1 | 51.2 | 37.8 | 29.6 | 46.8 | 9.1 | 36.3 | 34.7 | 55.3 | 35.1 |
| *Classification (%F1↑)* | | | | | | | | | | | | | | |
| Land Cover | [56.8] ± 0.4 | 53.6 ± 0.4 | 54.1 ± 0.5 | 43.5 ± 2.1 | 45.8 ± 0.4 | 55.6 ± 0.4 | 44.4 ± 0.0 | 43.4 ± 2.1 | 53.6 ± 0.5 | 53.7 ± 1.5 | 42.4 ± 2.5 | 41.5 ± 2.4 | 52.3 ± 0.4 | 44.4 ± 0.0 |
| Land Use – Coarse | [58.8] ± 0.3 | 58.3 ± 0.3 | 58.1 ± 0.5 | 56.1 ± 1.4 | 55.5 ± 0.6 | 57.4 ± 0.3 | 55.6 ± 2.5 | 55.2 ± 3.8 | 57.9 ± 0.4 | 55.5 ± 0.7 | 55.2 ± 2.6 | 53.6 ± 4.2 | - | 54.8 ± 3.8 |
| Land Use – Fine | [50.4] ± 0.2 | 50.3 ± 0.1 | **50.4** ± 0.2 | 46.1 ± 2.1 | 47.1 ± 0.2 | 50.0 ± 0.3 | 45.6 ± 1.7 | 45.1 ± 0.9 | 49.4 ± 0.3 | 47.7 ± 1.1 | 45.6 ± 1.7 | 45.5 ± 1.7 | - | 45.1 ± 1.2 |

*Detailed results in Tables 14, 15, and 16.

Table 13: **Evaluation of Coordinate-Only Spatial Encoding.** Best results are shown in **bold**, the second-best are underlined, and top scores across all models trained on the same dataset (PP2-M) are indicated in [brackets]. MLP results include standard deviations computed over 5 random seeds.

| | UrbanFusion | GAIR | GeoCLIP PP2-M | SatCLIP$_{L10}$ | SatCLIP$_{L40}$ | GeoCLIP MP-16 | SatCLIP$_{L10}$ S2-100K | SatCLIP$_{L40}$ S2-100K | GPS2Vec tag visual | CSP FMoW | CSP iNat | PDFM Google | Identity $y \sim g(c)$ |
|---|---|---|---|---|---|---|---|---|---|---|---|---|---|
| **Linear** | | | | | | | | | | | | | |
| *Regression (%R$^2$ ↑)* | | | | | | | | | | | | | |
| Cleanliness | [2.8] | 2.4 | 2.0 | 0.7 | 1.0 | 2.5 | 0.7 | 0.7 | - | 0.9 | 0.8 | - | 0.0 |
| Depressiveness | [6.5] | 5.4 | 5.8 | 3.8 | 3.8 | 5.3 | 3.5 | 3.8 | - | 3.1 | 3.1 | - | 0.7 |
| Beauty | [11.1] | 8.9 | 9.8 | 6.7 | 7.0 | 9.0 | 6.6 | 6.5 | - | 6.4 | 6.0 | - | 2.3 |
| Safety | [14.3] | 12.2 | 12.3 | 9.8 | 11.2 | 13.0 | 9.5 | 10.0 | - | 8.4 | 8.6 | - | 2.2 |
| Liveliness | [10.7] | 8.3 | 8.3 | 3.4 | 5.8 | 8.7 | 3.8 | 5.5 | - | 3.0 | 3.1 | - | 0.4 |
| Wealth | [11.5] | 9.6 | 10.0 | 7.5 | 7.9 | 10.0 | 7.2 | 7.8 | - | 7.0 | 6.9 | - | 2.1 |
| Average | [9.5] | 7.8 | 8.0 | 5.3 | 6.1 | 8.1 | 5.2 | 5.7 | - | 4.8 | 4.8 | - | 1.3 |
| **MLP** | | | | | | | | | | | | | |
| *Regression (%R$^2$ ↑)* | | | | | | | | | | | | | |
| Cleanliness | $-0.0 \pm 0.0$ | $-0.0 \pm 0.0$ | $-6.6 \pm 5.3$ | $[0.6] \pm 0.1$ | $0.0 \pm 0.4$ | $-2.1 \pm 2.6$ | $-0.0 \pm 0.0$ | $\underline{0.4 \pm 0.3}$ | - | $\mathbf{0.6} \pm 0.1$ | $-0.4 \pm 1.2$ | - | $\underline{0.4 \pm 0.3}$ |
| Depressiveness | $2.2 \pm 0.3$ | $1.6 \pm 0.2$ | $[3.7] \pm 0.2$ | $2.7 \pm 0.7$ | $1.5 \pm 3.0$ | $\mathbf{4.1} \pm 0.1$ | $2.9 \pm 0.1$ | $3.4 \pm 0.1$ | - | $3.2 \pm 0.1$ | $2.3 \pm 0.8$ | - | $2.8 \pm 0.3$ |
| Beauty | $[6.1] \pm 0.1$ | $5.2 \pm 0.1$ | $4.9 \pm 0.3$ | $3.7 \pm 1.8$ | $5.7 \pm 1.1$ | $\mathbf{7.2} \pm 0.1$ | $4.8 \pm 0.9$ | $5.1 \pm 1.3$ | - | $\underline{6.2} \pm 0.1$ | $5.6 \pm 0.2$ | - | $5.5 \pm 0.4$ |
| Safety | $10.8 \pm 0.1$ | $10.0 \pm 0.2$ | $[10.9] \pm 0.2$ | $6.7 \pm 1.1$ | $9.0 \pm 1.4$ | $\mathbf{12.3} \pm 0.3$ | $7.1 \pm 0.9$ | $8.0 \pm 0.6$ | - | $\underline{7.0} \pm 0.3$ | $7.1 \pm 0.6$ | - | $6.8 \pm 0.3$ |
| Liveliness | $6.2 \pm 0.2$ | $5.4 \pm 0.2$ | $[7.0] \pm 0.1$ | $2.4 \pm 0.4$ | $4.6 \pm 0.8$ | $\mathbf{7.2} \pm 0.3$ | $2.4 \pm 0.6$ | $2.9 \pm 1.1$ | - | $2.5 \pm 0.6$ | $2.4 \pm 0.7$ | - | $2.6 \pm 0.4$ |
| Wealth | $[7.9] \pm 0.1$ | $5.9 \pm 0.2$ | $7.7 \pm 0.1$ | $6.1 \pm 0.2$ | $7.7 \pm 0.0$ | $\mathbf{8.3} \pm 0.3$ | $5.4 \pm 1.1$ | $5.6 \pm 0.9$ | - | $5.5 \pm 1.2$ | $5.5 \pm 0.5$ | - | $5.2 \pm 0.4$ |
| Average | [5.5] | 4.7 | 4.6 | 3.7 | 4.7 | **6.2** | 3.8 | 4.2 | - | 4.2 | 3.8 | - | 3.9 |

Table 14: **Evaluation of Coordinate-Only Spatial Encoding** for the **Place Pulse 2.0 Urban Perception tasks.** Best results are shown in **bold**, the second-best are underlined, and top scores across all models trained on the same dataset (PP2-M) are indicated in [brackets]. MLP results include standard deviations computed over 5 random seeds.

**Linear Regression (%$R^2$ ↑)**

| | UrbanFusion | GAIR | GeoCLIP | SatCLIP L10 | SatCLIP L40 | GeoCLIP | SatCLIP L10 | SatCLIP L40 | GPS2vec | GPS2vec | FMoW | CSP | PDFM | Identity |
|---|---|---|---|---|---|---|---|---|---|---|---|---|---|---|
| | PP2-M | PP2-M | PP2-M | PP2-M | PP2-M | MP-16 | S2-100K | S2-100K | tag | visual | FMoW | iNat | Google | $y \sim g(c)$ |
| **Health** | | | | | | | | | | | | | | |
| High Cholesterol | [51.4] | 37.7 | 46.8 | 25.7 | −1.3 | 46.2 | −1.7 | 26.6 | 25.3 | 27.0 | −0.0 | 12.2 | **55.1** | −5.1 |
| Physical Health Not Good | [80.5] | 78.1 | 79.2 | 63.4 | 49.8 | 79.4 | 42.4 | 55.3 | 61.8 | 57.4 | 40.6 | 43.9 | **88.5** | −0.5 |
| Stroke | [71.8] | 61.2 | 67.1 | 50.5 | 40.7 | 62.8 | 30.9 | 45.5 | 65.7 | 56.8 | 35.7 | 35.0 | **84.7** | 1.8 |
| Binge Drinking | 68.9 | 70.3 | [76.3] | 60.3 | 49.7 | 68.5 | 40.4 | 53.3 | 68.1 | 57.6 | 19.6 | 23.9 | **85.4** | 4.2 |
| Physical Inactivity | [82.4] | 74.2 | 81.7 | 66.0 | 45.0 | 80.2 | 44.6 | 54.2 | 56.0 | 52.5 | 43.3 | 44.2 | **92.0** | 6.3 |
| Received Annual Checkup | [85.6] | 81.5 | 84.7 | 80.0 | 78.2 | 83.7 | 77.2 | 80.0 | 83.1 | 77.1 | 75.2 | 78.2 | **93.0** | 54.6 |
| Cancer (Excl. Skin) | [48.6] | 29.9 | 40.2 | 28.1 | 12.1 | 37.0 | 10.2 | 9.9 | 4.3 | 14.5 | 14.6 | 15.3 | **62.1** | 9.3 |
| Diabetes | 72.2 | 73.1 | [78.7] | 63.4 | 42.0 | 73.0 | 38.2 | 51.6 | 68.8 | 51.4 | 38.6 | 41.8 | **86.8** | 5.8 |
| Mental Health Not Good | [**69.0**] | 52.6 | 45.0 | 38.5 | 35.2 | 53.7 | 27.3 | 38.3 | −54.6 | 30.4 | 29.5 | 31.4 | 66.8 | 0.5 |
| Coronary Heart Disease | [65.5] | 56.4 | 60.5 | 48.9 | 42.5 | 60.8 | 27.6 | 48.5 | 50.4 | 46.5 | 32.4 | 35.2 | **74.8** | −4.5 |
| High Blood Pressure | [67.2] | 54.3 | 63.2 | 46.2 | 33.9 | 58.3 | 22.6 | 40.5 | 54.8 | 58.2 | 30.1 | 28.0 | **82.7** | 8.2 |
| High Blood Pressure (Medicated) | [57.6] | 41.6 | 50.2 | 36.5 | 35.7 | 48.2 | 28.4 | 36.7 | 38.4 | 49.4 | 19.0 | 35.4 | **65.8** | 10.4 |
| Obesity | 74.8 | 74.5 | [76.2] | 67.7 | 46.8 | 76.9 | 46.3 | 58.0 | 69.2 | 55.2 | 49.7 | 47.7 | **86.2** | 7.6 |
| Sleep Less Than 7 Hours | 81.0 | 77.1 | [82.3] | 73.5 | 60.4 | 77.4 | 55.5 | 62.0 | 76.1 | 59.4 | 53.1 | 54.4 | **92.2** | 20.6 |
| Smoking | [77.3] | 74.0 | 76.7 | 59.9 | 44.3 | 72.4 | 39.2 | 53.0 | 63.5 | 47.1 | 41.1 | 42.0 | **85.9** | 3.2 |
| Asthma | [80.0] | 70.7 | 71.0 | 55.8 | 48.8 | 69.2 | 44.4 | 51.8 | 68.1 | 48.9 | 45.4 | 47.8 | **83.4** | 9.9 |
| Chronic Kidney Disease | [72.7] | 59.9 | 68.5 | 53.7 | 38.3 | 64.7 | 29.1 | 42.4 | 58.9 | 59.2 | 33.3 | 35.0 | **82.8** | −1.5 |
| Arthritis | [60.1] | 45.8 | 55.7 | 36.9 | 37.8 | 50.3 | 28.4 | 40.5 | 42.2 | 46.5 | 9.7 | 29.5 | **74.3** | 3.1 |
| Chronic Obstructive Pulmonary Disease | [72.4] | 67.5 | 66.6 | 55.2 | 46.2 | 67.9 | 31.9 | 52.5 | 63.6 | 50.4 | 38.4 | 39.9 | **80.6** | −1.0 |
| Received Cholesterol Screening | [42.8] | 17.4 | 11.2 | 12.1 | 10.6 | 25.9 | 5.9 | 9.4 | −152.0 | 15.3 | 10.6 | 4.6 | **54.1** | 1.0 |
| Received Dental Visit | [79.3] | 75.0 | 75.1 | 58.9 | 43.6 | 73.4 | 30.6 | 49.8 | 64.0 | 45.7 | 35.3 | 38.5 | **87.3** | 17.3 |
| Health Average | [69.6] | 60.6 | 64.6 | 51.5 | 40.0 | 63.3 | 33.3 | 45.7 | 41.7 | 47.9 | 33.1 | 36.4 | **79.3** | 7.2 |
| **Socioeconomic** | | | | | | | | | | | | | | |
| Median Household Income | [62.9] | 47.4 | 50.2 | 36.6 | 33.9 | 45.2 | 31.5 | 35.3 | 39.7 | 22.9 | 32.4 | 32.9 | **66.2** | 0.9 |
| Median Home Value | [**81.4**] | 74.6 | 76.8 | 70.6 | 65.2 | 75.9 | 64.7 | 66.5 | 73.2 | 17.3 | 60.5 | 61.0 | 81.1 | 0.2 |
| Night Lights | [**74.8**] | 61.5 | 68.0 | 47.2 | 42.2 | 69.9 | 14.9 | 47.9 | 70.8 | 58.1 | 14.3 | 12.8 | 74.0 | 12.0 |
| Population Density | [60.1] | 48.3 | 54.0 | 31.2 | 24.6 | 55.4 | 26.0 | 33.3 | 54.5 | 29.7 | 30.3 | 30.6 | **62.6** | 2.3 |
| Poverty Rate | [**66.6**] | 59.1 | 57.2 | 34.2 | 32.2 | 55.5 | 31.9 | 34.9 | 36.9 | 24.3 | 26.4 | 29.3 | 65.7 | 4.9 |
| Socioeconomic Average | [69.2] | 58.2 | 61.2 | 44.0 | 39.6 | 60.4 | 33.8 | 43.6 | 55.0 | 30.5 | 32.8 | 33.3 | **69.9** | 4.1 |
| **Environment** | | | | | | | | | | | | | | |
| Elevation | 99.6 | 99.5 | 99.7 | [**99.8**] | [**99.8**] | **99.8** | **99.8** | **99.8** | 99.5 | 82.4 | **99.8** | **99.8** | 97.9 | 1.8 |
| Tree Cover | [**68.8**] | 50.5 | 51.5 | 35.8 | 36.0 | 54.1 | 30.2 | 35.4 | 40.7 | 52.3 | 17.5 | 14.0 | 47.8 | −6.5 |
| Environment Average | [**84.2**] | 75.0 | 75.6 | 67.8 | 67.9 | 77.0 | 65.0 | 67.6 | 70.1 | 67.4 | 58.6 | 56.9 | 72.8 | −2.4 |
| Average over Categories | [**74.3**] | 64.6 | 67.1 | 54.4 | 49.2 | 66.9 | 44.0 | 52.3 | 55.6 | 48.6 | 41.5 | 42.2 | 74.0 | 3.0 |

Table 15: **Evaluation of Coordinate-Only Spatial Encoding for various ZIP Code tasks.** Best results are shown in **bold**, the second-best are underlined, and top scores across all models trained on the same dataset (PP2-M) are indicated in [brackets].

**Table 16: Evaluation of *Coordinate-Only Spatial Encoding* for various ZIP Code tasks.** Best results are shown in **bold**, the second-best are underlined, and top scores across all models trained on the same dataset (PP2-M) are indicated in [brackets]. MLP results include standard deviations computed over 5 random seeds.

| MLP (%R² ↑) | UrbanFusion | GAIR | GeoCLIP PP2-M | SatCLIP$_{L10}$ | SatCLIP$_{L40}$ | GeoCLIP MP-16 | SatCLIP$_{L10}$ S2-100K | SatCLIP$_{L40}$ S2-100K | GPS2Vec tag | GPS2Vec visual | CSP FMoW | CSP iNat | PDFM Google | Identity $y \sim g(c)$ |
|---|---|---|---|---|---|---|---|---|---|---|---|---|---|---|
| **Health** | | | | | | | | | | | | | | |
| High Cholesterol | [35.0] ± 4.2 | 28.4 ± 5.8 | 33.8 ± 3.4 | 1.0 ± 5.4 | 15.2 ± 8.0 | **38.8 ± 2.4** | 5.8 ± 1.5 | 6.1 ± 3.2 | 7.0 ± 3.4 | −131.2 ± 11.1 | 5.7 ± 7.2 | 5.1 ± 1.8 | 11.3 ± 7.8 | 1.8 ± 3.2 |
| Physical Health Not Good | 76.1 ± 1.2 | 78.3 ± 1.7 | [80.7] ± 1.5 | 29.9 ± 0.7 | 46.1 ± 3.8 | 77.5 ± 0.3 | 29.8 ± 0.7 | 39.1 ± 3.0 | 59.6 ± 4.4 | 50.0 ± 2.7 | 28.4 ± 3.4 | 8.3 ± 21.7 | 81.4 ± 1.0 | 29.0 ± 2.8 |
| Stroke | 70.1 ± 1.5 | 68.4 ± 0.6 | [70.7] ± 2.3 | 20.1 ± 1.5 | 30.0 ± 4.8 | 64.3 ± 0.8 | 22.1 ± 1.6 | 26.3 ± 3.1 | 54.4 ± 5.0 | 49.1 ± 4.7 | 15.6 ± 4.7 | 22.9 ± 1.5 | 81.5 ± 1.0 | 20.6 ± 1.0 |
| Binge Drinking | [77.1] ± 1.7 | 72.4 ± 1.7 | 74.0 ± 2.3 | 5.2 ± 13.1 | 33.9 ± 5.5 | 73.3 ± 1.7 | −0.8 ± 5.4 | 31.6 ± 2.1 | 65.6 ± 1.0 | −0.4 ± 5.6 | 24.5 ± 1.6 | 18.4 ± 4.9 | 67.2 ± 1.7 | 26.5 ± 7.0 |
| Physical Inactivity | 77.3 ± 0.8 | 76.5 ± 1.9 | [81.1] ± 1.2 | 32.6 ± 0.7 | 44.4 ± 1.8 | 76.0 ± 1.2 | 32.4 ± 1.9 | 35.4 ± 3.1 | 62.8 ± 7.1 | 55.5 ± 1.8 | 33.8 ± 1.7 | 27.2 ± 3.8 | **85.6 ± 0.6** | 31.3 ± 2.0 |
| Received Annual Checkup | 76.0 ± 1.2 | 77.4 ± 2.1 | [81.7] ± 1.3 | 77.1 ± 0.3 | 76.7 ± 2.1 | [81.4] ± 2.5 | 76.8 ± 0.6 | 73.1 ± 2.4 | 65.6 ± 5.8 | −140.0 ± 47.2 | 68.8 ± 0.5 | 70.8 ± 3.1 | 18.0 ± 9.6 | 69.2 ± 4.0 |
| Cancer (Excl. Skin) | [43.5] ± 0.7 | 27.8 ± 3.1 | 33.9 ± 2.4 | 15.4 ± 0.4 | 6.5 ± 2.8 | 31.0 ± 3.2 | 11.1 ± 3.5 | 9.7 ± 1.1 | 6.6 ± 1.3 | −43.2 ± 7.9 | 10.3 ± 2.8 | 10.0 ± 4.2 | 61.1 ± 1.8 | 13.7 ± 1.8 |
| Diabetes | 75.1 ± 1.4 | 75.1 ± 2.1 | [78.4] ± 0.9 | 28.8 ± 1.3 | 43.4 ± 2.9 | 74.1 ± 0.4 | 28.2 ± 1.0 | 36.5 ± 3.6 | 63.0 ± 2.4 | 57.8 ± 2.0 | 19.6 ± 1.7 | 27.6 ± 2.1 | **84.5 ± 0.5** | 25.1 ± 2.9 |
| Mental Health Not Good | [53.9] ± 0.9 | 42.6 ± 2.6 | [50.1] ± 2.7 | 15.5 ± 0.5 | 24.9 ± 7.4 | 45.5 ± 2.5 | 13.0 ± 2.9 | 20.9 ± 2.0 | −161.0 ± 3.7 | −15.2 ± 3.0 | 12.9 ± 7.6 | −6.6 ± 0.5 | 29.6 ± 8.7 | 13.8 ± 3.4 |
| Coronary Heart Disease | 59.2 ± 2.3 | 59.8 ± 1.8 | [64.5] ± 1.6 | 20.0 ± 2.6 | 37.6 ± 3.8 | 60.5 ± 1.2 | 19.1 ± 1.9 | 26.3 ± 5.3 | 44.8 ± 3.0 | 40.9 ± 2.6 | 5.9 ± 8.0 | 15.6 ± 6.9 | **72.1 ± 2.6** | 16.3 ± 2.5 |
| High Blood Pressure | 65.0 ± 2.1 | 61.2 ± 0.3 | [65.9] ± 1.9 | 19.1 ± 6.3 | 27.5 ± 1.6 | 57.9 ± 1.2 | 16.9 ± 3.5 | 19.0 ± 4.3 | 48.8 ± 2.9 | 28.1 ± 8.1 | 12.3 ± 3.1 | 14.8 ± 2.7 | **72.7 ± 2.2** | 12.3 ± 1.9 |
| High Blood Pressure (Medicated) | 28.5 ± 2.3 | −1635.5 ± 3326.4 | [41.7] ± 3.5 | 24.8 ± 5.8 | −3318.7 ± 4083.7 | **45.5 ± 2.3** | 25.9 ± 2.4 | 16.1 ± 8.1 | −7.0 ± 9.7 | −236.6 ± 29.9 | 16.6 ± 4.1 | 15.7 ± 6.1 | −26.6 ± 17.6 | 19.1 ± 1.9 |
| Obesity | 74.1 ± 2.4 | 75.0 ± 0.6 | [77.9] ± 0.5 | 42.4 ± 1.5 | 51.6 ± 1.9 | 71.4 ± 3.4 | 41.1 ± 3.1 | 45.9 ± 1.4 | 62.3 ± 3.5 | 30.9 ± 3.7 | 40.0 ± 3.2 | 34.5 ± 2.1 | 67.2 ± 2.6 | 36.9 ± 1.1 |
| Sleep Less Than 7 Hours | [76.9] ± 1.9 | 76.2 ± 1.3 | [82.2] ± 0.8 | 47.9 ± 2.7 | 52.8 ± 1.5 | 75.9 ± 0.5 | 49.2 ± 1.6 | 49.8 ± 2.6 | 64.3 ± 5.4 | 22.7 ± 4.7 | 47.6 ± 2.1 | 40.6 ± 2.8 | 67.8 ± 2.1 | 44.9 ± 3.3 |
| Smoking | 74.1 ± 1.7 | 72.7 ± 1.6 | [76.7] ± 1.2 | 25.2 ± 2.4 | 37.4 ± 4.4 | 71.9 ± 0.5 | 26.3 ± 2.1 | 30.1 ± 2.0 | 57.0 ± 3.0 | 45.6 ± 3.3 | 26.7 ± 2.9 | 26.4 ± 2.7 | **80.0 ± 0.4** | 26.9 ± 2.2 |
| Asthma | 66.7 ± 2.0 | 66.2 ± 0.9 | [72.9] ± 1.2 | 35.5 ± 2.2 | 36.7 ± 3.8 | 65.1 ± 0.8 | 34.5 ± 2.0 | 39.3 ± 2.2 | 31.9 ± 10.0 | −9.5 ± 1.9 | 35.1 ± 1.9 | 31.4 ± 4.5 | 46.3 ± 2.8 | 33.4 ± 1.5 |
| Chronic Kidney Disease | 69.5 ± 1.2 | 66.3 ± 1.6 | [70.3] ± 1.1 | 19.5 ± 1.5 | 36.6 ± 1.4 | 64.9 ± 1.3 | 18.2 ± 1.4 | 26.1 ± 2.6 | 53.6 ± 2.9 | 45.9 ± 4.5 | 10.2 ± 7.7 | 16.4 ± 5.0 | **77.7 ± 2.2** | 20.3 ± 2.5 |
| Arthritis | [54.6] ± 1.6 | 45.9 ± 2.3 | 52.6 ± 1.9 | 22.5 ± 2.7 | 28.4 ± 2.2 | 48.1 ± 1.2 | 25.6 ± 2.0 | 22.1 ± 3.6 | 32.6 ± 4.0 | 0.1 ± 8.7 | 15.6 ± 5.4 | 18.1 ± 4.5 | **67.9 ± 2.5** | 11.7 ± 1.6 |
| Chronic Obstructive Pulmonary Disease | 70.2 ± 1.1 | 72.2 ± 0.5 | [72.3] ± 1.7 | 21.6 ± 1.6 | 36.0 ± 4.3 | 67.8 ± 0.7 | 23.0 ± 1.6 | 22.0 ± 6.9 | 54.0 ± 3.1 | 48.6 ± 2.7 | 11.4 ± 6.8 | 6.7 ± 7.8 | **78.1 ± 1.5** | 18.9 ± 2.9 |
| Received Cholesterol Screening | −19.8 ± 8.1 | −15.3 ± 7.4 | −16.1 ± 10.3 | [3.3] ± 3.5 | [3.3] ± 3.5 | −6.9 ± 13.9 | **10.3 ± 3.0** | −10.1 ± 2.6 | −52.7 ± 19.8 | −838.6 ± 50.8 | 0.9 ± 3.6 | −4.9 ± 7.9 | −264.6 ± 47.8 | 1.6 ± 1.5 |
| Received Dental Visit | 66.0 ± 1.4 | 68.9 ± 2.0 | [73.2] ± 0.6 | 31.4 ± 0.3 | 32.8 ± 3.9 | 66.0 ± 1.6 | 33.2 ± 0.3 | −534.3 ± 497.1 | 53.4 ± 3.6 | −36.2 ± 8.6 | 22.8 ± 3.6 | 23.3 ± 4.2 | **73.8 ± 2.4** | 22.9 ± 0.8 |
| Health Average | 60.4 | −20.9 | [62.8] | 25.7 | −124.6 | 59.5 | 25.8 | 1.5 | 31.7 | −46.5 | 22.1 | 20.1 | 44.4 | 23.6 |
| **Socioeconomic** | | | | | | | | | | | | | | |
| Median Household Income | 47.6 ± 3.4 | 41.3 ± 1.5 | [49.5] ± 2.0 | 26.4 ± 2.2 | 24.3 ± 3.6 | 45.0 ± 0.5 | 26.6 ± 1.5 | 24.2 ± 4.8 | 40.7 ± 4.6 | −1.9 ± 3.4 | 26.1 ± 3.1 | 26.8 ± 2.9 | **61.2 ± 1.1** | 17.2 ± 6.5 |
| Median Home Value | [79.7] ± 1.8 | 77.7 ± 1.1 | 76.1 ± 0.7 | 57.0 ± 1.5 | 61.0 ± 1.9 | 77.8 ± 0.4 | 57.2 ± 3.5 | 57.6 ± 2.7 | 71.4 ± 0.6 | 37.1 ± 2.6 | 56.4 ± 1.2 | 56.4 ± 1.9 | **81.7 ± 0.9** | 45.9 ± 6.9 |
| Night Lights | [73.7] ± 1.7 | 60.8 ± 2.0 | 65.8 ± 2.9 | 2.4 ± 12.9 | 35.8 ± 2.6 | 64.3 ± 3.1 | 13.2 ± 4.5 | 13.2 ± 4.5 | 56.9 ± 7.2 | −2.8 ± 3.9 | 41.6 ± 6.6 | 13.2 ± 1.0 | 41.6 ± 11.7 | 14.2 ± 1.4 |
| Population Density | [48.3] ± 3.6 | 34.5 ± 3.6 | 45.0 ± 5.3 | 27.6 ± 2.8 | 27.7 ± 3.0 | 38.5 ± 6.4 | 24.2 ± 2.7 | 24.3 ± 5.9 | 22.8 ± 10.1 | −10.1 ± 0.3 | 30.4 ± 1.7 | 22.2 ± 4.6 | −13.2 ± 7.3 | 31.5 ± 0.3 |
| Poverty Rate | [47.0] ± 5.1 | 41.9 ± 5.2 | 46.9 ± 7.1 | 11.1 ± 5.9 | 30.2 ± 2.1 | 43.6 ± 6.0 | 20.0 ± 2.9 | 26.5 ± 4.1 | 28.8 ± 6.4 | 26.1 ± 4.2 | 22.0 ± 2.2 | 6.4 ± 7.8 | **56.6 ± 7.2** | 14.1 ± 4.5 |
| Socioeconomic Average | [59.3] | 51.2 | 56.7 | 24.9 | 35.8 | 53.8 | 29.2 | 29.2 | 44.1 | 9.7 | 29.3 | 25.0 | 45.6 | 24.6 |
| **Environment** | | | | | | | | | | | | | | |
| Elevation | 99.7 ± 0.0 | [99.8] ± 0.0 | [99.8] ± 0.0 | [99.7] ± 0.0 | [99.8] ± 0.0 | 29.3 ± 57.6 | [99.7] ± 0.0 | [99.7] ± 0.0 | [99.8] ± 0.0 | 84.7 ± 3.8 | [99.7] ± 0.0 | [99.8] ± 0.0 | 98.4 ± 0.5 | [99.7] ± 0.0 |
| Tree Cover | [66.7] ± 2.2 | 46.7 ± 1.2 | 52.9 ± 3.8 | 17.4 ± 2.8 | 29.1 ± 4.5 | 51.4 ± 2.4 | 17.2 ± 2.1 | 16.7 ± 1.8 | 29.5 ± 4.7 | 43.5 ± 3.9 | 15.5 ± 1.1 | 18.3 ± 2.3 | [53.1] ± 1.0 | 14.2 ± 3.0 |
| Environment Average | [83.2] | 73.2 | 76.4 | 58.6 | 64.4 | 40.4 | 58.4 | 58.2 | 64.6 | 64.1 | 57.6 | 59.0 | 75.8 | 57.0 |
| Average over Categories | [67.6] | 34.5 | 65.3 | 36.4 | −8.1 | 51.2 | 37.8 | 29.6 | 46.8 | 9.1 | 36.3 | 34.7 | 55.3 | 35.1 |

| | UrbanFusion | GAIR | GeoCLIP PP2-M | $SatCLIP_{L10}$ | $SatCLIP_{L40}$ | GeoCLIP MP-16 | $SatCLIP_{L10}$ S2-100K | $SatCLIP_{L40}$ S2-100K | GPS2Vec tag | GPS2Vec visual | CSP FMoW | CSP iNat | PDFM Google | Identity $y \sim g(c)$ |
|---|---|---|---|---|---|---|---|---|---|---|---|---|---|---|
| **Linear** | | | | | | | | | | | | | | |
| *Regression (%R² ↑)* | | | | | | | | | | | | | | |
| Crime Incidence | [88.5] | 85.4 | 84.0 | 67.2 | 69.1 | 86.3 | 67.4 | 66.9 | 84.4 | 76.5 | 50.8 | 52.5 | 74.5 | 22.1 |
| Urban Perception (6 tasks*) | [18.8] | 17.4 | 15.5 | 9.5 | 9.5 | **19.2** | 9.6 | 9.5 | - | - | 4.8 | 4.8 | - | 1.3 |
| ZIP Code (29 tasks*) | [75.1] | 70.5 | 70.0 | 69.2 | 69.7 | 69.2 | 69.4 | 68.8 | 55.6 | 48.6 | 41.5 | 42.2 | 74.0 | 3.0 |
| *Classification (%F1↑)* | | | | | | | | | | | | | | |
| Land Cover | 65.6 | 65.4 | [67.1] | 55.9 | 56.1 | **69.1** | 55.1 | 56.3 | 53.3 | 54.4 | 39.3 | 44.4 | 51.3 | 34.4 |
| Land Use – Coarse | 59.3 | [61.7] | 61.6 | 57.5 | 57.2 | **62.2** | 56.2 | 57.3 | 58.6 | 54.2 | 54.4 | 54.5 | - | 48.2 |
| Land Use – Fine | [55.2] | 54.7 | 54.2 | 48.9 | 49.2 | 55.1 | 48.0 | 47.3 | 48.3 | 46.6 | 45.7 | 45.7 | - | 42.7 |
| **MLP** | | | | | | | | | | | | | | |
| *Regression (%R² ↑)* | | | | | | | | | | | | | | |
| Crime Incidence | [90.8] ± 0.2 | 87.5 ± 0.2 | 86.5 ± 0.1 | 76.4 ± 0.3 | 76.6 ± 0.3 | 89.2 ± 0.2 | 74.2 ± 0.6 | 74.6 ± 0.9 | 86.6 ± 0.9 | 73.3 ± 0.7 | 27.0 ± 0.3 | 27.1 ± 0.2 | 78.2 ± 1.3 | 28.1 ± 0.2 |
| Urban Perception (6 tasks*) | [13.8] | 6.4 | 4.8 | 8.7 | 7.0 | 13.4 | 8.9 | 7.9 | - | - | 4.2 | 3.8 | - | 3.9 |
| ZIP Code (29 tasks*) | [71.8] | 68.8 | −15.9 | 64.0 | 66.2 | 22.1 | 64.5 | 68.9 | 46.8 | 9.1 | 36.3 | 34.7 | 55.3 | 35.1 |
| *Classification (%F1↑)* | | | | | | | | | | | | | | |
| Land Cover | 67.1 ± 0.2 | 67.3 ± 0.5 | [68.0] ± 0.4 | 58.2 ± 0.3 | 57.3 ± 0.3 | **68.0** ± 0.4 | 56.3 ± 0.6 | 56.9 ± 0.6 | 53.6 ± 0.5 | 53.7 ± 1.5 | 42.4 ± 2.5 | 41.5 ± 2.4 | 52.3 ± 0.4 | 44.4 ± 0.0 |
| Land Use – Coarse | [61.7] ± 0.6 | [61.7] ± 0.6 | 61.2 ± 0.6 | 59.1 ± 1.5 | 57.4 ± 0.2 | **62.2** ± 0.5 | 58.2 ± 0.8 | 58.4 ± 0.5 | 57.9 ± 0.4 | 55.5 ± 0.7 | 55.2 ± 2.6 | 53.6 ± 4.2 | - | 54.8 ± 3.8 |
| Land Use – Fine | [55.7] ± 0.6 | 55.2 ± 0.8 | 53.6 ± 0.8 | 49.2 ± 0.7 | 49.7 ± 0.2 | **56.4** ± 0.6 | 49.5 ± 0.5 | 49.3 ± 0.7 | 49.4 ± 0.3 | 47.7 ± 1.1 | 45.6 ± 1.7 | 45.5 ± 1.7 | - | 45.1 ± 1.2 |
| *Selected Modalities* | | | | | | | | | | | | | | |
| Crime (USA) | OSM+Coords | Coords | Coords | RS+Coords | RS+Coords | Coords | RS+Coords | RS+Coords | Coords | Coords | Coords | Coords | Coords | - |
| Land Use (USA) | SV+POI | SV+RS | SV+Coords | RS+Coords | RS+Coords | SV+Coords | RS+Coords | RS+Coords | Coords | Coords | Coords | Coords | Coords | - |
| Land Use (EU) – Coarse | SV+OSM | SV+Coords | SV+Coords | RS+Coords | RS+Coords | SV+Coords | RS+Coords | RS+Coords | Coords | Coords | Coords | Coords | Coords | - |
| Land Use (EU) – Fine | SV+OSM+POI+Coords | SV+RS+Coords | SV+Coords | RS+Coords | RS+Coords | SV+Coords | RS+Coords | RS+Coords | Coords | Coords | Coords | Coords | Coords | - |

*Detailed results in Tables 18, 19, 21, and 20.

Table 17: **Evaluation of *Multimodal Spatial Encoding*.** Best results are shown in **bold**, the second-best are underlined, and top scores across all models trained on the same dataset (PP2-M) are indicated in [brackets]. MLP results include standard deviations computed over 5 random seeds. The full names of all modality abbreviations are provided in Appendix A.

| | UrbanFusion | GAIR | GeoCLIP PP2-M | SatCLIP$_{L10}$ | SatCLIP$_{L40}$ | GeoCLIP MP-16 | SatCLIP$_{L10}$ S2-100K | SatCLIP$_{L40}$ S2-100K | GPS2Vec tag visual | CSP FMoW | CSP iNat | PDFM Google | Identity $y \sim g(c)$ |
|---|---|---|---|---|---|---|---|---|---|---|---|---|---|
| **Linear** | | | | | | | | | | | | | |
| *Regression (%R² ↑)* | | | | | | | | | | | | | |
| Cleanliness | [**7.4**] | 6.7 | 5.2 | 3.2 | 3.5 | **7.2** | 3.3 | 3.0 | - | 0.9 | 0.8 | - | 0.0 |
| Depressiveness | [12.5] | 11.0 | 9.5 | 6.9 | 6.9 | **12.7** | 6.9 | 7.0 | - | 3.1 | 3.1 | - | 0.7 |
| Beauty | [22.7] | 21.1 | 19.6 | 11.4 | 11.5 | **23.3** | 12.1 | 12.2 | - | 6.4 | 6.0 | - | 2.3 |
| Safety | [27.1] | 25.7 | 22.5 | 14.5 | 14.1 | **27.6** | 13.9 | 14.0 | - | 8.4 | 8.6 | - | 2.2 |
| Liveliness | [**23.7**] | 21.2 | 19.2 | 10.1 | 10.1 | 23.6 | 10.4 | 10.1 | - | 3.0 | 3.1 | - | 0.4 |
| Wealth | [19.1] | 19.0 | 17.1 | 11.1 | 10.7 | **20.8** | 10.8 | 10.7 | - | 7.0 | 6.9 | - | 2.1 |
| Average | [18.8] | 17.4 | 15.5 | 9.5 | 9.5 | **19.2** | 9.6 | 9.5 | - | 4.8 | 4.8 | - | 1.3 |
| **MLP** | | | | | | | | | | | | | |
| *Regression (%R² ↑)* | | | | | | | | | | | | | |
| Cleanliness | [**2.9**] ± 0.3 | −0.0 ± 0.0 | −5.3 ± 0.2 | −0.5 ± 1.0 | −2.7 ± 5.4 | 1.7 ± 0.3 | −0.0 ± 0.0 | −0.0 ± 0.0 | - | 0.6 ± 0.1 | −0.4 ± 1.2 | - | 0.4 ± 0.3 |
| Depressiveness | [**8.1**] ± 0.2 | −2.9 ± 5.5 | 3.8 ± 0.1 | 6.1 ± 0.1 | 5.1 ± 0.1 | 5.9 ± 0.4 | 6.1 ± 0.1 | 5.9 ± 0.2 | - | 3.2 ± 0.1 | 2.3 ± 0.8 | - | 2.8 ± 0.3 |
| Beauty | [17.7] ± 0.2 | 9.9 ± 0.3 | 9.4 ± 0.4 | 11.1 ± 0.1 | 9.7 ± 0.4 | **17.9** ± 0.2 | 11.5 ± 0.4 | 9.6 ± 0.3 | - | 6.2 ± 0.1 | 5.6 ± 0.2 | - | 5.5 ± 0.4 |
| Safety | [22.1] ± 0.2 | 12.5 ± 0.3 | 7.9 ± 0.3 | 14.2 ± 0.1 | 12.5 ± 0.3 | **21.3** ± 0.2 | 14.2 ± 0.3 | 13.5 ± 0.1 | - | 7.0 ± 0.3 | 7.1 ± 0.6 | - | 6.8 ± 0.3 |
| Liveliness | [17.5] ± 0.1 | 10.3 ± 0.4 | 6.1 ± 0.3 | 10.8 ± 0.1 | 8.7 ± 0.2 | **18.2** ± 0.2 | 11.0 ± 0.1 | 9.8 ± 0.1 | - | 2.5 ± 0.6 | 2.4 ± 0.7 | - | 2.6 ± 0.4 |
| Wealth | [14.6] ± 0.2 | 8.7 ± 0.2 | 6.6 ± 0.3 | 10.3 ± 0.1 | 8.7 ± 0.1 | **15.4** ± 0.2 | 10.6 ± 0.1 | 8.7 ± 0.2 | - | 5.5 ± 1.2 | 5.5 ± 0.5 | - | 5.2 ± 0.4 |
| Average | [**13.8**] | 6.4 | 4.8 | 8.7 | 7.0 | 13.4 | 8.9 | 7.9 | - | 4.2 | 3.8 | - | 3.9 |
| **Selected modalities** | | | | | | | | | | | | | |
| Cleanliness | SV | SV | SV | RS+Coords | RS | SV | RS | RS | - | Coords | Coords | - | - |
| Depressiveness | SV+OSM | SV | SV+Coords | RS+Coords | RS+Coords | SV | RS+Coords | RS+Coords | - | Coords | Coords | - | - |
| Beauty | SV+OSM | SV | SV | RS+Coords | RS+Coords | SV | RS+Coords | RS+Coords | - | Coords | Coords | - | - |
| Safety | SV+OSM | SV | SV | RS+Coords | RS+Coords | SV+Coords | RS+Coords | RS+Coords | - | Coords | Coords | - | - |
| Liveliness | SV+OSM | SV | SV | RS+Coords | RS+Coords | SV | RS+Coords | RS+Coords | - | Coords | Coords | - | - |
| Wealth | SV+OSM+POI | SV | SV | RS+Coords | RS+Coords | SV | RS+Coords | RS+Coords | - | Coords | Coords | - | - |

Table 18: **Evaluation of *Multimodal Spatial Encoding* for the Place Pulse 2.0 Urban Perception tasks.** Best results are shown in **bold**, the second-best are underlined, and top scores across all models trained on the same dataset (PP2-M) are indicated in [brackets]. MLP results include standard deviations computed over 5 random seeds. The full names of all modality abbreviations are provided in Appendix A.

Table 19: **Evaluation of Multimodal Spatial Encoding** for various ZIP Code tasks. Best results are shown in **bold**, the second-best are underlined, and top scores across all models trained on the same dataset (PP2-M) are indicated in [brackets].

| | UrbanFusion | GAIR | GeoCLIP PP2-M | SatCLIP_L10 PP2-M | SatCLIP_L40 | GeoCLIP MP-16 | SatCLIP_L10 S2-100K | SatCLIP_L40 S2-100K | GPS2Vec tag | GPS2Vec visual | CSP FMoW | CSP iNat | PDFM Google | Identity $y \sim g(c)$ |
|---|---|---|---|---|---|---|---|---|---|---|---|---|---|---|
| *Linear Regression (%$R^2$ ↑)* | | | | | | | | | | | | | | |
| **Health** | | | | | | | | | | | | | | |
| High Cholesterol | [52.9] | 51.2 | 50.2 | 49.7 | 50.2 | 48.4 | 50.5 | 52.5 | 25.3 | 27.0 | −0.0 | 12.2 | **55.1** | −5.1 |
| Physical Health Not Good | [82.8] | 79.3 | 81.0 | 72.0 | 76.3 | 81.3 | 72.0 | 70.5 | 61.8 | 57.4 | 40.6 | 43.9 | **88.5** | −0.5 |
| Stroke | [72.1] | 64.8 | 68.7 | 59.4 | 66.9 | 64.6 | 61.0 | 61.2 | 65.7 | 56.8 | 35.7 | 35.0 | **84.7** | 1.8 |
| Binge Drinking | 70.5 | [73.7] | [76.3] | 66.2 | 69.5 | 68.5 | 49.9 | 50.1 | 68.1 | 57.6 | 19.6 | 23.9 | **85.4** | 4.2 |
| Physical Inactivity | [83.8] | 79.7 | 82.2 | 70.5 | 74.2 | 82.2 | 69.9 | 67.7 | 56.0 | 52.5 | 43.3 | 44.2 | **92.0** | 6.3 |
| Received Annual Checkup | 84.2 | 81.7 | [84.7] | 80.7 | 81.1 | 83.7 | 81.9 | 81.9 | 83.1 | 77.1 | 75.2 | 78.2 | **93.0** | 54.6 |
| Cancer (Excl. Skin) | 44.2 | 43.3 | 42.4 | [46.2] | 40.3 | 39.1 | 44.4 | 44.7 | 4.3 | 14.5 | 14.6 | 15.3 | **62.1** | 9.3 |
| Diabetes | 78.6 | 75.3 | [79.0] | 69.1 | 76.9 | 75.1 | 70.7 | 71.3 | 68.8 | 51.4 | 38.6 | 41.8 | **86.8** | 5.8 |
| Mental Health Not Good | [66.6] | 41.3 | 45.0 | 40.3 | 44.3 | 55.3 | 35.3 | 30.4 | −54.6 | 30.4 | 29.5 | 31.4 | **66.8** | 0.5 |
| Coronary Heart Disease | 62.1 | 59.4 | 63.8 | 63.2 | [65.9] | 62.9 | 62.1 | 62.2 | 50.4 | 46.5 | 32.4 | 35.2 | **74.8** | −4.5 |
| High Blood Pressure | [66.4] | 59.8 | 63.9 | 54.8 | 58.3 | 60.4 | 55.5 | 57.2 | 54.8 | 58.2 | 30.1 | 28.0 | **82.7** | 8.2 |
| High Blood Pressure (Medicated) | 54.7 | [56.9] | 50.2 | 54.2 | 51.6 | 50.8 | 54.5 | 54.3 | 38.4 | 49.4 | 19.0 | 35.4 | **65.8** | 10.4 |
| Obesity | [79.9] | 77.2 | 78.6 | 71.4 | 72.2 | 78.1 | 71.1 | 71.4 | 69.2 | 55.2 | 49.7 | 47.7 | **86.2** | 7.6 |
| Sleep Less Than 7 Hours | [83.3] | 80.9 | 83.0 | 73.3 | 77.9 | 78.5 | 73.2 | 75.5 | 76.1 | 59.4 | 53.1 | 54.4 | **92.2** | 20.6 |
| Smoking | [80.7] | 76.4 | 77.8 | 69.8 | 73.6 | 75.5 | 66.6 | 69.3 | 63.5 | 47.1 | 41.1 | 42.0 | **85.9** | 3.2 |
| Asthma | [79.7] | 70.8 | 73.1 | 59.5 | 62.5 | 69.5 | 57.9 | 61.6 | 68.1 | 48.9 | 45.4 | 47.8 | **83.4** | 9.9 |
| Chronic Kidney Disease | 72.5 | 65.3 | 70.5 | 59.7 | 68.9 | 66.7 | 62.6 | 62.3 | 58.9 | 59.2 | 33.3 | 35.0 | **82.8** | −1.5 |
| Arthritis | [60.1] | 53.3 | 55.7 | 54.2 | 56.3 | 53.4 | 56.0 | 57.3 | 42.2 | 46.5 | 9.7 | 29.5 | **74.3** | 3.1 |
| Chronic Obstructive Pulmonary Disease | [75.4] | 70.4 | 68.9 | 68.9 | 70.9 | 67.9 | 67.1 | 68.4 | 63.6 | 50.4 | 38.4 | 39.9 | **80.6** | −1.0 |
| Received Cholesterol Screening | [31.2] | 16.4 | 18.3 | 25.5 | 17.4 | 26.3 | 21.1 | 15.8 | −152.0 | 15.3 | 10.6 | 4.6 | **54.1** | 1.0 |
| Received Dental Visit | [78.2] | 59.9 | 75.0 | 63.7 | 64.5 | 75.0 | 60.6 | 55.6 | 64.0 | 45.7 | 35.3 | 38.5 | **87.3** | 17.3 |
| Health Average | [69.5] | 63.7 | 66.1 | 60.6 | 62.8 | 64.9 | 59.2 | 59.1 | 41.7 | 47.9 | 33.1 | 36.4 | **79.3** | 7.2 |
| **Socioeconomic** | | | | | | | | | | | | | | |
| Median Household Income | [56.9] | 41.1 | 54.9 | 44.8 | 42.3 | 47.9 | 41.5 | 39.3 | 39.7 | 22.9 | 32.4 | 32.9 | **66.2** | 0.9 |
| Median Home Value | [78.0] | 77.5 | 77.5 | 75.7 | 73.3 | [78.3] | 75.5 | 76.4 | 73.2 | 17.3 | 60.5 | 61.0 | **81.1** | 0.2 |
| Night Lights | 75.7 | 76.1 | 73.6 | 76.7 | [77.2] | 72.7 | _77.1_ | 75.9 | 70.8 | 58.1 | 14.3 | 12.8 | 74.0 | 12.0 |
| Population Density | [67.1] | 64.1 | 56.8 | 60.8 | 58.4 | 59.0 | _64.8_ | 63.1 | 54.5 | 29.7 | 30.3 | 30.6 | 62.6 | 2.3 |
| Poverty Rate | [67.7] | 48.8 | 57.2 | 50.4 | 47.8 | 56.8 | 46.5 | 43.1 | 36.9 | 24.3 | 26.4 | 29.3 | _65.7_ | 4.9 |
| Socioeconomic Average | [69.1] | 61.5 | 64.0 | 61.7 | 59.8 | 62.9 | 61.1 | 59.6 | 55.0 | 30.5 | 32.8 | 33.3 | **69.9** | 4.1 |
| **Environment** | | | | | | | | | | | | | | |
| Elevation | 99.3 | 99.5 | [99.7] | [99.8] | [99.8] | **99.8** | **99.8** | **99.8** | 99.5 | 82.4 | **99.8** | **99.8** | 97.9 | 1.8 |
| Tree Cover | [74.3] | 73.2 | 60.1 | 71.1 | 73.0 | 59.6 | 76.1 | _75.7_ | 40.7 | 52.3 | 17.5 | 14.0 | 47.8 | −6.5 |
| Environment Average | [86.8] | 86.4 | 79.9 | 85.4 | 86.4 | 79.7 | _87.9_ | _87.8_ | 70.1 | 67.4 | 58.6 | 56.9 | 72.8 | −2.4 |
| Average over Categories | [75.1] | 70.5 | 70.0 | 69.2 | 69.7 | 69.2 | 69.4 | 68.8 | 55.6 | 48.6 | 41.5 | 42.2 | _74.0_ | 3.0 |

| MLP (%R² ↑) | UrbanFusion | GAIR | GeoCLIP PP2-M | SatCLIP$_{L10}$ | SatCLIP$_{L40}$ | GeoCLIP MP-16 | SatCLIP$_{L10}$ S2-100K | SatCLIP$_{L40}$ S2-100K | GPS2Vec tag | GPS2Vec visual | CSP FMoW | CSP iNat | PDFM Google | Identity $y \sim g(c)$ |
|---|---|---|---|---|---|---|---|---|---|---|---|---|---|---|
| **Health** | | | | | | | | | | | | | | |
| High Cholesterol | 38.8±2.5 | 33.8±4.0 | 35.1±2.1 | [48.9]±2.1 | 45.1±5.7 | 38.6±5.7 | 36.3±3.2 | **48.9**±3.7 | 7.0±3.4 | −131.2±11.1 | 5.7±7.2 | 5.1±1.8 | 11.3±7.8 | 1.8±3.2 |
| Physical Health Not Good | 78.9±1.9 | **84.1**±0.7 | 82.7±0.7 | 76.1±0.8 | 76.3±1.1 | 78.0±0.3 | 76.6±1.1 | 72.1±1.2 | 59.6±4.4 | 50.0±2.7 | 28.4±3.4 | 8.3±21.7 | 81.4±1.0 | 29.0±2.8 |
| Stroke | 71.8±1.0 | 73.8±0.3 | 68.9±1.1 | 71.4±1.5 | 70.1±3.6 | 65.2±0.6 | 71.0±0.6 | 64.3±1.1 | 54.4±5.0 | 49.1±4.7 | 15.6±4.7 | 22.9±1.5 | **81.5**±1.0 | 20.6±1.0 |
| Binge Drinking | **78.0**±1.0 | 70.1±2.4 | 74.0±2.3 | 61.6±0.9 | 70.2±1.9 | 73.3±1.7 | 64.7±1.0 | 59.6±2.7 | 65.6±1.0 | −0.4±5.6 | 24.5±1.6 | 18.4±4.9 | 67.2±1.7 | 26.5±7.0 |
| Physical Inactivity | 80.9±2.2 | 83.9±1.3 | 81.9±1.3 | 75.3±1.3 | 75.0±0.5 | 78.1±0.3 | 74.6±0.9 | 70.6±1.8 | 62.8±7.1 | 55.5±1.8 | 33.8±1.7 | 27.2±3.8 | **85.6**±0.6 | 31.3±2.0 |
| Received Annual Checkup | 77.1±2.5 | 77.4±1.7 | 81.7±1.3 | 82.3±1.3 | [83.8]±1.0 | 81.4±2.5 | 81.6±0.9 | **83.3**±0.7 | 65.6±5.8 | −140.0±47.2 | 68.8±0.5 | 70.8±3.1 | 18.0±9.6 | 69.2±4.0 |
| Cancer (Excl. Skin) | 39.4±3.2 | 35.9±0.5 | 28.2±2.2 | 37.3±2.8 | 37.8±3.9 | 34.9±1.3 | 40.2±2.6 | 44.5±3.8 | 6.6±1.3 | −43.2±7.9 | 10.3±2.8 | 10.0±4.2 | **61.1**±1.8 | 13.7±1.8 |
| Diabetes | 76.6±1.5 | 78.7±1.0 | [79.9]±0.5 | 76.6±0.4 | 78.4±1.8 | 74.9±0.3 | 76.7±0.6 | 72.9±0.4 | 63.0±2.4 | 57.8±2.0 | 19.6±1.7 | 27.6±2.1 | **84.5**±0.5 | 25.1±2.9 |
| Mental Health Not Good | [57.5]±2.7 | 52.5±1.7 | 50.1±2.7 | −9.4±28.2 | 41.5±2.8 | 47.1±2.1 | 34.3±7.3 | 41.4±3.6 | −161.0±3.7 | −15.2±3.0 | 12.9±7.6 | −6.6±0.5 | 29.6±8.7 | 13.8±3.4 |
| Coronary Heart Disease | 62.8±0.8 | [65.4]±0.8 | 62.2±2.1 | 64.4±1.3 | 64.8±3.2 | 61.5±0.7 | 63.8±1.2 | 60.4±1.8 | 44.8±3.0 | 40.9±2.6 | 5.9±8.0 | 15.6±0.9 | 72.1±2.6 | 16.3±2.5 |
| High Blood Pressure | [69.5]±0.6 | 67.7±3.3 | 65.6±1.6 | 63.5±3.0 | 66.2±3.4 | 59.5±1.7 | 65.2±1.7 | 60.8±1.3 | 48.8±2.9 | 28.1±8.1 | 12.3±3.1 | 14.8±2.7 | 72.7±2.2 | 12.3±1.9 |
| High Blood Pressure (Medicated) | [41.8]±3.5 | 12.8±4.1 | 41.7±3.5 | 11.8±2.5 | 50.6±1.7 | −1906.8±3890.5 | 2.9±7.1 | 11.6±11.1 | −7.0±9.7 | −236.6±29.9 | 16.6±4.1 | 15.7±6.1 | −26.6±17.6 | 19.1±1.9 |
| Obesity | 77.1±1.7 | 78.0±0.6 | [78.5]±1.0 | 77.4±1.9 | 75.1±1.7 | 75.4±0.6 | 75.2±0.8 | 74.1±1.0 | 62.3±3.5 | 30.9±3.7 | 40.0±3.2 | 34.5±2.1 | 67.2±2.6 | 36.9±1.1 |
| Sleep Less Than 7 Hours | 81.6±0.7 | 82.1±1.5 | [84.4]±0.3 | 77.6±0.7 | 76.6±2.5 | 74.5±1.0 | 79.0±0.8 | 74.3±1.8 | 64.3±5.4 | 22.7±4.7 | 47.6±2.1 | 40.6±4.2 | 67.8±2.1 | 44.9±3.3 |
| Smoking | 76.6±1.1 | [78.7]±1.9 | 76.8±1.5 | 74.5±1.0 | 72.5±1.9 | 73.1±0.1 | 73.0±1.8 | 69.6±1.5 | 57.0±3.0 | 45.6±3.3 | 26.7±2.9 | 26.4±2.7 | **80.0**±0.4 | 26.9±2.2 |
| Asthma | 67.8±2.6 | [69.3]±1.7 | [74.1]±0.8 | 63.6±1.2 | 62.6±1.4 | 65.4±0.9 | 67.2±0.6 | 61.3±0.6 | 31.9±10.0 | −9.5±1.9 | 35.1±1.9 | 31.4±5.5 | 46.3±2.8 | 33.4±1.5 |
| Chronic Kidney Disease | 69.7±3.3 | [71.6]±1.5 | 68.7±1.9 | 69.6±1.5 | 70.9±2.4 | 59.6±1.5 | 68.8±1.0 | 65.5±1.8 | 53.6±2.9 | 45.9±4.5 | 10.2±7.7 | 16.4±5.0 | 77.7±2.2 | 20.3±2.5 |
| Arthritis | 54.6±1.6 | 57.9±2.2 | 52.6±1.9 | [61.1]±1.6 | 59.0±4.1 | 49.1±0.8 | 60.5±2.7 | 56.9±1.7 | 32.6±4.0 | 0.1±8.7 | 15.6±5.4 | 18.1±4.5 | 67.9±2.5 | 11.7±1.6 |
| Chronic Obstructive Pulmonary Disease | 72.0±0.7 | [75.3]±1.0 | 68.1±4.6 | 71.3±1.9 | 71.8±1.8 | 68.6±0.7 | 70.7±1.1 | 66.8±2.8 | 54.0±3.1 | 48.6±2.7 | 11.4±6.8 | 6.7±7.8 | 78.1±1.5 | 18.9±2.9 |
| Received Cholesterol Screening | [−13.0]±12.2 | −73.9±9.5 | −5244.3±10410.4 | −87.2±12.0 | −112.2±8.8 | −10.3±3.9 | −108.1±32.6 | −57.3±22.6 | −52.7±19.8 | −838.6±50.8 | 0.9±3.6 | −4.9±7.9 | −264.6±47.8 | **1.6**±1.5 |
| Received Dental Visit | [70.5]±1.0 | 60.2±0.6 | 69.2±2.3 | 63.3±1.3 | 61.5±1.9 | 66.4±2.5 | 61.7±1.6 | 63.7±1.3 | 53.4±3.6 | −36.2±8.6 | 22.8±3.6 | 23.3±4.2 | **73.8**±2.4 | 22.9±0.8 |
| Health Average | [63.3] | 58.8 | −186.7 | 53.9 | 57.0 | −33.0 | 54.1 | 55.5 | 31.7 | −46.5 | 22.1 | 20.1 | 44.4 | 23.6 |
| **Socioeconomic** | | | | | | | | | | | | | | |
| Median Household Income | [51.0]±3.1 | 44.8±4.0 | 48.2±1.3 | 42.9±1.5 | 41.8±1.7 | 41.4±0.7 | 43.5±2.2 | 47.3±1.5 | 40.7±4.6 | −1.9±3.4 | 26.1±3.1 | 26.8±2.9 | **61.2**±1.1 | 17.2±6.5 |
| Median Home Value | [80.2]±1.6 | 77.1±0.6 | 77.3±0.3 | 79.1±0.4 | 76.9±2.8 | 75.6±1.7 | 79.0±1.4 | 78.9±0.6 | 71.4±0.6 | 37.1±2.6 | 56.4±1.2 | 56.4±1.9 | **81.7**±0.9 | 45.9±6.9 |
| Night Lights | [74.2]±1.1 | 71.7±0.3 | 68.8±1.5 | 67.8±1.2 | 67.5±1.0 | 68.7±0.9 | 70.5±1.0 | 56.9±7.2 | 56.9±7.2 | −2.8±3.9 | 11.5±6.6 | 13.2±2.1 | 11.6±11.7 | 14.2±1.4 |
| Population Density | 59.5±4.6 | [64.2]±1.7 | 50.0±3.8 | 44.1±3.5 | 50.4±1.3 | 49.6±2.5 | 39.3±24.9 | **69.9**±1.8 | 22.8±10.1 | −10.1±0.3 | 30.4±1.7 | 22.2±4.6 | −13.2±7.3 | 31.5±0.3 |
| Poverty Rate | [58.8]±6.6 | 55.5±2.6 | [58.0]±1.1 | 46.9±2.2 | 44.3±2.0 | 47.9±4.1 | 54.0±1.7 | 51.9±3.9 | 28.8±6.4 | 26.1±4.2 | 22.0±2.2 | 6.4±7.8 | 56.6±7.2 | 14.1±4.5 |
| Socioeconomic Average | [64.7] | 62.7 | 60.5 | 56.2 | 56.8 | 56.6 | 52.2 | 64.5 | 44.1 | 9.7 | 29.3 | 25.0 | 45.6 | 24.6 |
| **Environment** | | | | | | | | | | | | | | |
| Elevation | 99.7±0.1 | [99.8]±0.0 | [99.8]±0.0 | 99.7±0.0 | [99.8]±0.0 | 29.3±57.6 | 99.1±0.1 | 99.7±0.0 | **99.8**±0.0 | 84.7±3.8 | 99.7±0.0 | **99.8**±0.0 | 98.4±0.5 | 99.7±0.0 |
| Tree Cover | [75.2]±0.7 | 69.8±3.1 | 57.2±0.6 | 63.8±2.1 | 69.8±1.2 | 56.1±2.7 | **75.4**±1.3 | 73.9±0.6 | 29.5±4.7 | 43.5±3.9 | 15.5±1.1 | 18.3±2.3 | 53.1±1.0 | 14.2±3.0 |
| Environment Average | [87.4] | 84.8 | 78.5 | 81.8 | 84.8 | 42.7 | 87.2 | 86.8 | 64.6 | 64.1 | 57.6 | 59.0 | 75.8 | 57.0 |
| Average over Categories | [71.8] | 68.8 | −15.9 | 64.0 | 66.2 | 22.1 | 64.5 | 68.9 | 46.8 | 9.1 | 36.3 | 34.7 | 55.3 | 35.1 |

Table 20: **Evaluation of *Multimodal Spatial Encoding* for various ZIP Code tasks.** Best results are shown in **bold**, the second-best are underlined, and top scores across all models trained on the same dataset (PP2-M) are indicated in [brackets]. MLP results include standard deviations computed over 5 random seeds.

| | UrbanFusion | GAIR $_{PP2\text{-}M}$ | GeoCLIP $_{PP2\text{-}M}$ | SatCLIP$_{L10}$ | SatCLIP$_{L40}$ | GeoCLIP $_{MP\text{-}16}$ | SatCLIP$_{L10}$ S2-100K | SatCLIP$_{L40}$ S2-100K | GPS2Vec tag | GPS2Vec visual | CSP FMoW | CSP iNat | PDFM Google | Identity $y \sim g(c)$ |
|---|---|---|---|---|---|---|---|---|---|---|---|---|---|---|
| **Health** | | | | | | | | | | | | | | |
| High Cholesterol | RS+OSM+POI+Coords | RS | SV+Coords | RS+Coords | RS+Coords | SV+Coords | RS | RS+Coords | Coords | Coords | Coords | Coords | Coords | - |
| Physical Health Not Good | SV+RS+POI+Coords | SV+RS+Coords | SV+Coords | RS+Coords | RS+Coords | SV+Coords | RS+Coords | RS+Coords | Coords | Coords | Coords | Coords | Coords | - |
| Stroke | SV+RS+Coords | SV+RS+Coords | SV+Coords | RS+Coords | RS+Coords | SV+Coords | RS+Coords | RS+Coords | Coords | Coords | Coords | Coords | Coords | - |
| Binge Drinking | OSM+Coords | RS+Coords | Coords | RS+Coords | RS+Coords | Coords | RS | RS | Coords | Coords | Coords | Coords | Coords | - |
| Physical Inactivity | SV+RS+POI+Coords | SV+RS+Coords | SV+Coords | RS+Coords | RS+Coords | SV+Coords | RS+Coords | RS+Coords | Coords | Coords | Coords | Coords | Coords | - |
| Received Annual Checkup | SV+RS+OSM+Coords | RS+Coords | Coords | RS+Coords | RS+Coords | Coords | RS | RS+Coords | Coords | Coords | Coords | Coords | Coords | - |
| Cancer (Excl. Skin) | RS | RS | SV+Coords | RS | RS+Coords | SV+Coords | RS | RS+Coords | Coords | Coords | Coords | Coords | Coords | - |
| Diabetes | SV+RS+POI+Coords | SV+RS+Coords | SV+Coords | RS+Coords | RS+Coords | SV+Coords | RS+Coords | RS+Coords | Coords | Coords | Coords | Coords | Coords | - |
| Mental Health Not Good | SV+RS+OSM+POI+Coords | SV+RS | Coords | RS | RS+Coords | SV+Coords | RS | RS+Coords | Coords | Coords | Coords | Coords | Coords | - |
| Coronary Heart Disease | OSM+Coords | SV+RS+Coords | SV+Coords | RS+Coords | RS+Coords | SV+Coords | RS+Coords | RS+Coords | Coords | Coords | Coords | Coords | Coords | - |
| High Blood Pressure | RS+OSM+Coords | SV+RS+Coords | SV+Coords | RS+Coords | RS+Coords | SV+Coords | RS+Coords | RS+Coords | Coords | Coords | Coords | Coords | Coords | - |
| High Blood Pressure (Medicated) | RS+OSM+Coords | RS | Coords | RS | RS+Coords | SV+Coords | RS | RS+Coords | Coords | Coords | Coords | Coords | Coords | - |
| Obesity | RS+Coords | SV+RS+Coords | SV+Coords | RS+Coords | RS+Coords | SV+Coords | RS+Coords | RS+Coords | Coords | Coords | Coords | Coords | Coords | - |
| Sleep Less Than 7 Hours | SV+RS+Coords | SV+RS+Coords | SV+Coords | RS+Coords | RS+Coords | SV+Coords | RS+Coords | RS+Coords | Coords | Coords | Coords | Coords | Coords | - |
| Smoking | SV+RS+POI+Coords | SV+RS+Coords | SV+Coords | RS+Coords | RS+Coords | SV+Coords | RS | RS+Coords | Coords | Coords | Coords | Coords | Coords | - |
| Asthma | SV+RS+OSM+POI+Coords | RS+Coords | SV+Coords | RS+Coords | RS+Coords | SV+Coords | RS+Coords | RS+Coords | Coords | Coords | Coords | Coords | Coords | - |
| Chronic Kidney Disease | RS+OSM+Coords | SV+RS+Coords | SV+Coords | RS+Coords | RS+Coords | SV+Coords | RS+Coords | RS+Coords | Coords | Coords | Coords | Coords | Coords | - |
| Arthritis | Coords | SV+RS+Coords | Coords | RS+Coords | RS+Coords | SV+Coords | RS+Coords | RS+Coords | Coords | Coords | Coords | Coords | Coords | - |
| Chronic Obstructive Pulmonary Disease | SV+RS+POI+Coords | SV+RS+Coords | SV+Coords | RS+Coords | RS+Coords | SV+Coords | RS+Coords | RS+Coords | Coords | Coords | Coords | Coords | Coords | - |
| Received Cholesterol Screening | RS+OSM+POI+Coords | RS | SV+Coords | RS | RS | SV+Coords | RS | RS | Coords | Coords | Coords | Coords | Coords | - |
| Received Dental Visit | SV+RS+POI+Coords | SV+RS | SV+Coords | RS+Coords | RS+Coords | SV+Coords | RS | RS | Coords | Coords | Coords | Coords | Coords | - |
| **Socioeconomic** | | | | | | | | | | | | | | |
| Median Household Income | SV+RS+POI+Coords | SV+RS | SV+Coords | RS+Coords | RS+Coords | SV+Coords | RS+Coords | RS | Coords | Coords | Coords | Coords | Coords | - |
| Median Home Value | SV+RS+OSM+POI+Coords | SV+RS+Coords | SV+Coords | RS+Coords | RS+Coords | SV+Coords | RS+Coords | RS+Coords | Coords | Coords | Coords | Coords | Coords | - |
| Night Lights | RS+OSM+POI+Coords | SV+RS | SV+Coords | RS+Coords | RS+Coords | SV+Coords | RS+Coords | RS+Coords | Coords | Coords | Coords | Coords | Coords | - |
| Population Density | RS+OSM+POI | SV+RS | SV | RS | RS | SV | RS | RS+Coords | Coords | Coords | Coords | Coords | Coords | - |
| Poverty Rate | SV+RS+OSM+POI+Coords | RS | SV+Coords | RS | RS | SV+Coords | RS | RS | Coords | Coords | Coords | Coords | Coords | - |
| **Environment** | | | | | | | | | | | | | | |
| Elevation | OSM+Coords | RS+Coords | Coords | RS+Coords | Coords | Coords | RS+Coords | RS+Coords | Coords | Coords | Coords | Coords | Coords | - |
| Tree Cover | RS+OSM+POI | RS | SV+Coords | RS | RS | SV+Coords | RS | RS | Coords | Coords | Coords | Coords | Coords | - |

Table 21: **Selected modalities of *Multimodal Spatial Encoding* for various ZIP Code tasks.** The full names of all modality abbreviations are provided in Appendix A.

| | UrbanFusion | GAIR | GeoCLIP PP2-M | SatCLIP$_{L10}$ | SatCLIP$_{L40}$ | Identity $y \sim g(c)$ |
|---|---|---|---|---|---|---|
| **Linear** | | | | | | |
| *Regression* (%R$^2$ ↑) | | | | | | |
| Crime Incidence | **76.7** | 68.0 | 44.3 | 63.4 | 63.6 | 10.4 |
| Urban Perception (avg. 6 tasks*) | **21.2** | 20.4 | 20.0 | 12.9 | 13.2 | 6.6 |
| ZIP Code (weighted avg. 29 tasks*) | 56.7 | **62.5** | 42.1 | 60.8 | 59.8 | 17.7 |
| | | | | | | |
| *Classification* (%F1↑) | | | | | | |
| Land Cover | **70.9** | 69.9 | 68.6 | 61.3 | 61.1 | 53.9 |
| Land Use – Coarse | **66.7** | 65.9 | 60.6 | 60.6 | 59.4 | 55.1 |
| Land Use – Fine | **61.0** | 60.4 | 55.3 | 53.5 | 53.7 | 49.5 |
| **MLP** | | | | | | |
| *Regression* (%R$^2$ ↑) | | | | | | |
| Crime (USA) | **85.3** ± 0.3 | 77.1 ± 0.2 | 38.7 ± 0.6 | 74.3 ± 0.6 | 74.0 ± 0.3 | 60.8 ± 4.1 |
| Perception PP 2.0 (avg. 6 tasks*) | **17.0** | 11.9 | 11.5 | 7.0 | 7.4 | 8.0 |
| ZIP Code (weighted avg. 29 tasks*) | 42.5 | 21.6 | −241.3 | 39.5 | 35.0 | **52.0** |
| | | | | | | |
| *Classification* (%F1↑) | | | | | | |
| Land Use (USA) | **72.1** ± 0.5 | 69.5 ± 0.4 | 69.5 ± 0.4 | 62.6 ± 0.8 | 62.4 ± 0.2 | 55.5 ± 0.9 |
| Land Use (EU) – Coarse | 66.9 ± 0.6 | **67.5** ± 0.4 | 65.0 ± 0.7 | 60.9 ± 0.4 | 61.3 ± 0.6 | 55.1 ± 0.0 |
| Land Use (EU) – Fine | **62.0** ± 0.3 | 61.6 ± 0.4 | 59.4 ± 0.8 | 56.1 ± 1.0 | 56.2 ± 1.5 | 49.5 ± 0.0 |
| **Selected modalities** | | | | | | |
| Crime (USA) | RS+OSM | RS | SV | RS | RS | - |
| Land Use (USA) | SV | SV | SV | RS | RS | - |
| Land Use (EU) – Coarse | SV+RS | SV+RS | SV | RS | RS | - |
| Land Use (EU) – Fine | SV+RS+OSM | SV+RS | SV | RS | RS | - |

*Detailed results in Tables 23, 24, 26, and 25.

Table 22: ***Cross-Regional Generalization* using all available modalities as inputs.** Best results are in **bold**, the second-best are underlined. MLP results include standard deviations across 5 random seeds. The full names of all modality abbreviations are provided in Appendix A.

| | UrbanFusion | GAIR | GeoCLIP PP 2.0 | SatCLIP$_{L10}$ | SatCLIP$_{L40}$ | Identity $y \sim g(c)$ |
|---|---|---|---|---|---|---|
| **Linear** | | | | | | |
| *Regression* (%R$^2$ ↑) | | | | | | |
| Cleanliness | **9.3** | 9.1 | 8.5 | 3.9 | 4.3 | 1.0 |
| Depressiveness | **15.2** | 14.5 | 14.1 | 9.3 | 9.9 | 5.2 |
| Beauty | **24.8** | 24.7 | 23.5 | 14.3 | 15.0 | 8.1 |
| Safety | **30.1** | 28.5 | 28.5 | 18.3 | 18.3 | 9.5 |
| Liveliness | **24.7** | 23.4 | 22.9 | 14.1 | 14.4 | 5.5 |
| Wealth | **23.4** | 22.5 | 22.2 | 17.3 | 17.4 | 10.0 |
| Average | **21.2** | 20.4 | 20.0 | 12.9 | 13.2 | 6.6 |
| **MLP** | | | | | | |
| *Regression* (%R$^2$ ↑) | | | | | | |
| Cleanliness | **4.4** ± 0.2 | −1.3 ± 0.2 | −0.0 ± 0.0 | −1.8 ± 3.5 | −0.0 ± 0.0 | 1.2 ± 0.5 |
| Depressiveness | **10.1** ± 0.4 | 5.2 ± 0.2 | 3.9 ± 0.2 | 2.0 ± 0.1 | 1.7 ± 0.2 | 5.7 ± 0.3 |
| Beauty | **20.7** ± 0.2 | 16.6 ± 0.1 | 15.6 ± 0.3 | 9.4 ± 0.3 | 10.7 ± 0.1 | 9.7 ± 0.1 |
| Safety | **27.0** ± 0.2 | 20.1 ± 0.1 | 21.3 ± 0.3 | 12.0 ± 0.2 | 12.4 ± 0.3 | 12.3 ± 0.4 |
| Liveliness | **20.2** ± 0.3 | 14.9 ± 0.1 | 12.6 ± 0.2 | 8.3 ± 0.4 | 8.4 ± 0.1 | 7.4 ± 0.2 |
| Wealth | **19.5** ± 0.8 | 15.9 ± 0.4 | 15.7 ± 0.4 | 11.9 ± 0.4 | 11.5 ± 0.5 | 11.7 ± 0.4 |
| Average | **17.0** | 11.9 | 11.5 | 7.0 | 7.4 | 8.0 |
| **Selected modalities** | | | | | | |
| Cleanliness | SV | SV | SV | RS | RS | - |
| Depressiveness | SV | SV+RS | SV | RS | RS | - |
| Beauty | SV+OSM | SV | SV | RS | RS | - |
| Safety | SV | SV | SV | RS | RS | - |
| Liveliness | SV | SV | SV | RS | RS | - |
| Wealth | SV | SV | SV | RS | RS | - |

Table 23: ***Cross-Regional Generalization* using all available modalities as inputs for the Place Pulse 2.0 Urban Perception tasks.** Best results are in **bold**, the second-best are underlined. MLP results include standard deviations across 5 random seeds. The full names of all modality abbreviations are provided in Appendix A.

| | UrbanFusion | GAIR | GeoCLIP | SatCLIP$_{L10}$ | SatCLIP$_{L40}$ | Identity |
|---|---|---|---|---|---|---|
| | | | PP 2.0 | | | $y \sim g(c)$ |
| *Linear* (%R$^2$ ↑) | | | | | | |
| **Health** | | | | | | |
| High Cholesterol | 20.5 | **29.2** | 28.5 | 20.3 | 19.9 | 1.1 |
| Physical Health Not Good | **79.5** | 74.2 | 49.8 | 74.8 | 73.1 | 11.9 |
| Stroke | **67.7** | 65.1 | 47.7 | 64.4 | 60.5 | 19.4 |
| Binge Drinking | **80.4** | 74.5 | 71.2 | 73.9 | 71.2 | 40.6 |
| Physical Inactivity | **76.8** | 73.5 | 53.5 | 72.4 | 71.7 | 17.1 |
| Received Annual Checkup | 66.1 | **75.7** | 60.7 | 72.9 | 67.2 | 51.0 |
| Cancer (Excl. Skin) | 15.0 | 30.7 | 20.6 | **32.9** | 26.8 | −1.4 |
| Diabetes | **77.3** | 70.6 | 57.6 | 70.4 | 67.3 | 28.8 |
| Mental Health Not Good | 64.6 | **69.5** | 25.8 | 66.5 | 66.6 | −0.5 |
| Coronary Heart Disease | 42.4 | 48.0 | 43.1 | **51.1** | 47.0 | 13.9 |
| High Blood Pressure | 70.4 | **74.3** | 61.3 | 73.3 | 67.2 | 38.4 |
| High Blood Pressure (Medicated) | 30.2 | **44.0** | 35.0 | 34.7 | 26.4 | 17.1 |
| Obesity | **81.4** | 77.0 | 63.3 | 74.0 | 74.3 | 28.7 |
| Sleep Less Than 7 Hours | **71.9** | 66.8 | 54.1 | 67.6 | 65.4 | 27.7 |
| Smoking | 69.7 | **70.2** | 40.8 | 67.3 | 67.0 | 1.9 |
| Asthma | **75.1** | 74.4 | 19.8 | 73.9 | 72.2 | 8.2 |
| Chronic Kidney Disease | 55.9 | 63.1 | 49.5 | **63.8** | 58.9 | 17.7 |
| Arthritis | 39.3 | **47.6** | 38.0 | 38.0 | 32.5 | 11.2 |
| Chronic Obstructive Pulmonary Disease | 64.1 | **65.6** | 39.8 | 61.8 | 61.4 | 6.6 |
| Received Cholesterol Screening | 47.4 | **58.9** | 19.6 | 55.7 | 55.8 | −0.3 |
| Received Dental Visit | 68.4 | **68.5** | 44.1 | 66.4 | 66.2 | 8.9 |
| Health Average | 60.2 | **62.9** | 44.0 | 60.8 | 58.0 | 16.6 |
| **Socioeconomic** | | | | | | |
| Median Household Income | 50.5 | 66.6 | 40.6 | 65.4 | **68.7** | 4.4 |
| Median Home Value | 63.1 | 73.4 | 56.0 | 69.9 | **74.3** | 34.9 |
| Night Lights | 70.0 | **71.7** | 46.0 | 61.1 | 65.1 | 12.2 |
| Population Density | **64.0** | 57.2 | 58.9 | 60.5 | 57.6 | 34.6 |
| Poverty Rate | **58.8** | 55.7 | 19.8 | 57.3 | 55.8 | −0.6 |
| Socioeconomic Average | 61.3 | **64.9** | 44.3 | 62.8 | 64.3 | 17.1 |
| **Environment** | | | | | | |
| Elevation | 44.7 | **63.2** | 33.8 | 61.5 | 54.9 | 13.5 |
| Tree Cover | 52.4 | 56.5 | 42.4 | 56.2 | **59.0** | 25.1 |
| Environment Average | 48.6 | **59.8** | 38.1 | 58.8 | 57.0 | 19.3 |
| Average over Categories | 56.7 | **62.5** | 42.1 | 60.8 | 59.8 | 17.7 |

Table 24: *Cross-Regional Generalization* **using all available modalities as inputs for ZIP Code tasks.** Best results are in **bold**, the second-best are underlined.

| | UrbanFusion | GAIR | GeoCLIP PP2-M | SatCLIP$_{L10}$ | SatCLIP$_{L40}$ | Identity $y \sim g(c)$ |
|---|---|---|---|---|---|---|
| *MLP* (%R$^2$ ↑) | | | | | | |
| **Health** | | | | | | |
| High Cholesterol | $\underline{16.1} \pm 4.0$ | $-49.4 \pm 2.4$ | $-58.1 \pm 6.0$ | $-68.7 \pm 15.2$ | $-7.3 \pm 3.7$ | $\mathbf{25.9} \pm 2.7$ |
| Physical Health Not Good | $\mathbf{79.2} \pm 2.3$ | $71.0 \pm 3.9$ | $52.8 \pm 8.1$ | $68.6 \pm 2.9$ | $\underline{76.0} \pm 2.6$ | $72.6 \pm 5.0$ |
| Stroke | $57.7 \pm 5.1$ | $\underline{61.8} \pm 5.2$ | $50.5 \pm 2.9$ | $54.8 \pm 4.3$ | $60.4 \pm 4.0$ | $\mathbf{66.1} \pm 3.3$ |
| Binge Drinking | $79.3 \pm 0.7$ | $81.9 \pm 1.8$ | $-110.4 \pm 150.4$ | $\mathbf{83.8} \pm 1.0$ | $75.0 \pm 1.4$ | $\underline{82.0} \pm 1.4$ |
| Physical Inactivity | $\mathbf{73.4} \pm 2.0$ | $\underline{73.2} \pm 2.3$ | $49.2 \pm 1.5$ | $66.8 \pm 5.2$ | $72.3 \pm 1.0$ | $72.0 \pm 2.6$ |
| Received Annual Checkup | $\underline{59.0} \pm 5.8$ | $-54.4 \pm 10.9$ | $-17348.5 \pm 21225.8$ | $-145.0 \pm 13.0$ | $-64.9 \pm 14.5$ | $\mathbf{67.8} \pm 2.7$ |
| Cancer (Excl. Skin) | $8.3 \pm 5.2$ | $11.2 \pm 8.7$ | $\mathbf{25.0} \pm 2.5$ | $-4.2 \pm 6.4$ | $-1.2 \pm 6.2$ | $-2.6 \pm 5.8$ |
| Diabetes | $\mathbf{72.6} \pm 3.9$ | $69.5 \pm 2.9$ | $55.7 \pm 6.9$ | $64.7 \pm 2.7$ | $69.5 \pm 2.7$ | $\underline{71.6} \pm 2.3$ |
| Mental Health Not Good | $57.6 \pm 6.6$ | $\mathbf{65.5} \pm 2.2$ | $31.6 \pm 8.6$ | $58.5 \pm 3.3$ | $\underline{62.3} \pm 3.4$ | $53.7 \pm 4.6$ |
| Coronary Heart Disease | $\underline{50.3} \pm 4.7$ | $48.4 \pm 2.4$ | $42.2 \pm 6.3$ | $-1.4 \pm 17.6$ | $43.7 \pm 2.8$ | $\mathbf{50.9} \pm 1.7$ |
| High Blood Pressure | $72.2 \pm 3.2$ | $65.9 \pm 1.5$ | $55.1 \pm 10.4$ | $52.9 \pm 10.4$ | $\underline{72.9} \pm 2.8$ | $\mathbf{75.2} \pm 3.7$ |
| High Blood Pressure (Medicated) | $\underline{27.6} \pm 4.3$ | $-80.4 \pm 28.3$ | $-62.2 \pm 25.7$ | $-163.2 \pm 46.8$ | $-59.0 \pm 10.7$ | $\mathbf{34.3} \pm 11.0$ |
| Obesity | $\mathbf{81.5} \pm 2.3$ | $76.6 \pm 1.9$ | $66.5 \pm 3.5$ | $18.6 \pm 124.3$ | $\underline{80.7} \pm 0.9$ | $78.4 \pm 4.5$ |
| Sleep Less Than 7 Hours | $\mathbf{60.6} \pm 1.6$ | $54.9 \pm 6.0$ | $50.0 \pm 9.4$ | $40.3 \pm 15.3$ | $53.7 \pm 6.4$ | $\underline{55.1} \pm 4.4$ |
| Smoking | $65.4 \pm 3.0$ | $\mathbf{70.0} \pm 3.2$ | $47.1 \pm 5.8$ | $55.2 \pm 14.2$ | $\underline{67.1} \pm 5.9$ | $63.2 \pm 2.5$ |
| Asthma | $68.9 \pm 3.6$ | $73.9 \pm 4.3$ | $47.7 \pm 3.5$ | $69.3 \pm 10.8$ | $\underline{74.6} \pm 2.0$ | $\mathbf{78.0} \pm 3.6$ |
| Chronic Kidney Disease | $\mathbf{63.9} \pm 2.7$ | $\underline{59.9} \pm 0.9$ | $-11.3 \pm 1.1$ | $41.5 \pm 5.9$ | $56.0 \pm 2.5$ | $59.3 \pm 7.7$ |
| Arthritis | $\underline{39.7} \pm 2.8$ | $20.2 \pm 11.1$ | $28.4 \pm 6.5$ | $14.6 \pm 5.5$ | $28.1 \pm 4.9$ | $\mathbf{47.5} \pm 2.5$ |
| Chronic Obstructive Pulmonary Disease | $\underline{64.7} \pm 2.2$ | $56.1 \pm 2.9$ | $-7.7 \pm 0.8$ | $56.9 \pm 4.1$ | $64.4 \pm 1.3$ | $\mathbf{70.1} \pm 1.5$ |
| Received Cholesterol Screening | $\underline{22.3} \pm 24.7$ | $-80.6 \pm 13.4$ | $-34.0 \pm 8.8$ | $-109.4 \pm 25.6$ | $-82.5 \pm 12.5$ | $\mathbf{34.9} \pm 6.0$ |
| Received Dental Visit | $\underline{70.3} \pm 2.3$ | $69.3 \pm 3.7$ | $40.3 \pm 7.6$ | $\mathbf{71.0} \pm 2.0$ | $63.9 \pm 3.2$ | $51.9 \pm 6.3$ |
| Health Average | $\underline{56.7}$ | $36.4$ | $-809.1$ | $15.5$ | $38.4$ | $\mathbf{57.5}$ |
| | | | | | | |
| **Socioeconomic** | | | | | | |
| Median Household Income | $52.9 \pm 3.5$ | $56.6 \pm 3.8$ | $44.4 \pm 2.0$ | $\underline{64.5} \pm 1.0$ | $\mathbf{64.9} \pm 4.1$ | $50.1 \pm 15.6$ |
| Median Home Value | $\underline{65.7} \pm 1.8$ | $\mathbf{69.5} \pm 1.6$ | $53.0 \pm 4.7$ | $64.4 \pm 4.9$ | $\underline{65.7} \pm 4.5$ | $27.2 \pm 2.3$ |
| Night Lights | $\mathbf{63.8} \pm 9.7$ | $49.0 \pm 6.2$ | $\underline{51.6} \pm 4.4$ | $34.5 \pm 7.4$ | $35.2 \pm 9.8$ | $34.2 \pm 12.1$ |
| Population Density | $\mathbf{60.9} \pm 4.8$ | $-112.4 \pm 66.7$ | $51.1 \pm 5.5$ | $-3.0 \pm 0.7$ | $2.1 \pm 10.8$ | $\underline{57.8} \pm 2.0$ |
| Poverty Rate | $58.6 \pm 3.9$ | $\underline{64.4} \pm 4.1$ | $27.3 \pm 17.5$ | $55.1 \pm 3.5$ | $57.1 \pm 2.4$ | $\mathbf{65.9} \pm 2.3$ |
| Socioeconomic Average | $\mathbf{60.4}$ | $25.4$ | $45.5$ | $43.1$ | $45.0$ | $\underline{47.0}$ |
| | | | | | | |
| **Environment** | | | | | | |
| Elevation | $44.9 \pm 7.4$ | $\mathbf{53.2} \pm 1.9$ | $33.1 \pm 5.0$ | $49.7 \pm 3.5$ | $22.5 \pm 49.8$ | $\underline{49.8} \pm 6.0$ |
| Tree Cover | $-24.3 \pm 66.4$ | $-47.1 \pm 0.5$ | $46.3 \pm 4.3$ | $\mathbf{70.4} \pm 1.3$ | $20.8 \pm 52.0$ | $\underline{53.0} \pm 3.2$ |
| Environment Average | $10.3$ | $3.1$ | $39.7$ | $\mathbf{60.0}$ | $21.6$ | $\underline{51.4}$ |
| | | | | | | |
| Average over Categories | $\underline{42.5}$ | $21.6$ | $-241.3$ | $39.5$ | $35.0$ | $\mathbf{52.0}$ |

Table 25: *Cross-Regional Generalization* **using all available modalities as inputs for ZIP Code tasks.** Best results are in **bold**, the second-best are underlined. MLP results include standard deviations across 5 random seeds.

| | UrbanFusion | GAIR | GeoCLIP PP 2.0 | SatCLIP$_{L10}$ | SatCLIP$_{L40}$ | Identity $y \sim g(c)$ |
|---|---|---|---|---|---|---|
| **Health** | | | | | | |
| High Cholesterol | SV+RS+OSM+POI | SV+RS | SV | RS | RS | - |
| Physical Health Not Good | RS+POI | SV+RS | SV | RS | RS | - |
| Stroke | SV+RS+POI | SV+RS | SV | RS | RS | - |
| Binge Drinking | SV+RS+POI | SV+RS | SV | RS | RS | - |
| Physical Inactivity | RS+POI | RS | SV | RS | RS | - |
| Received Annual Checkup | SV+RS | SV+RS | SV | RS | RS | - |
| Cancer (Excl. Skin) | RS+POI | RS | SV | RS | RS | - |
| Diabetes | SV+RS+POI | SV+RS | SV | RS | RS | - |
| Mental Health Not Good | RS+POI | RS | SV | RS | RS | - |
| Coronary Heart Disease | SV+RS | SV | SV | RS | RS | - |
| High Blood Pressure | SV+RS | SV+RS | SV | RS | RS | - |
| High Blood Pressure (Medicated) | RS+OSM | SV+RS | SV | RS | RS | - |
| Obesity | RS+POI | SV+RS | SV | RS | RS | - |
| Sleep Less Than 7 Hours | RS+POI | RS | SV | RS | RS | - |
| Smoking | RS+POI | RS | SV | RS | RS | - |
| Asthma | RS+POI | RS | SV | RS | RS | - |
| Chronic Kidney Disease | SV+RS | SV+RS | SV | RS | RS | - |
| Arthritis | SV+RS+POI | SV+RS | SV | RS | RS | - |
| Chronic Obstructive Pulmonary Disease | SV+RS+POI | SV+RS | SV | RS | RS | - |
| Received Cholesterol Screening | RS+OSM | RS | SV | RS | RS | - |
| Received Dental Visit | RS+POI | RS | SV | RS | RS | - |
| | | | | | | |
| **Socioeconomic** | | | | | | |
| Median Household Income | SV+POI | RS | SV | RS | RS | - |
| Median Home Value | SV+OSM+POI | RS | SV | RS | RS | - |
| Night Lights | RS+OSM+POI | SV+RS | SV | RS | RS | - |
| Population Density | SV+OSM | SV+RS | SV | RS | RS | - |
| Poverty Rate | RS+POI | RS | SV | RS | RS | - |
| | | | | | | |
| **Environment** | | | | | | |
| Elevation | SV+RS+OSM+POI | SV+RS | SV | RS | RS | - |
| Tree Cover | SV+RS+OSM | SV+RS | SV | RS | RS | - |

Table 26: **Selected modalities of *Cross-Regional Generalization* for various ZIP Code tasks.** The full names of all modality abbreviations are provided in Appendix A.

