# OpenReview forum: "UrbanFusion: Stochastic Multimodal Fusion for Contrastive Learning of Robust Spatial Representations"
_ICLR.cc/2026/Conference — Submitted to ICLR 2026_

### Official Review · Reviewer_ijWt · 2025-10-19

**Soundness:** 2
**Presentation:** 2
**Contribution:** 2
**Rating:** 4
**Confidence:** 4

**Summary:**

This paper presents UrbanFusion which is a multimodal geospatial representation learning framework designed to integrate multiple data sources, e.g., remote sensing imagery, street-view imagery, POI and OSM, and location information (lat – lon). The paper uses a self-supervised learning strategy, termed SMF, which combines contrastive learning for aligning representations from different modality subsets with a modality reconstruction objective for fusion.

**Strengths:**

A pertinent challenge for spatial data and spatial foundation model, especially the fusion of data of different modalities.
The tasks are comprehensive, including many downstream tasks and 56 cities.
The paper is generally well-written and clearly structured.

**Weaknesses:**

Since the features from different modalities are extracted from frozen backbone encoders, the the primary contribution seems to be on fusion, which refers to specific combination strategy SMF involving contrastive learning and reconstruction with modality masking. This combination is applied effectively to spatial data, but the underlying principle of combining these two types of self-supervised objectives is not entirely new in the broader representation learning literature. It is noteworthy that empirical evidence is ample, while I doubt this is the focus of ICLR.
To the ICLR community, the novel insight given specifically by the paper (if we talk about those beyond showing that combining these existing techniques works well for this multimodal geospatial setting) is limited. This means that it seems to be more of empirical finding paper.
There are some minor points as well. Using image encoder for OSM naturally leads to information loss and there are papers using object-level information for feature extraction. An example is
Bai, Lubian, et al. "GeoLink: Empowering Remote Sensing Foundation Model with OpenStreetMap Data." arXiv preprint arXiv:2509.26016 (2025).
The rationality of the baseline comparisons is questionable. The pretrained backbone of the compared models differs (e.g., some are based on CLIP, others on GeoCLIP), which undermines the fairness of the comparison. Would it be more appropriate to compare with multimodal remote sensing foundation models that integrate OSM data? It would be also useful to compare a baseline with all of your frozen features concatenated, since the main contribution is on fusion.

**Questions:**

See weakness
Further details of section 4.5 would be useful (not easy to follow)

---

> ### Author Response · Authors · 2025-11-21
>
> Thank you for the review of our paper. Below, we provide additional experiments and clarifications that we believe address the raised concerns.
>
> [W1: combining contrastive loss and reconstruction loss is not new] We would like to clarify that our contribution is not merely a combination of two losses, but a novel training strategy (SMF), as noted by reviewer Qd7u and Reviewer jLmY. The core part of SMF is the masking strategy shown in Figure 1: A set of modalities is masked and the embedding is computed, then the complementary set is masked and the embedding is computed, and we use a contrastive loss on these embeddings. This is a new version of a contrastive loss function that has several advantages, including 1) support of more modalities (in contrast to standard contrastive loss, we do not need to take the loss between all pairs of modalities, which scales quadratically in the number of modalities, 2) ease of finetuning on new modalities, 3) enables training on incomplete modality sets (Section 4.6).
>
> [W2: too much empirical evidence] In addition to empirical evidence, we also provide a theoretical analysis of our method, initially in Appendix D.2, now added to the main paper (section 3.1). This theoretical analysis, together with the experiment on synthetic data in section 4.5, shows that our method brings general advantages for retaining information in multimodal representation learning, which is a core focus of ICLR. We kindly ask you to consider this part of section 3.1 in your evaluation of the paper.
>
> [W3: better OSM vector encoder such as GeoLink] Thank you for this interesting reference, which indeed could improve the performance of our model. We could not incorporate the GeoLink encoder at the time of writing our paper, since the GeoLink paper and code was only published on the 30th of September. To the best of our knowledge, GeoLink was the first vector-based OSM encoding method, as also stated in their related work section. Furthermore, developing good modality encoders is a research problem on its own and out of the scope of this paper, where we intentionally used the same encoders as the baseline methods whenever possible to ensure comparability.
>
> We have added this point to the Discussion section: “Incorporating better encoders such as recent work on vectorized OSM embeddings~\cite{bai2025geolink} could improve the performance.” (see revised paper)
>
> [W4: GeoCLIP vs CLIP backbone] We believe this is a misunderstanding. GeoCLIP is one of the baseline methods that we compare to. It uses CLIP as a backbone for encoding Street View Images (and other baselines also use CLIP). No other method is using GeoCLIP as a backbone.
>
> For our method, we take the same encoders as prior work for the sake of maximal comparability. Specifically, we use 1) CLIP for encoding SV images (as GeoCLIP and GAIR are doing), 2) the same ViT for encoding remote sensing as SatCLIP, 3) the same coordinate encoder as GeoCLIP and GAIR. With this setup, the comparison is fair and the performance gains of our method can clearly be attributed to the new training scheme, which 1) allows for training on more modalities than all other methods, and 2) fuses the modalities instead of treating them independently.
>
> [W5: Comparison to multimodal remote sensing models] We agree that multimodal RS models integrating OSM data are highly relevant in the broader geospatial learning space. The goal of our work is, however, to provide a better location embedding method, i.e., a method that takes geographic coordinates and encodes them into a learnt latent representation. This is a very different goal than the purpose of remote sensing foundation models models, which aim to classify or segment RS images. Thus, these approaches are incomparable.

---

> ### Author Response · Authors · 2025-11-21
>
> [W6: comparison to concatenated features] Thank you for this interesting suggestion. We have conducted a new experiment where we concatenate the encodings of all modalities to compare against the results of our fused representation. However, we would like to clarify that this comparison is not possible for the primary use case of our method—location-only embedding—because, in that scenario, only location data is available at inference time, making concatenation with other modalities infeasible. Therefore, we performed this experiment in the context described in Section 4.3, where all modalities are available at inference and results are reported for the best-performing subset. The results are presented below in the table:
>
> | Task                           | Fused representation (compare Table 3) | Concatenated encodings
> |--------------------------------|---------------------------|-------------------------|
> | Crime incidence                | 88.5                     | 62.6                       |
> | Urban perception (avg 6 tasks) | 18.8                       | 17.1                   |
> | ZIP Code (avg 29 tasks)        | 75.1                     | 70.8                   |
> | Land cover classification      | 65.6                     | 67.3                    |
> | Land use (coarse)              | 59.3                      | 59.7                      |
> | Land use (fine)                | 55.2                         | 53.1                          |
>
> On the regression tasks, the fused representation clearly outperforms the concatenated encodings. Furthermore, this experiment actually clarifies the discrepancies noted in Table 3: In Table 3, our model performed best in all tasks except for the “land cover” and “coarse land use” classification tasks. The new findings show that our concatenated encoding outperforms the fused embedding exactly in these tasks, indicating that these tasks benefit from access to the full encoded information rather than a fused representation.
>
> We have updated the manuscript to reflect these findings (see Section 4.3): *“UrbanFusion underperforms comparable baselines only in land use classification tasks. Further investigation (see Appendix C.7) revealed that simply concatenating the output of encoding models results in better performance for UrbanFusion in these particular tasks. This suggests that such tasks may not necessitate a fused representation.”*
>
> [Q1: Further details of section 4.5 would be useful] Thank you for this suggestion. We have added a paragraph that describes the synthetic data generation in detail. Please refer to the revised paper for the full text. In summary, we construct random geographic data with two synthetic modalities that have three dimensions: Two dimensions that identify the location and are thus redundant between both modalities, and one dimension with unique information. We can then evaluate three downstream tasks on this synthetic data: Geolocalizaition (requires redundant information), predicting the unique features (requires retaining unique information), and predicting the sum of unique features (requires synergistic information).

---

> > ### Comment · Reviewer_ijWt · 2025-11-22
> >
> > Response appreciated. The inclusion of the PID theoretical analysis and the concatenation baseline results are interesting and provide clarity on some certain points, but my concerns about the fundamental novelty and the handling of OSM data remain. SMF’s complementary masking combined with reconstruction and contrastive training resembles many established multimodal masked autoencoding objective, albeit the reconstruction is elevated to embedding level.
> > I am still concerned about how the basemap is encoded. It is an acceptable argument that GeoLink is almost a concurrent paper, but a quick search revealed quite some papers using vector-based and sort of object-oriented methods to encode spatial vector data, such as Li, Yi, et al. "Urban region representation learning with openstreetmap building footprints." Proceedings of the 29th ACM SIGKDD Conference on Knowledge Discovery and Data Mining. 2023. More can be found in this area. Rasterizing vector data almost inevitably leads to information loss e.g., topology and precise object semantics, which I reckon is a considerable limitation for this paper. In this way, I still think that the contribution of the paper is on the thinner side.

---

> > > ### Author Response · Authors · 2025-11-26
> > >
> > > **Regarding "SMF’s complementary masking combined with reconstruction and contrastive training resembles many established multimodal masked autoencoding objective"**
> > >
> > > Although SMF shares some surface-level similarities with prior multimodal masked-autoencoding frameworks, its new masking strategy results in a fundamentally different contrastive alignment. In contrast to prior work on location encoders such as GeoCLIP or SatCLIP (also published at major learning conferences), our method does not use a standard contrastive loss. Instead, it introduces a genuinely new training scheme that is relevant beyond geospatial data. This scheme enables learning a fused embedding across many modalities and directly addresses a major gap in the field, as noted in a recent review by Mai et al.: "there are several challenges remaining to be solved. [...] Representation learning models for different spatial data modalities are currently developed separately as different models, while a unified representation learning model is needed to seamlessly handle various spatial data formats simultaneously." [1].
> > >
> > > We demonstrate the advantages of our method on synthetic data and 41 real-world tasks, and we even provide a theoretical proof of its benefits. Again, only the SMF approach — not standard multimodal masking — enables (1) learning unique and synergistic information, (2) scaling to a larger number of modalities, and (3) training on heterogeneous datasets where some samples contain only subsets of modalities. We are not aware of any prior work that satisfies these properties.
> > >
> > > [1] Mai, Gengchen, et al. "The Evolution of Geospatial Artificial Intelligence." GeoAI and Human Geography: The Dawn of a New Spatial Intelligence Era. Cham: Springer Nature Switzerland, 2025. 13-27.
> > >
> > > **Regarding "Rasterizing vector data almost inevitably leads to information loss e.g., topology and precise object semantics, which I reckon is a considerable limitation for this paper"**
> > >
> > > While we agree that vector-based representations are an important emerging direction, rasterization remains a widely used and well-established practice for encoding cartographic and geospatial vector data. Notably, the paper cited by the reviewer  (Li, Yi, et al. "Urban region representation learning with openstreetmap building footprints." Proceedings of the 29th ACM SIGKDD Conference on Knowledge Discovery and Data Mining. 2023.) also rasterizes building footprints and encodes them with a CNN, stating: “we employ an ImageNet pre-trained ResNet-18 to encode building polygons into visual features.”
> > > Our approach differs in that we train encoders directly on cartographic basemaps across multiple scales, rather than relying on frozen, ImageNet-trained models that are not optimized for cartographic semantics. Moreover, the primary contribution of our work is the SMF learning framework, not a new vector-geometry encoder—advancing such encoders is a substantial research effort of its own and orthogonal to our methodological focus.
> > > For these reasons, we do not consider the rasterization-based encoder to be a major limitation; it is an intentional and empirically validated design choice that supports the framework’s core contribution.

---

> ### Author Response · Authors · 2025-12-03
> **Summary of rebuttal**
>
> Dear AC, we summarize our rebuttal to Reviewer ijWt in the following. Please see above for the full responses.
>
> **[W1]** The reviewer argued that our new training strategy, SMF, is mainly a combination of contrastive loss and reconstruction loss. This seems to be a misunderstanding of our method, since SMF is a different type of contrastive loss rather than a combination of the two (we actually show also results without reconstruction loss). The novelty is that we mask a subset of the modalities, compute the embedding, and compute the contrastive loss wrt the embedding of the complementary subset. This strategy has unique advantages that we did not find in prior work, such as 1) scalability to many modalities (no pairwise losses needed), 2) enabling training on incomplete datasets with part of the modalities, 3) learning synergistic and unique information accordion to PID. We are not aware of any other method that achieves these properties, and the reviewer also did not name any. The novelty of SMF was also pointed out as a strength by Reviewer qd7u and Reviewer jLmY.
>
> **[W2]** The reviewer argued that “empirical evidence is ample, which is not a focus of ICLR”. We believe that empirical evidence is key for showing the advantages of our method, and disagree that evaluating on 41 tasks is a weakness of our paper. However, we have revised our paper to include a theoretical proof of our method in the main paper (rather than just the appendix as before).
>
> **[W3]** The reviewer suggested that we incorporate a vector-based OSM encoder. Since his/her suggested paper was only published after the ICRL deadline, and no such methods existed previously, we believe this comment is not relevant to this submission.
>
> **[W4]** We clarified a misunderstanding of the reviewer, who criticized that “the pretrained backbone of the compared models differs (e.g., some are based on CLIP, others on GeoCLIP)”. This is not the case – GeoCLIP is one of the baseline methods that we compare to. It uses CLIP as a backbone for encoding Street View Images (and other baselines also use CLIP). No other method is using GeoCLIP as a backbone.
>
> **[W5]** The reviewer asked whether we should not rather compare to multimodal remote sensing foundation models. However, remote sensing is just one of the modalities we use, and our method clearly builds up on general location encoders such as SatCLIP or GeoCLIP, which is a very different use case than remote sensing foundation models.
>
> **[W6]** The reviewer asked for a comparison between our fused embedding and a simple concatenation of all modality-encodings. This was an interesting suggestion, and we added a new experiment in Appendix C.7, which shows that fusion is generally better, but more beneficial in some tasks than in others.
>
> **[Q1]** The reviewer asked for more details in section 4.5. We expanded this section in the revised paper.

---

### Official Review · Reviewer_Uw1v · 2025-10-28

**Soundness:** 2
**Presentation:** 3
**Contribution:** 2
**Rating:** 2
**Confidence:** 4

**Summary:**

The paper proposes UrbanFusion, a multimodal framework that fuses several inputs (street view images, satellite images, POI data and map data) using a transformer based module. One of the main contributions of the paper is the proposed stochastic multimodal fusion training strategy, which applies a combined contrastive and reconstruction loss on random subsets of modalities to learn a unified representation. The approach is evaluated on 41 downstream urban prediction tasks.

**Strengths:**

[S1] The work addresses a timely problem of learning unified location embeddings for heterogeneous spatial modalities.

[S2] The proposed stochastic multimodal fusion module provides a flexible approach for aligning and reconstructing random subsets of modalities.

[S3] The paper is well written and the methodology is easy to follow.

[S4] The empirical results show improvements over the selected baselines.

**Weaknesses:**

[W1] One of my main **concerns** and confusion is the decision to **append the raw coordinates** to the learned embeddings during downstream evaluation: *"Following prior work [1], raw geographical coordinates are concatenated to the model embeddings for evaluation."* Location is highly correlated with many of the downstream labels (e.g. house prices), so it is unclear if the performance gains come from the approach or from leveraging location information. Moreover, the cited reference [1] is misinterpreted: In SATCLIP, location embedding (not raw coordinates) are concatenated with image features to perform image classification.

[W2] The fusion of different urban modalities has been studied extensively, and more sophisticated fusion approaches exist in the literature (e.g., [2, 3]). For the specific modalities handled here, the authors seem to bypass the core challenge of different spatial resolutions by operating at the patch level [4]. Moreover, the handling of the cartographic basemap is fairly naïve, as it relies on rasterization, rather than leveraging spatial and topological structure, which has been addressed more effectively in prior work [5, 6]. Given that, the SMF component seem to be the main technical contribution here. However, its evaluation is weak: it is validated on a toy example, and the real-world ablation shows that the combination of the reconstruction loss and contrastive loss is not the best overall in the majority of the cases (Appendix C.3.).

[W3] The foundation model claim is an overstatement given the small-scale pretraining dataset and the lack of diversity across tasks. Specifically, the evaluated tasks are variations of point-based prediction problems (regression or classification) confined to the urban domain, which does not demonstrate the broad, cross-domain applicability or diverse reasoning capabilities expected of a true foundation model.

[W4] There are instances where the authors categorize SatCLIP and GeoCLIP as unimodal, although both are inherently multimodal: *"Both models clearly outperform unimodal baselines such as SatCLIP and GeoCLIP, demonstrating the advantages of
multimodal representation learning for geographic generalization"*.

[W5] Similar to W1, several "new" features are introduced when testing on housing value prediction (i.e., # of bedrooms, # of bathrooms etc.), which are already strong predictors of the housing prices on their own. Again, it's unclear whether the improvement comes from the learned embeddings.

[1] Satclip: Global, general-purpose location embeddings with satellite imagery. AAAI 2025.

[2] Urbanclip: Learning text-enhanced urban region profiling with contrastive language-image pretraining from the web. WWW 2024.

[3] Refound: Crafting a foundation model for urban region understanding upon language and visual foundations. KDD 2024.

[4] Gair: Improving multimodal geo-foundation model with geo-aligned implicit representations. arxiv 2025.

[5] Poly2Vec: Polymorphic Fourier-Based Encoding of Geospatial Objects for GeoAI Applications. ICML 2025.

[6] Geo2Vec: Shape-and Distance-Aware Neural Representation of Geospatial Entities. arxiv 2025.

**Questions:**

Could the authors address the issues mentioned in the weaknesses section? Specifically, are raw geographical coordinates appended to the embeddings for all baselines, or only for your model? Can the authors provide results on some of the tasks without appending the raw coordinates?

---

> ### Author Response · Authors · 2025-11-21
>
> Thank you for the thorough review of our paper. We provide additional experiments and clarifications in the following that should resolve the concerns.
>
> **[W1 concatenating coordinates]** We apologize for the confusion - of course we also concatenated the coordinates to the baseline methods. It thus did not impact comparability. The reason for adding the raw coordinates was the following line from the SatCLIP paper, section 3.4: “For all downstream tasks, we train multi-layer perceptron (MLP) models g with location embeddings and raw latitude/longitude coordinates as input to predict a (continuous or discrete) outcome variable y.” We believe this means they did use the raw coordinates in addition to the embedding.
> Nevertheless, we ran an additional experiment where we computed the performance on downstream task without concatenating the coordinates. Full results are added in Appendix C.5; for convenience we copy the main results (coordinates-only location embedding) here:
> | Task                           | UrbanFusion (with coords) | UrbanFusion (no coords) | Identity (only coords) |
> |--------------------------------|---------------------------|-------------------------|------------------------|
> | Housing prices                 | 78.7                      | 78.7                    | 66.2                   |
> | Energy consumption             | 20.1                      | 20.1                    | 1.5                    |
> | Crime incidence                | 87.4                      | 87.3                    | 22.1                   |
> | Urban perception (avg 6 tasks) | 9.5                       | 9.4                     | 1.3                    |
> | ZIP Code (avg 29 tasks)        | 74.3                      | 74.0                    | 3.0                    |
> | Land cover classification      | 56.9                      | 57.1                    | 34.4                   |
> | Land use (coarse)              | 58.9                      | 59.2                    | 48.2                   |
> | Land use (fine)                | 47.7                      | 50.6                    | 42.7                   |
>
> The results show that there are minimal differences between UrbanFusion with or without concatenating coordinates, while using only the coordinates performs poorly in comparison. This shows that the performance gains can clearly be attributed to our embedding approach and not the coordinates.
>
> **[W2 Basemap encoding, fusion mechanism, evaluation]**
>
> **Basemap encoding:** We agree that rasterization is a simplification. However, cartographic basemaps contain highly heterogeneous elements—points, lines, polygons, and text—and, to the best of our knowledge at the time of submission, no method provides a general and scalable solution for jointly encoding this mixture of geometric and semantic information. Designing such an encoder would be a substantial contribution on its own. We added this point to the discussion section: “Incorporating better encoders such as recent work on vectorized OSM embeddings~\cite{bai2025geolink} could improve the performance.”
>
> **Fusion mechanism**: SMF is designed as a principled, information-theoretic fusion method motivated by partial information decomposition (Appendix D.2), rather than as a more complex architecture. Its contribution lies in handling arbitrary missing-modality patterns and integrating redundant and synergistic information in a controlled manner. Methods like UrbanCLIP [2], while strong in their domain, fuse only two modalities and cannot extend to the multi-modal, coordinate-aware setting we address.
> We have added a discussion of these prior works to the related works section “Beyond coordinate–image contrastive encoders, recent research has explored richer multimodal and geometry-aware spatial representations. UrbanCLIP \citep{yan2024urbanclip} and ReFound \citep{xiao2024refound} leverage web-scale text–image alignment to enhance urban region profiling, GeoLink \citep{bai2025geolink} fuses satellite imagery with OSM-derived structural vectors, highlighting the complementary nature of cartographic and visual signals.”
>
> **Evaluation:** The synthetic experiments are not intended as a benchmark but as a controlled validation of the theory, complementing our extensive real-world ablations—the largest set of downstream tasks evaluated in this literature. Regarding the loss functions, the “best” choice is task- and modality-dependent. Crucially, for coordinate-only encoding, the main setting in spatial representation learning, the combined contrastive + reconstruction loss performs best, consistent with our theoretical motivation (see added theoretical analysis in section 3.1).

---

> > ### Author Response · Authors · 2025-11-21
> >
> > **[W3 foundation model]** Just to explain our reasoning: We believe a “foundation model” is characterized not by the size of the training data, but by a wide applicability to diverse tasks. We used the term for our model since we tested on 41 tasks, including classification & regression tasks, multiple domains like health, perception, economics and social indicators. Note that we are not confined to point-based problems, but show improved performance on the postal code datasets over Google’s Population Dynamics Foundation Model.
> >
> > However, we do not want to overstate our contribution (which is a general training method for multimodal data and a location embedding model), so we have rephrased the occurrences of “foundation model” in the text (see revised paper).
> >
> > **[W4 unimodal baselines]** We used the word “unimodal” to express that those baselines only encode a single modality aside from the geographic coordinates. We understand that this is misleading and have rephrased these occurrences in the text (see revised paper).
> >
> > **[W5 other features for housing prices]** Those features are not introduced by us, but are part of the housing price dataset we are using for the experiment. More importantly, we always compare to the “Identity” baseline, which is using the exact same features but the raw set of coordinates instead of the learned embeddings. This comparison shows clearly that the improvement comes from the learned embedding.
> >
> > **[Q1 geographic coordinates]** We refer to the results above, where we clarified that geographical coordinates were appended for all baselines, and we provide results on all tasks without appending raw coordinates.
> >
> > We believe that these clarifications address all concerns, in particular regarding the fairness of the experiments. Thank you for your consideration!

---

> ### Author Response · Authors · 2025-12-03
> **Summary of rebuttal**
>
> Dear AC, we summarize our rebuttal to Reviewer Uw1v in the following. Please see above for the full responses.
>
> **W1**: The reviewer’s “main concern” was that concatenating raw coordinates to the learnt embeddings could disturb the results and comparability to the baseline. This was a misunderstanding, since we - of course - did the same for the baseline methods, such that the fairness of the comparison was not compromised. Nevertheless, we added experiments in Appendix C.5 showing that there the results with and without concatenating coordinates differ negligibly. We believe that this should have fully answered the reviewers concern.
>
> **W2:** The reviewer observed that our main contribution is the training strategy, called SMF, since we do not provide a new method for encoding Open Street Map data or other geography-specific enhancements. We agree: If we had developed a new OSM encoder, we would not have submitted to a learning conference. Furthermore, we respectfully disagree with the reviewer on the point of our evaluation of SMF “just on a toy dataset”: We evaluate our model trained with SMF on 41 real-world tasks, we show its general advantages on a synthetic dataset, and we even provide a theoretical proof showing its advantages. As we now moved the proof to the main paper (previously in the appendix and thus potentially not visible to the reviewer), there should be clearly sufficient evidence of the superiority of our method.
>
> **W3:** Wording of our method as a “foundation model”: We have revised the paper to change the wording.
>
> **W4**: Wording of some baselines as “unimodal”: We have explained our reasoning and revised the paper to avoid the term.
>
> **W5:** The reviewer asked whether adding other features to the learnt embeddings for specific tasks (e.g., “number of rooms” for housing prices) confounds the results. Again, this is done for all baselines and thus does not impact comparability.
>
> **Q1:** Same as W1
>
> We are confident that we fully resolved the reviewer's main concern and also addressed the other concerns. It is thus unfortunate that the reviewer could not answer to our rebuttal. We thank you in advance for your consideration of these important clarifications.

---

### Official Review · Reviewer_qd7u · 2025-10-30

**Soundness:** 3
**Presentation:** 3
**Contribution:** 3
**Rating:** 8
**Confidence:** 4

**Summary:**

### Problem:
- The authors aim to build better geospatial foundation models that can understand and predict geospatial dynamics and urban phenomena at scale
- A good foundation model should:
    - Be task-agnostic
    - Be multi-modal
    - Synthesize information across modalities
    - Be robust to missing modalities
    - Have good generalization to unseen geographic regions


### Solution

- Authors propose a novel foundation model called UrbanFusion
    - UrbanFusion processes several different modalities (street-view, satellite imagery, maps, POIs, and coordinates)
    - Each modality is encoded separately, then fused with a Transformer
    - UrbanFusion is trained via a novel objective called Stochastic Multimodal Fusion (SMF), which combines contrastive learning with self-supervised re-construction
        -  Modalities are randomly masked
        - The model is trained to align representations contrastively across different modality subsets
        - Reconstruct missing modality embeddings from the fused representation

**Strengths:**

- UrbanFusion is flexible wrt what modalities are available inference time. Even though it is outperformed by some baselines, this flexibility is a significant benefit to the community
- Strong (though not dominant) empirical results
- UrbanFusion can effectively generalize to new cities not seen in training
- Effectively integrates cross-modal information, while preserving intra-model information

**Weaknesses:**

- Empirical performance is not dominant. On tasks/datasets, UrbanFusion is outperformed by baselines
- Some overlap with prior work. Likely worth citing some of these:
    - Jenkins et al. CIKM’19. Unsupervised Representation Learning of Spatial Data via Multimodal Embedding
    - Yan et al. 2024. UrbanCLIP: Learning Text-enhanced Urban Region Profiling with Contrastive Language-Image Pretraining from the Web
- No estimates of variability (e.g., mean + std across several training runs) in experiments.

**Questions:**

- How do they define the locations? How much engineering is required to get UrbanFusion to work in a new city? What if the spatial resolution of the data in the new city does not match?
- What is the impact of the image encoder? DinoV3 (Siméoni et al. 2025) was pretrained extensively on satellite imagery; perhaps this would yield a meaningful performance increase? But this likely would not be a fair comparison to GeoCLIP. Could be worth studying in the future
- Why does the multimodal fusion encoder use average pooling instead of the CLS token?

---

> ### Author Response · Authors · 2025-11-21
>
> Thank you for your positive feedback to our paper! We address your remaining points in the following:
>
> [W1 Empirical performance] We agree that there are tasks where UrbanFusion is outperformed by other methods. We attribute this to
> * the size of the dataset: Due to computational resource constraints and data availability, we could not train on as large datasets as other methods such as GeoCLIP which uses a street view dataset that is 64x as big as ours. These methods are better on tasks that can be predicted well from SV images. When training the baseline methods on our dataset, our method performs best in almost all cases.
> * need for fusion: As analyzed empirically in Figure 3 and theoretically in Appendix D2, our method excels in tasks where unique and synergic information is required to solve the downstream tasks. This might not be necessary for all downstream tasks. Our method might thus be on par with other methods on some of the selected datasets, but has a much broader applicability to more complex applications.
>
> We have added these points to the Limitations-Section (see revised paper): “UrbanFusion achieves the state-of-the-art performance on a majority of tasks, with its few underperformances primarily due to the smaller dataset size for specific modalities (e.g. trained on 64x fewer SV images than GeoCLIP), while its strength lies in tasks requiring fusion to retain unique and synergistic information.”
>
> [W2 Citing prior work] Thank you for the suggestions. We have added them to the Related Work section.
>
> [W3 estimates of variability] First, note that we do report mean and std for the MLP results, see Table 10 in Appendix I. For linear downstream models, std is not relevant as it is deterministic. We did not provide the variability over pretraining runs, since pretraining is computationally expensive. We ask for your understanding, considering also that the baselines only reported results for a single pretrained model. To some extent, the experiment in section 4.6, where we retrain the base model with incomplete multimodal data, shows that the model performance of the main model is quite stable across runs, as 99.35% of the performance can be retained even with bimodal data. Furthermore, our newly added ablation study in Appendix C.6 shows that the performance is stable even across different architectural design choices (number of layers and pooling strategy).
>
> [Q1.1 definition of locations] We define locations as geographic coordinates (longitude, latitude). This is the standard case in work on location embedding (GeoCLIP, GAIR, etc), but of course, it poses difficulties to the applicability to data on other scales. We have clarified this in the Discussion section: "Limitations include imperfect temporal alignment across modalities and our focus on point- or postal code-level data in urban environments, which may limit applicability to rural areas or data on other scales."
>
> [Q1.2 apply to new city] UrbanFusion generalizes very well to new geographic regions without any engineering, as shown in Section 4.4., where we reported results on cities that were left out from the training data. Since street view images, remote sensing data, OSM maps, and POIs show similar patterns in different cities, the model generalizes well to downstream tasks in unseen cities without any finetuning. If finetuning is nevertheless desired, the engineering effort would be mainly the preparation of data (notably, this does not necessarily need to be all the modalities that the model was trained on, it could be a subset). We would also like to note that our published code already includes a LoRA implementation, enabling parameter-efficient finetuning of the pretrained model weights.
>
> [Q1.3 spatial resolution] Since we conduct inference on point data, the spatial resolution is the same. However, we showed in our experiments on the 29 ZIP Code tasks that the model also transfers to areal inference, in which we sample points from the area and average the point-wise inference results within each area.
>
> [Q2 DinoV3 encoder] Thank you for pointing this out. Indeed, DinoV3 or other encoders specifically trained on geographic data could improve the performance. We also agree that changing this for the experiments is not desirable since it would be an unfair comparison to the baselines. For a general learning conference as ICLR, we want to focus on comparing our learning strategy, SMF, with minimal changes to the architecture, to disentangle the effect of both. However, we published our code as open-source and with a modular setup that allows to easily replace individual components (see here https://anonymous.4open.science/r/SpatialFoundationModel-9551/srl/encoders/RS_encoder/RS_encoder.py ;  the encoder can easily be replaced by any other encoder in line 49).

---

> > ### Author Response · Authors · 2025-11-21
> >
> > [Q3 average pooling vs CLS] We chose average pooling over a dedicated CLS token because of recent analyses showing that, while the effect is task-dependent, average pooling often yields better performance across a wide range of models and settings [1]. We additionally now added an ablation study in Appendix C.6 showing that average pooling is superior over the CLS-token approach in most tasks:
> > | Task                           | Average pooling | CLS token
> > |--------------------------------|---------------------------|-------------------------|
> > | Housing prices                 | 78.7                      | 78.8                         |
> > | Energy consumption             | 20.1                      | 19.3                    |
> > | Crime incidence                | 87.4                      | 87.7                       |
> > | Urban perception (avg 6 tasks) | 9.5                       | 9.3                   |
> > | ZIP Code (avg 29 tasks)        | 74.3                      | 69.6                   |
> > | Land cover classification      | 56.9                      | 56.0                     |
> > | Land use (coarse)              | 58.9                      | 58.5                       |
> > | Land use (fine)                | 47.7                         | 50.5                          |
> >
> > [1] Ennadir, S., Zólyomi, L., Smirnov, O., Wang, T., Pertoft, J., Cornell, F., & Cao, L. (2025). Pool Me Wisely: On the Effect of Pooling in Transformer-Based Models. arXiv preprint arXiv:2510.03339.

---

> ### Author Response · Authors · 2025-12-03
> **Summary of rebuttal**
>
> Dear AC, we summarize our rebuttal to Reviewer qd7u in the following. The reviewer was very positive about our work. We answered to the remaining points as follows:
>
> **W1:** The reviewer observed that there are cases where our method is outperformed by other methods. We attribute this to larger training datasets of the baseline methods which helps for specific tasks (as expected when testing on 41 tasks in total), and the question whether modality-fusion is needed in all cases. We revised the discussion section to include these points.
>
> **W2:** Missing references - we added them to the related works section now.
>
> **W3:** Missing estimates of variability: We referred to Table 10 in Appendix I for variances of the downstream task model, and our analysis in section 4.6 and Appendix C.6 for evidence of low variance of the pretrained foundation model.
>
> **Q1:** The reviewer asked for the definition of locations and transferability on new cities. We clarified the questions and referred to our results in Section 4.4 showing strong generalization capabilities when applied to new cities zero-shot.
>
> **Q2:** The question came up whether other image encoders could improve performance. We agree, but, as the reviewer points out him/herself, using other encoders would not be a fair comparison to the baseline methods. We added instructions to our code base how to use another encoder.
>
> **Q3:** The reviewer asked why average pooling was used rather than the CLS token. In response, we conducted a new ablation experiment, showing that average pooling performs better empirically (see Appendix C.6).

---

### Official Review · Reviewer_jLmY · 2025-10-31

**Soundness:** 2
**Presentation:** 3
**Contribution:** 2
**Rating:** 4
**Confidence:** 3

**Summary:**

This paper presents UrbanFusion, a geofoundation model that learns unified spatial embeddings through Stochastic Multimodal Fusion (SMF). The model encodes heterogeneous urban data — including street-view images, remote sensing, maps, and POIs — using modality-specific encoders, and fuses them via a Transformer-based architecture. It is trained with a contrastive alignment objective between complementary modality subsets, together with latent reconstruction to address missing-modality scenarios. Evaluated across 41 tasks in 56 cities, UrbanFusion achieves state-of-the-art performance and strong cross-region generalization, demonstrating a practical and scalable paradigm for multimodal GeoAI.

**Strengths:**

•  Innovative training paradigm: The proposed Stochastic Multimodal Fusion (SMF) framework provides a practically meaningful contribution by integrating stochastic modality masking, symmetric InfoNCE alignment, and latent modality reconstruction into a unified training process.
•  Comprehensive evaluation: The paper conducts extensive and well-structured experiments across multiple tasks and scenarios, including coordinate-only, multimodal, and cross-region zero-shot generalization settings.
•  Reproducibility and practicality: The framework is conceptually simple, reproducible, and demonstrates clear potential for deployment in real-world multimodal GeoAI applications.

**Weaknesses:**

•  Limited architectural novelty: While the training objective is interesting, the overall fusion design (“multimodal tokens + a single-layer Transformer + average pooling”) closely resembles standard multimodal architectures. Consequently, the primary novelty lies in the learning strategy rather than in architectural design.
•  Insufficient ablation analysis: The paper lacks systematic ablations on key factors such as masking strategy and ratio, the impact of fusion depth (single- vs. multi-layer), and pooling mechanisms. Including these analyses would strengthen the methodological insight and clarify design choices.

**Questions:**

•  Could you report cross-domain results without concatenating raw coordinates, and analyze the model's sensitivity to coordinate normalization or projection?
•  Could you include ablations on the masking ratio/strategy, pooling mechanisms, and number of fusion layers (single vs. multi-layer)?
•  The paper positions multimodal fusion as a core contribution, yet most baselines are single-modal. This makes it difficult to disentangle the effect of the fusion mechanism from the benefit of using multiple modalities. While the "concatenated multimodal embeddings + linear probing" baseline is informative, it is not equivalent to native multimodal fusion. Please consider adding one or two native multimodal fusion baselines for a fairer comparison.

---

> ### Author Response · Authors · 2025-11-16
> **Request for clarification**
>
> Thank you for your constructive feedback. We are preparing detailed responses and additional results; however, we would appreciate a brief clarification to ensure we correctly address your comment regarding multimodal fusion baselines. You noted that "one or two native multimodal fusion baselines" would provide a fairer comparison. Our understanding is that, in the domain of learned location embeddings, existing multimodal approaches primarily rely on concatenation rather than learned fusion mechanisms. To the best of our knowledge, no prior work introduces a native fusion architecture applicable to our setting. If you had specific representative methods in mind (e.g., from related multimodal learning domains), we would be grateful for guidance so we can incorporate the most appropriate baselines.

---

> > ### Author Response · Authors · 2025-11-21
> >
> > [Q1: w/o concatenating raw coordinates] We conducted an additional experiment where we computed the performance on the downstream tasks without concatenating the coordinates. Full results are added in Appendix C.5; for convenience we copy the main results (coordinates-only location embedding) here:
> > | Task                           | UrbanFusion (with coords) | UrbanFusion (no coords) | Identity (only coords) |
> > |--------------------------------|---------------------------|-------------------------|------------------------|
> > | Housing prices                 | 78.7                      | 78.7                    | 66.2                   |
> > | Energy consumption             | 20.1                      | 20.1                    | 1.5                    |
> > | Crime incidence                | 87.4                      | 87.3                    | 22.1                   |
> > | Urban perception (avg 6 tasks) | 9.5                       | 9.4                     | 1.3                    |
> > | ZIP Code (avg 29 tasks)        | 74.3                      | 74.0                    | 3.0                    |
> > | Land cover classification      | 56.9                      | 57.1                    | 34.4                   |
> > | Land use (coarse)              | 58.9                      | 59.2                    | 48.2                   |
> > | Land use (fine)                | 47.7                      | 50.6                    | 42.7                   |
> >
> > The results show that there are minimal differences between UrbanFusion with or without concatenating coordinates, while using only the coordinates performs poorly in comparison. This shows that the performance gains can clearly be attributed to our embedding approach and not the coordinates.
> >
> > We incorporate raw geographic coordinates into the spatial representations used for evaluation to remain consistent with prior work such as SatCLIP, given that this information is always available. The coordinates are normalized to have zero mean and unit variance, which is standard practice for models employing weight decay. We acknowledge that raw geographic coordinates can be suboptimal under certain circumstances: they introduce artefacts near the poles, as extensively analyzed in prior research [1], where alternative encodings may be preferable. For the coordinate encoding within the spatial representation model itself, we follow the methodology using Equal Earth projection used by GeoCLIP and GAIR to ensure a fair comparison. Additionally, we conducted an in-depth analysis of spherical-harmonics–based coordinate encodings as a potential alternative, given their advantage of avoiding such artifacts entirely. However, we found this approach unsuitable for urban tasks due to its limited ability to capture high-frequency spatial variations, as discussed in detail in Chapter C.4.
> >
> > [1] Marc Rußwurm, Konstantin Klemmer, Esther Rolf, Robin Zbinden, and Devis Tuia. Geographic location encoding with spherical harmonics and sinusoidal representation networks. In Proceedings of the International Conference on Learning Representations (ICLR), 2024.
> >
> > [Q2: ablation studies] See [W2]
> >
> > [Q3 comparison to “native multimodal fusion”] Please see our comment above, where we requested clarification on this problem. We are afraid we did not fully understand this suggestion, since 1) all baselines claim to be multimodal (SatCLIP: location + RS, GeoCLIP: location + SV, GAIR: location + RS + SV, etc), and 2) we are unsure what “native multimodal fusion” refers to. If you refer to the fact that linear probing is not fusing the information, we refer to Appendix B, where we reported the performance of an MLP for comparison.

---

> > > ### Author Response · Authors · 2025-12-03
> > > **Summary of rebuttal**
> > >
> > > Dear AC, for convenience, we summarize our rebuttal to Reviewer jLmY in the following. Please see above for full responses.
> > >
> > > **W1:** The reviewer called it a weakness that “primary novelty lies in the learning strategy rather than in architectural design”. We believe that this is actually a positive point - a new learning strategy for multimodal learning is, from our point of view, a much stronger contribution than minor architectural changes.
> > >
> > > **W2:** The reviewer suggested additional ablation studies. As suggested, we added ablations on the number of layers, the architecture, and the pooling mechanism in Appendix C.6, showing that our chosen design shows best performance overall.
> > >
> > > **Q1:** The reviewer asked about the influence of our choice of concatenating the raw coordinates to the learnt embeddings. We conducted an additional experiment and found that the differences with and without concatenating the raw coordinates are negligible (see Appendix C.5).
> > >
> > > **Q2:** [see W2]
> > >
> > > **Q3:** The reviewer asked us to compare to “native multimodal fusion”. Since this point was unclear to us, we commented on OpenReview two days after the reviews were released and asked for clarification, but never received a response. We believe this point is most likely a misunderstanding, since there has been no fusion method in the related work so far.
> > >
> > > Thank you for your consideration.

---

> ### Author Response · Authors · 2025-11-21
>
> Thank you for your positive feedback to our paper! We address your remaining points in the following:
>
> [W1 novelty of learning strategy vs architecture] Thank you for pointing this out. We fully agree that our work indeed focuses on a new learning paradigm rather than architectural complexity. Fusing multiple modalities with a Transformer is standard practice, but this is not sufficient for learning a joint embedding without having labeled data. In prior work, embeddings were learnt with contrastive learning between all pairs of modalities. As analyzed in Section 2, this poses limitations in what modality interactions they can represent, and scales poorly with the number of modalities. The central contribution of UrbanFusion is the Stochastic Multimodal Fusion (SMF) training strategy, which alleviates these issues, and enables flexible fusion under arbitrary missing-modality patterns and is theoretically grounded in partial information decomposition (PID), proof in Appendix D.2. This is also the aspect highlighted as a strength by Reviewer qd7u and Reviewer jLmY, who noted that our approach provides flexibility, robustness to incomplete modalities, and an innovative training objective.
>
> [W2 Ablation analysis] Thank you for this valuable suggestion.  We keep the fusion module intentionally simple to avoid confounding factors and to highlight the impact of the SMF learning strategy. Now we have conducted additional ablation studies to understand the impact of the number of layers, pooling mechanism, and model architecture (LSTM vs Transformer). For each variation, we reported the results across all downstream tasks in the new Appendix C.6. In summary:
> 1. The results are consistent across architectural configurations, suggesting that the model’s performance primarily stems from the learning framework that is robust to architectural variations.
> 2. Transformers outperform LSTMs, with single-layer Transformer encoders achieving the strongest overall performance.
> 3. Both pooling approaches perform well, though average pooling shows a slight advantage on most tasks, confirming the parameter choices made for our main results.
>
> Our experiments in Section 4.6 and Appendix C.1 (Training With Incomplete Multimodal Data) already provide insights into masking behavior: The bimodal setting corresponds to a fixed masking strategy in which all modalities except one are masked in each forward pass, while the partial setting effectively performs modality dropout with the constraint that at least one modality remains. Thus, these results implicitly evaluate how different masking strategies affect performance.

---

### Meta-Review · Area_Chair_vV6v · 2025-12-23

**Summary:**

Reviewers consistently questioned the level of novelty of the proposed method. While the Stochastic Multimodal Fusion (SMF) framework is clearly presented and empirically validated, its main contribution is viewed as a training strategy rather than a fundamentally new modeling or learning principle. Additional concerns included the simplicity of the fusion architecture, limitations of rasterized OSM/cartographic encoding, and the scope of the “foundation model” claim relative to the evaluated tasks.

**Reviewer Concerns:**

The rebuttal addressed several technical and clarity issues, including coordinate concatenation, additional ablations, and fusion vs. concatenation baselines, which improved the rigor and fairness of the evaluation. However, core concerns remain: some reviewers remain unconvinced that SMF offers sufficient conceptual novelty beyond existing multimodal contrastive or masked-reconstruction frameworks, and concerns about cartographic data representation persist.

**Reviewer Scores:**

Reviewer jLmY may have become slightly more positive after the rebuttal. Reviewers Uw1v and ijWt are unlikely to significantly change their assessments. Overall reviewer sentiment remains mixed but leans negative.

---

### Decision · Program_Chairs · 2026-01-26

Reject